

# Yangian bootstrap for massive Feynman integrals

Florian Loebbert[1⋆], Julian Miczajka[1†], Dennis Müller[2‡] and Hagen Münkler[3∘]

**1** Institut für Physik, Humboldt-Universität zu Berlin,
Zum Großen Windkanal 6, 12489 Berlin, Germany
**2** Niels Bohr Institute, Copenhagen University, Blegdamsvej 17, 2100 Copenhagen, Denmark
**3** Institut für Theoretische Physik, Eidgenössische Technische Hochschule Zürich,
Wolfgang-Pauli-Strasse 27, 8093 Zürich, Switzerland

⋆ loebbert@physik.hu-berlin.de, † miczajka@physik.hu-berlin.de,
‡ dennis.mueller@nbi.ku.dk, ∘ muenkler@itp.phys.ethz.ch

## Abstract

We extend the study of the recently discovered Yangian symmetry of massive Feynman integrals and its relation to massive momentum space conformal symmetry. After proving the symmetry statements in detail at one and two loop orders, we employ the conformal and Yangian constraints to bootstrap various one-loop examples of massive Feynman integrals. In particular, we explore the interplay between Yangian symmetry and hypergeometric expressions of the considered integrals. Based on these examples we conjecture single series representations for all dual conformal one-loop integrals in $D$ spacetime dimensions with generic massive propagators.



# 1 Introduction and Summary

Conformal symmetry plays an important role in theoretical physics. On the one hand it represents an (approximate) symmetry of many interesting models. On the other hand, it furnishes a powerful tool that puts strong constraints on a theory's observables and leads to intriguing mathematical structures. In this paper we explore an extension of conformal symmetry into two directions: its applicability to situations with masses as well as its embedding into an infinite dimensional Yangian symmetry. While there is a clear phenomenological motivation for going beyond the realm of massless particles, the extension to a conformal Yangian brings us to the theory of integrable models where one may expect that physical quantities of interest are fixed completely by the underlying symmetry.

While conformal symmetry can be studied on different levels of a given model, here we are interested in its impact on the elementary building blocks of quantum field theory, i.e. on Feynman integrals. The focus of the present paper lies on the question of how to bootstrap massive Feynman integrals by using conformal symmetry or its Yangian extension. In fact, we will discuss two instances of conformal symmetry, i.e. an 'ordinary' conformal symmetry that is here naturally formulated in momentum space and dual conformal symmetry acting on dual region momenta. In both cases we will discuss representations of the symmetry that also act on the particles' masses, which can be interpreted as extra-dimensional components of the

coordinate vectors. The Yangian algebra employed here is then understood as the closure of these two conformal algebras, cf. [1–3].

The dual conformal symmetry of certain Feynman integrals has a long history in the massless as well as in the massive situation, see e.g. [4–10], and it strongly reduces the number of variables a function of interest depends on. Among the dual conformal Feynman integrals, infinite classes of diagrams of fishnet structure feature an even larger Yangian symmetry as was shown in [2, 11] for the massless case and a massive version was recently found in [3]. The Yangian algebra is well known to underly rational integrable models [12–15], and it includes the dual conformal symmetry at the zeroth level of its infinite set of generators. In the case of two-dimensional field theories, this nonlocal symmetry typically fixes the scattering matrix completely [16]. The distinguished role of fishnet-type Feynman integrals can be understood from the fact that their conformal Yangian symmetry is inherited from planar $\mathcal{N} = 4$ super Yang–Mills (SYM) theory via a particular double scaling limit of its gamma-deformation. This double scaling limit yields a massless fishnet theory [17], whose correlators or scattering amplitudes are in one-to-one correspondence with individual Feynman graphs of fishnet structure. Similarly, a massive fishnet theory can be obtained from a double-scaling limit of $\mathcal{N} = 4$ SYM theory on the Coulomb branch, allowing to identify massive Feynman integrals with Yangian invariant scattering amplitudes [18]. Also the massive version of the Yangian can be understood as the closure of massive dual conformal symmetry and a novel massive extension of ordinary conformal symmetry [3]. In this paper we will study the constraints of the Yangian and its (dual) conformal sub-algebras.

The idea to bootstrap Feynman integrals using their Yangian symmetry was first discussed in [19] for the examples of the massless box, hexagon and double box integrals. While the 2-variable box integral was shown to be completely fixed by its symmetries, in this first approach it was not possible to fix the linear combination of formal Yangian invariant building blocks for the 9-variable hexagon and double box integrals. Here an important step was recently made in [20], where this linear combination was determined using a multi-variable extension of the Mellin–Barnes techniques, cf. [21, 22]. In order to refine the algorithmic approach towards Feynman integrals from Yangian symmetry it would be desirable to study examples that interpolate between the above 2- and 9-variable cases. Here the recent extension of Yangian symmetry to Feynman integrals with massive propagators comes in handy, since switching on individual masses allows us to slowly increase the number of variables. In fact, initial examples of massive integrals were obtained from Yangian symmetry in [3].

In the present paper we expand on the details of the massive Yangian and conformal symmetry presented in [3]. First we prove the symmetry statements at one- and two-loop orders in detail. We then elaborate on the relation between Yangian symmetry and the massive extension of momentum space conformal symmetry. We explicitly discuss the implications of this momentum space symmetry on a few examples in Section 6. In the massless limit, the resulting constraints are precisely those that have recently been studied in the context of the momentum space conformal bootstrap (see e.g. [23–33]) with applications in cosmology or condensed matter physics.

We then systematically apply the bootstrap approach to massive Feynman integrals with generic propagator powers at one loop order. Since the treatment strongly depends on the choice of variables, it is useful to discuss different examples in detail, even if simpler cases can (in principle) be obtained as limits of more complex cases. In particular, we will discuss

representations of the considered integrals in terms of different hypergeometric functions. The results are summarized in the following table:

| Points | Dual Conf. | Masses | Ratios$_{\text{Parameters}}$ | Solution Basis | Section |
|--------|-----------|--------|------------------------------|----------------|---------|
| 2 | no/yes | 00 | $0_3/0_2$ | rational/rational | 7.2/— |
| 2 | no/yes | $m_1 0$ | $1_3/0_2$ | Gauß $_2F_1$/rational | 7.3/8.1 |
| 2 | no/yes | $m_1 m_2$ | $2_3/1_2$ | Kampé de Fériet/Legendre | 7.4/8.2 |
| 3 | no/yes | 000 | $2_4/0_3$ | Appell $F_4$/rational | 6.1/(6.2) |
| 3 | no/yes | $m_1 00$ | $3_4/1_3$ | Lauricella/Gauß $_2F_1$ | 7.6/8.4 |
| 3 | no/yes | $m_1 m_2 m_3$ | $5_4/3_3$ | —/Srivastava $H_C$ | —/8.5 |
| $n$ | no/yes | $m_1 \dots m_n$ | see eq. (2.30) | —/conjecture | —/9 |

Based on the intuition gained with these examples, we finally conjecture two different series representations for the most generic massive $n$-point integrals at one-loop order, with propagator powers $a_j$ that obey the dual conformal constraint, i.e. they add up to the spacetime dimension $D = \sum_j a_j$:

$$\int \frac{\mathrm{d}^D x_0}{\prod_{j=1}^n (x_{0j}^2 + m_j^2)^{a_j}} = \quad \text{(diagram)} \quad . \tag{1.1}$$

The two conjectured series representations correspond to expressing the integral in terms of two different sets of variables

$$\text{Region A:} \quad u_{ij} = \frac{x_{ij}^2 + (m_i - m_j)^2}{-4 m_i m_j}, \qquad \text{Region B:} \quad v_{ij} = \frac{x_{ij}^2 + m_i^2 + m_j^2}{2 m_i m_j}. \tag{1.2}$$

The A-series represents an $n$-point generalization of Gauß' hypergeometric function $_2F_1$ and Srivastava's triple hypergeometric series $H_C$ for 2 and 3 points, respectively. The B-series closely resembles a representation given by Aomoto [34].[1] Moroever we note that the $v$-type variables are distinguished since in the case of unit propagator powers $a_j = 1$, elegant polylogarithmic expressions for this class of integrals are known up to five points [35]. Interestingly, this family of all-mass $n$-gon integrals has been found to have beautiful relations to geometry, see e.g. [35–40]. Our analysis suggests that this beauty is closely connected to the underlying Yangian symmetry which essentially fixes these integrals completely. Note the hint at integrability in the 1998 paper [36] by Davydychev and Delbourgo, where the simplification of the integral representation for the constraint $\sum_j a_j = D$ was already called a "generalization of the so-called *uniqueness* formula for massless triangle diagrams".[2] In the non-dual-conformal case of unconstrained propagator powers, hypergeometric representations for these one-loop integrals were obtained in [42].

Throughout this paper we use the following notation for non-dual-conformal and dual conformal one-loop $n$-point integrals with propagator weights $a_j$, respectively, which is also reflected in the subsection titles (cf. the above table of contents):

$$I_n^{m_1 \dots m_n}[a_1, \dots, a_n], \qquad I_{n\bullet}^{m_1 \dots m_n}[a_1, \dots, a_n]. \tag{1.3}$$

In the dual conformal case denoted by $\bullet$, the propagator weights obey $\sum_j a_j = D$. We close the paper with an outlook in Section 10.

---

[1] We thank Christian Vergu for bringing this to our attentention.

[2] The *uniqueness formula* is also called the *star-triangle relation* which represents a characteristic feature of many integrable models, cf. (6.2) and e.g. [41].

## 2 Massive Dual Conformal Symmetry

In this section we discuss the massive dual conformal symmetry of Feynman integrals. Integrals with this symmetry are particularly interesting in the context of the present paper since only these are invariant under the whole tower of Yangian generators and thus maximally constrained. The distinguished feature of these integrals is that the powers of propagators entering into an integration vertex obey the dual conformal constraint

$$\sum_{j=1}^{n} a_j = D \,. \tag{2.1}$$

After having discussed the case of one-loop integrals in large detail, we will comment on generalizations to higher loop orders.

**One-Loop Integrals and Dual Conformal Transformations.** We begin by discussing the dual conformal symmetry of Feynman integrals with massive external legs. As a simple example, we consider the one-loop $n$-gon integral with arbitrary propagator powers,

$$I_n = \int \frac{\mathrm{d}^D x_0}{\prod_{j=1}^{n} (x_{0j}^2 + m_j^2)^{a_j}} = \quad . \tag{2.2}$$

The above integral is immediately invariant under four-dimensional Poincaré transformations. In order to have an object that is additionally scale and conformally invariant, we introduce a prefactor $V_n$,

$$I_n = V_n \phi_n \,. \tag{2.3}$$

Any appropriate prefactor will lead to scale invariance of the combined object under simultaneous rescalings of the $x$-coordinates and the masses. If the propagator weights satisfy the constraint (2.1), the function $\phi_n$ is also invariant under the $(D+1)$-dimensional inversion

$$\mathcal{I} : x^{\hat{\mu}} \mapsto \frac{x^{\hat{\mu}}}{\hat{x}^2} \,. \tag{2.4}$$

Here, we note that the index $\hat{\mu}$ runs from 1 to $D+1$ and the additional component of the vectors $x_j$ is given by $x_j^{D+1} = m_j$. To denote an index running from 1 to $D$, we employ the unhatted version $\mu$. Moreover, we use the abbreviation

$$\hat{x}^2 = x^2 + m^2 \,. \tag{2.5}$$

For the action of the inversion map, we note that

$$(x_{0j}^2 + m_j^2) \mapsto \frac{(x_{0j}^2 + m_j^2)}{x_0^2 (x_j^2 + m_j^2)} \,, \tag{2.6}$$

where we have applied the ordinary $D$-dimensional inversion to $x_0$, which can be achieved by using an appropriate substitution under the integral. Correspondingly, the integral transforms as

$$I_n \mapsto \int \frac{\mathrm{d}^D x_0 (x_0^2)^{-D + \sum a_j} \prod_{j=1}^{n} (x_j^2 + m_j^2)^{a_j}}{\prod_{j=1}^{n} (x_{0j}^2 + m_j^2)^{a_j}} \,. \tag{2.7}$$

If the conformal constraint (2.1) is satisfied, the integral thus transforms by a factor. For the function $\phi_n$ to be an invariant, we hence need to construct a prefactor which transforms as

$$V_n \mapsto V_n \prod_{j=1}^{n} (x_j^2 + m_j^2)^{a_j}. \tag{2.8}$$

There are several possible choices for such a prefactor, in particular since we can obtain the respective scalings from combinations of $\hat{x}_{ij}^2$ or $m_i$. A simple choice of prefactor satisfying the above constraint is given by

$$V_n = \prod_{j=1}^{n} m_j^{-a_j}, \tag{2.9}$$

however, other choices of prefactors can be more convenient and we will also employ different ones below.

The combination of the above inversion with $D$-dimensional translations yields the special conformal transformations in $D+1$ dimensions,

$$x^{\hat{\mu}} \mapsto \frac{x^{\hat{\mu}} + c^{\hat{\mu}} x_{\hat{\nu}} x^{\hat{\nu}}}{1 + 2c_{\hat{\nu}} x^{\hat{\nu}} + c_{\hat{\rho}} c^{\hat{\rho}} x_{\hat{\nu}} x^{\hat{\nu}}}, \tag{2.10}$$

albeit with the extra-dimensional component $c^{D+1}$ set to zero, since $I_n$ is not invariant under the respective translation. These transformations are generated by the conformal generator

$$\tilde{K}^{\mu} = \sum_{j=1}^{n} \tilde{K}_j^{\mu}, \qquad\qquad \tilde{K}_j^{\mu} = -i\left(2x_j^{\mu} x_j^{\hat{\nu}} - \eta^{\mu\hat{\nu}} \hat{x}_j^2\right)\partial_{\hat{\nu}}. \tag{2.11}$$

Next we note that the invariance of $\phi_n$ under the above generator can be translated to an invariance statement for $I_n$ with an adapted generator,

$$0 = \tilde{K}^{\mu}\phi_n = V_n^{-1} K^{\mu} I_n, \qquad K^{\mu} = \tilde{K}^{\mu} + V_n \tilde{K}^{\mu} V_n^{-1} = \tilde{K}^{\mu} - 2i\sum_{j=1}^{n} a_j x_j^{\mu}, \tag{2.12}$$

which is easy to see using the explicit prefactor given in (2.9) but holds for any prefactor satisfying (2.8). The integral $I_n$ is hence invariant under the dual conformal generators

$$P_j^{\mu} = -i\partial_{x_j}^{\mu}, \qquad\qquad L_j^{\mu\nu} = ix_j^{\mu}\partial_{x_j}^{\nu} - ix_j^{\nu}\partial_{x_j}^{\mu},$$
$$D_j = -ix_{j\hat{\mu}}\partial_{x_j}^{\hat{\mu}} - i\Delta_j, \qquad K_j^{\mu} = -i\left(2x_j^{\mu} x_j^{\hat{\nu}} - \eta^{\mu\hat{\nu}} \hat{x}_j^2\right)\partial_{\hat{\nu}} - 2i\Delta_j x_j^{\mu}, \tag{2.13}$$

if we set the weights $\Delta_j$ equal to the propagator powers $a_j$. That is, for $J^a$ denoting one of the above generators we have

$$J^a I_n = 0. \tag{2.14}$$

We remind the reader that the index $\hat{\mu}$ runs from 1 to $D+1$ while $\mu$ runs from 1 to $D$. The generators can hence also be understood as massless generators in $D+1$ dimensions. Note however that in order to have invariance, we have to restrict to indices $\mu, \nu$, e.g. the integral is not invariant under translations in the mass dimension. The generators given above satisfy the conformal algebra

$$\left[D_j, P_k^{\hat{\mu}}\right] = i\delta_{jk} P_k^{\hat{\mu}}, \qquad\qquad \left[D_j, K_k^{\hat{\mu}}\right] = -i\delta_{jk} K_k^{\hat{\mu}},$$
$$\left[P_j^{\hat{\mu}}, L_k^{\hat{\nu}\hat{\rho}}\right] = i\delta_{jk}\left(\eta^{\hat{\mu}\hat{\nu}} P_k^{\hat{\rho}} - \eta^{\hat{\mu}\hat{\rho}} P_k^{\hat{\nu}}\right), \qquad \left[K_j^{\hat{\mu}}, L_k^{\hat{\nu}\hat{\rho}}\right] = i\delta_{jk}\left(\eta^{\hat{\mu}\hat{\nu}} K_k^{\hat{\rho}} - \eta^{\hat{\mu}\hat{\rho}} K_k^{\hat{\nu}}\right),$$
$$\left[K_j^{\hat{\mu}}, P_k^{\hat{\nu}}\right] = 2i\delta_{jk}\left(\eta^{\hat{\mu}\hat{\nu}} D_k - L_k^{\hat{\mu}\hat{\nu}}\right), \qquad \left[L_j^{\hat{\mu}\hat{\nu}}, L_k^{\hat{\rho}\hat{\sigma}}\right] = i\delta_{jk}\left(\eta^{\hat{\mu}\hat{\sigma}} L_k^{\hat{\rho}\hat{\nu}} + (3\text{ more})\right). \tag{2.15}$$

On a massless leg $j$, the same representation applies with $m_j \equiv x_j^{D+1} = 0$.

Summing up, we have found that the one-loop graph (2.2) is invariant under the dual conformal generators (2.13) provided that the propagator weights satisfy the constraint $\sum a_j = D$.

**Higher Loop Integrals.** The above invariance statement carries over to higher loop graphs if we demand that at each vertex, the joining propagator weights sum up to the spacetime dimension,

$$\sum_{\text{vertex}} a_j + \sum_{\text{vertex}} b_k = D. \tag{2.16}$$

Here, the variables $a_j$ denote the weights of external propagators whereas the $b_k$ correspond to internal propagators. In order to see this, consider a multi-loop integral of the form

$$I = \dots \int \frac{\mathrm{d}^D y_i}{\rho_i \sigma_i} \dots, \tag{2.17}$$

with

$$\rho_i = \prod_{j \in V_i} \left[ (x_j - y_i)^2 + m_j^2 \right]^{a_j}, \qquad \sigma_i = \prod_{k \in \tilde{V}_i} (y_k - y_i)^{2b_{ki}}, \tag{2.18}$$

where $V_i$ and $\tilde{V}_i$ denote the set of external or internal points connected to $y_i$. The internal propagators need to be massless in order to have dual conformal symmetry, since we are not integrating over the $(D+1)$-component of the internal points, i.e. the mass. After carrying out the inversion given in (2.6), we pick up a factor of

$$\left[ y_i^2 \right]^{\sum_{j \in V_i} a_i + \sum_{k \in V_i} b_{ki} - D}.$$

Given the above constraint, this factor cancels at each vertex.

**Conformal Variables.** Due to its invariance, the function $\phi_n$ can be parametrized in terms of conformally invariant variables, simplifying its functional form considerably. In the massless case, the natural variables are the well-known conformal four-point cross ratios. In the massive case, there are (at least) three natural kinds of massive conformal variables:

$$u_{ij} = \frac{m_i m_j}{\hat{x}_{ij}^2}, \qquad v_{ij}^k = \frac{m_k^2 \hat{x}_{ij}^2}{\hat{x}_{ik}^2 \hat{x}_{jk}^2}, \qquad w_{ij}^{kl} = \frac{\hat{x}_{ij}^2 \hat{x}_{kl}^2}{\hat{x}_{ik}^2 \hat{x}_{jl}^2}. \tag{2.19}$$

Here we use the abbreviation

$$\hat{x}_{ij}^2 = x_{ij}^2 + (m_i - m_j)^2. \tag{2.20}$$

Sometimes it is useful to multiply these variables by overall constants, to add constants to them, or to consider the inverse of these variables, which may lead to a more natural form of the resulting differential equations. Clearly, the $v_{ij}^k$ and the $w_{ij}^{kl}$ are not independent of the $u_{ij}$, since

$$v_{ij}^k = \frac{u_{ik} u_{jk}}{u_{ij}}, \qquad w_{ij}^{kl} = \frac{u_{ik} u_{jl}}{u_{ij} u_{kl}} = \frac{v_{ij}^k}{v_{lj}^k}. \tag{2.21}$$

Therefore, in the general $n$-point case with all masses non-vanishing, one would try to find an independent set among the $n(n-1)/2$ different $u_{ij}$. However, the other cross ratios become important as soon as we consider special cases where some of the masses are set to zero. Setting $k$ masses to zero reduces the number of degrees of freedom by $k$, but it leads to the vanishing of $\sum_{i=1}^{k}(n-i)$ of the $u_{ij}$. Therefore, one may need to extend the set of non-vanishing $u_{ij}$ by an independent subset of the $v_{ij}^k$ and $w_{ij}^{kl}$. We note that we can and will use the freedom to select a set of independent cross ratios in order to simplify the form of the Yangian PDEs.

**Independent Variables.** In the general case it can be difficult to make sure that a given set of the above variables is indeed independent, and which combinations of values can be reached by choosing appropriate $x_j^{\hat{\mu}}$. In order to answer such questions systematically, we can employ the construction of the Dirac cone [43], which is also used in the construction of the conformal compactification of Minkowski or Euclidean space. To this end, we map $x^{\hat{\mu}}$ to a $(D+3)$-dimensional lightlike vector with components

$$X^0 = 1 + \hat{x}^2, \qquad X^{\hat{\mu}} = 2x^{\hat{\mu}}, \qquad X^{D+2} = 1 - \hat{x}^2. \qquad (2.22)$$

We can map $X$ back to $x$ via

$$x^{\hat{\mu}} = \frac{X^{\hat{\mu}}}{X^0 + X^{D+2}}. \qquad (2.23)$$

Note that the latter mapping is invariant under a rescaling of $X$ and hence we consider equivalence classes of lightlike vectors $X$ or the light-cone in projective space.

The main advantage of this approach is that conformal transformations of $x$ correspond to linear mappings of $X$, which makes them much easier to treat. In our concrete case we act with transformations belonging to $\mathrm{SO}(1, D+1)$, embedded in such a way that it acts trivially on the $(D+1)$-component of $X$, which corresponds to the mass component of the spacetime vector $x$.

In the following we consider a configuration of 4 points $x_i^{\hat{\mu}}$ and successively exhaust our freedom to employ $\mathrm{SO}(1, D+1)$ transformations in order to reach a set of fixed configurations. This approach gives a different parametrization of the conformally invariant degrees of freedom of the configuration. It has the advantage that the variables we obtain are independent by construction and their range is clear. We can then check if a given set of (generalized) conformal cross ratios of the form (2.19) is indeed independent by expressing them in terms of the new variables.

We employ the notation

$$[X] = \left[ X^0 : X^{D+2} : X^{\hat{\mu}} \right], \qquad (2.24)$$

such that the (massive) conformal symmetry $\mathrm{SO}(1, D+1)$ keeps the last element of the above vector fixed. We consider the case of at least one of the external legs being massive and assume without loss of generality that $m_1 \neq 0$. The case of all masses vanishing was discussed in detail in [19]. Next, note that the vector

$$\left( X_1^0, X_1^{D+2}, X_1^{\mu} \right)$$

is timelike (since we are leaving out a nonvanishing spatial component of a lightlike vector), and we can hence find a transformation in $\mathrm{SO}(1, D+1)$, such that

$$[X_1] = [1 : 0 : \ldots : 0 : 1]. \qquad (2.25)$$

We note that this corresponds to

$$x_1 = (0, \ldots, 0, 1) \qquad (2.26)$$

and we have effectively used the freedom to scale our variables to set $m_1 = 1$ and the remaining masses are effectively measured in units of $m_1$, which we make explicit in the following by using the notation $\tilde{m}_i = m_i/m_1$.

Table 1: Number of degrees of freedom for $n$ massive particles in $D$ dimensions after exhausting dual conformal symmetry.

| $n$ | $D = 3$ | $D = 4$ | $D = 5$ | $D = 6$ |
|---|---|---|---|---|
| 2 | 1 | 1 | 1 | 1 |
| 3 | 3 | 3 | 3 | 3 |
| 4 | 6 | 6 | 6 | 6 |
| 5 | 10 | 10 | 10 | 10 |
| 6 | 14 | 15 | 15 | 15 |
| 7 | 18 | 20 | 21 | 21 |

The stabilizer of the above vector $X_1$ in $SO(1, D+1)$ is given by the obvious $SO(D+1)$ and fixing the following vectors is a straight-forward exercise leading to the configuration

$$[X_2] = \left[1 + \tilde{m}_2^2 : 1 - \tilde{m}_2^2 : 0 : 0 : 0 : 0 : 2\tilde{m}_2\right], \tag{2.27}$$

$$[X_3] = \left[1 + z_1^2 + \tilde{m}_3^2 : 1 - z_1^2 - \tilde{m}_3^2 : 2z_1 : 0 : 0 : 0 : 2\tilde{m}_3\right], \tag{2.28}$$

$$\left[X_4\right] = \left[1 + z_2^2 + z_3^2 + \tilde{m}_4^2 : 1 - z_2^2 - z_3^2 - \tilde{m}_4^2 : 2z_2 : 2z_3 : 0 : 0 : 2\tilde{m}_4\right]. \tag{2.29}$$

Clearly, after fixing $n$ points, we have a stabilizer of $SO(D+2-n)$, provided that $n \le D+1$. The number of independent, conformally invariant variables is thus given by

$$n(D+1) - \dim(SO(1, D+1)) + \theta(D+1-n)\dim(SO(D+2-n))$$
$$= n(D+1) - \frac{1}{2}(D+1)(D+2) + \frac{\theta(D+1-n)}{2}(D+1-n)(D+2-n), \tag{2.30}$$

see also Table 1 for the case of few particles. The above derivation assumes that at least one of the masses is non-vanishing. We can conclude that, as long as one non-vanishing mass remains, we only need to subtract one for every constraint such as masses being equal or vanishing in order to find the corresponding number of degrees of freedom. In fact, for $n \ge 3$, this procedure remains valid for the case of all masses vanishing, cf. Appendix A in [19].

**Example: 3 Points, $m_1 m_2 m_3$.** As a simple example, we consider the case of three external points and three distinct, non-zero masses. We take the generalized conformal cross ratios to be

$$u = -\frac{u_{12}^{-1}}{4} = \frac{\hat{x}_{12}^2}{-4m_1 m_2}, \qquad v = -\frac{u_{13}^{-1}}{4} = \frac{\hat{x}_{13}^2}{-4m_1 m_3}, \qquad w = -\frac{u_{23}^{-1}}{4} = \frac{\hat{x}_{23}^2}{-4m_2 m_3}. \tag{2.31}$$

From the configurations obtained above, we note (setting $D = 4$ for the moment),

$$x_1 = (0, 0, 0, 0, 1), \qquad x_2 = (0, 0, 0, 0, \tilde{m}_2), \qquad x_3 = (z_1, 0, 0, 0, \tilde{m}_3). \tag{2.32}$$

The cross ratios are thus given by

$$u = -\frac{(1-m_2)^2}{4m_2}, \qquad v = -\frac{(1-m_3)^2 + z_1^2}{4m_3}, \qquad w = -\frac{(m_2 - m_3)^2 + z_1^2}{4m_2 m_3}. \tag{2.33}$$

These expressions can in principle be employed to find out what values the triple $(u, v, w)$ can take by solving for $m_i$.

# 3 Massive Yangian Symmetry

In this section, we show that one- and two-loop diagrams with massive external propagators are Yangian invariant. The Yangian algebra extends an underlying Lie algebra symmetry to an infinite tower of symmetry generators, grouped into levels $n$. For the levels zero (J) and one $(\widehat{J})$, we note the commutation relations

$$\left[J^a, J^b\right] = f^{ab}{}_c J^c, \qquad\qquad \left[J^a, \widehat{J}^b\right] = f^{ab}{}_c \widehat{J}^c. \qquad (3.1)$$

Higher level generators can be constructed by repeated commutations of level-one generators. These commutators are constrained by the Serre relations, cf. e.g. [14].

In our case, the generators of the Yangian algebra are constructed from the generator densities of massive, dual-conformal symmetry given in (2.13).[3] These generators combine to form level-zero and level-one generators on $n$-point functions

$$J^a = \sum_{j=1}^{n} J^a_j, \qquad\qquad \widehat{J}^a = \tfrac{1}{2} f^a{}_{bc} \sum_{j<k} J^c_j J^b_k + \sum_{j=1}^{n} s_j J^a_j. \qquad (3.2)$$

Here, $f^a{}_{bc}$ denote the dual structure constants of the above massive, dual-conformal algebra, i.e. the spacetime indices are summed from 1 to $D+1$. Introducing a free parameter $y$, the level-one momentum generator reads

$$\widehat{P}^\mu = \tfrac{i}{2} \sum_{j<k} \left(P^\mu_j D_k + P_{j\,\nu} L^{\mu\nu}_k - (j \leftrightarrow k)\right) + \sum_{j=1}^{n} s_j P^\mu_j + y \widehat{P}^\mu_{\text{extra}}, \qquad (3.3)$$

where

$$\widehat{P}^\mu_{\text{extra}} = \tfrac{i}{2} \sum_{j<k} \left(P_{jD+1} L^{\mu D+1}_k - (j \leftrightarrow k)\right). \qquad (3.4)$$

The other Yangian level-one and extra generators are listed in (A.1) and (A.2). Setting $y = 1$ corresponds to the choice of considering the whole algebra $\mathfrak{so}(1, D+2)$ for the construction of the level-one Yangian generators. Leaving out the contribution to the summation from $\hat{\mu} = D+1$ corresponds to setting $y = 0$. It is interesting to note that at the one-loop level $\widehat{P}^\mu$ is a symmetry for any value of $y$, as we will see below.

Let us pause here for a moment and introduce some additional notation. We denote a generator acting on the sites $l$ through $r$ of an $n$-site object as

$$\widehat{J}^a_{(l,r),n} = \tfrac{1}{2} f^a{}_{bc} \sum_{k>j=l}^{r} J^c_j J^b_k + \sum_{j=l}^{r} s^{(n)}_j J^a_j, \qquad\qquad J^a_{(l,r),n} = \sum_{j=l}^{r} J^a_j. \qquad (3.5)$$

We will drop the subscripts if they are clear from context. For a 2-point Yangian level-one generator acting on legs $j$ and $k$, we introduce the notation

$$\widehat{J}^a_{jk} = \tfrac{1}{2} f^a{}_{bc} J^c_j J^b_k + s^{(2)}_j J^a_j + s^{(2)}_k J^a_k. \qquad (3.6)$$

---

[3]We have verified the Serre relations for the generators in (2.13) for $D = 2, 3, 4$ [44]. Note that for our bootstrap purposes below we have solely used the level-zero and level-one symmetries without an appeal to the infinite tower of Yangian generators.

## 3.1 The Symmetry at One Loop

We show that the above level-one generator is a symmetry of generic scalar $n$-point Feynman integrals at one-loop order with massive propagators,

$$I_n = \int \frac{\mathrm{d}^D x_0}{\prod_{j=1}^n (x_{0j}^2 + m_j^2)^{a_j}} = \quad . \tag{3.7}$$

The propagator powers $a_j$ and the spacetime dimension $D$ are arbitrary, and we use the notation $x_{jk}^\mu = x_j^\mu - x_k^\mu$. These integrals are invariant under all permutations of the external legs which are accompanied by the respective permutations of the propagator weights $a_j$ and the masses $m_j$. It turns out that already the integrand is invariant, which implies that

$$\widehat{\mathrm{J}}^a I_n = 0. \tag{3.8}$$

Before we go on to prove the above result, let us discuss the implications of the permutation symmetry of the $n$-gon integral $I_n$. Concretely, we consider the Yangian level-one generator

$$\widehat{\mathrm{J}}^a_{(1,n),n} = \tfrac{1}{2} f^a{}_{bc} \sum_{k>j=1}^n \mathrm{J}_j^c \mathrm{J}_k^b + \sum_{j=1}^n s_j^{(n)} \mathrm{J}_j^a, \tag{3.9}$$

with the evaluation parameters $s_j^{(n)}$ given by

$$s_j^{(n)} = \tfrac{1}{2} a_{(j+1,n)} - \tfrac{1}{2} a_{(1,j-1)}, \qquad\qquad a_{(j,k)} = \sum_{i=j}^k a_i, \tag{3.10}$$

for the one-loop integral we consider. We note that we can split up the generator as

$$\widehat{\mathrm{J}}^a_{(1,n),n} = \widehat{\mathrm{J}}^a_{(1,n-2),n} + \tfrac{1}{2} f^a{}_{bc} \mathrm{J}^c_{(1,n-2)} \mathrm{J}^b_{(n-1,n)} + \widehat{\mathrm{J}}^a_{(n-1,n),n}\big|_{\text{bi-local}} + s_{n-1}^{(n)} \mathrm{J}^a_{n-1} + s_n^{(n)} \mathrm{J}_n^a. \tag{3.11}$$

Here, the restriction to the bi-local part corresponds to leaving out the local contribution governed by the evaluation parameters. Next, we consider the permutation $\mathrm{P} = (n-1, n)$, along with the respective permutations of the weights $a_j$, and note that

$$\mathrm{P}^{-1} \widehat{\mathrm{J}}^a_{(1,n),n} \mathrm{P} = \widehat{\mathrm{J}}^a_{(1,n-2),n} + \tfrac{1}{2} f^a{}_{bc} \mathrm{J}^c_{(1,n-2)} \mathrm{J}^b_{(n-1,n)} - \widehat{\mathrm{J}}^a_{(n-1,n),n}\big|_{\text{bi-local}} + \tilde{s}_{n-1}^{(n)} \mathrm{J}_n^a + \tilde{s}_n^{(n)} \mathrm{J}^a_{n-1}, \tag{3.12}$$

where

$$s_{n-1}^{(n)} = \tfrac{1}{2} a_n - \tfrac{1}{2} a_{(1,n-2)}, \qquad\qquad s_n^{(n)} = -\tfrac{1}{2} a_{n-1} - \tfrac{1}{2} a_{(1,n-2)}, \tag{3.13}$$

$$\tilde{s}_{n-1}^{(n)} = \tfrac{1}{2} a_{n-1} - \tfrac{1}{2} a_{(1,n-2)}, \qquad\qquad \tilde{s}_n^{(n)} = -\tfrac{1}{2} a_n - \tfrac{1}{2} a_{(1,n-2)}. \tag{3.14}$$

Due to the permutation symmetry of the integral $I_n$, the transformed generator is a symmetry as well, and hence so is the difference of the generators

$$\widehat{\mathrm{J}}^a_{(1,n)} - \mathrm{P}^{-1} \widehat{\mathrm{J}}^a_{(1,n)} \mathrm{P} = f^a{}_{bc} \mathrm{J}^c_{n-1} \mathrm{J}^b_n - a_{n-1} \mathrm{J}_n^a + a_n \mathrm{J}^a_{n-1} = 2 \widehat{\mathrm{J}}^a_{n-1,n}. \tag{3.15}$$

The $n$-point invariance thus implies a two-point invariance which, due to the full permutation invariance of the integral, holds for any given pair of points. Conversely, it is easy to see that the evaluation parameters are designed in such a way that

$$\widehat{\mathrm{J}}^a_{(1,n)} = \sum_{k>j=1}^n \widehat{\mathrm{J}}^a_{jk}. \tag{3.16}$$

**Two-Point Level-One Invariance.**     Finally, let us prove the two-point level-one invariance of the integrand (3.7). We begin by considering the level-one momentum density at $y = 0$

$$\widehat{\mathrm{P}}_{jk}^{\mu} = \frac{i}{2} \left( \mathrm{P}_j^{\mu} \mathrm{D}_k + \mathrm{P}_{j\nu} \mathrm{L}_k^{\mu\nu} - i a_k \mathrm{P}_j^{\mu} - (j \leftrightarrow k) \right). \tag{3.17}$$

Plugging the level-zero densities (2.13) into the above expression yields

$$\widehat{\mathrm{P}}_{jk}^{\mu} = \frac{i}{2} \left( T^{\nu\mu\rho} \partial_{x_j,\rho} \partial_{x_k,\nu} + \left( 2a_j + m_j \partial_{m_j} \right) \partial_{x_k}^{\mu} - \left( 2a_k + m_k \partial_{m_k} \right) \partial_{x_j}^{\mu} \right), \tag{3.18}$$

where

$$T^{\nu\mu\rho} = x_{jk}^{\nu} \eta^{\mu\rho} + x_{jk}^{\rho} \eta^{\mu\nu} - x_{jk}^{\mu} \eta^{\nu\rho}. \tag{3.19}$$

Since the integrand of (3.7) is factorized and the level-one density (3.17) only contains derivatives with respect to points $j$ and $k$, it suffices to consider the action of generator density on a product of two propagators. Using the abbreviation $\hat{x}_{0j}^2 = x_{0j}^2 + m_j^2$ we find[4]

$$\widehat{\mathrm{P}}_{jk}^{\mu} (\hat{x}_{0j}^2)^{-a_j} (\hat{x}_{0k}^2)^{-a_k} = 2i a_j a_k (\hat{x}_{0j}^2)^{-a_j-1} (\hat{x}_{0k}^2)^{-a_k-1} \left[ T^{\nu\mu\rho} x_{0j,\rho} x_{0k,\nu} + x_{0k}^{\mu} x_{0j}^2 - x_{0j}^{\mu} x_{0k}^2 \right]. \tag{3.20}$$

Carrying out the contractions yields

$$T^{\nu\mu\rho} x_{0j,\rho} x_{0k,\nu} = x_{0j}^{\mu} x_{0k}^2 - x_{0k}^{\mu} x_{0j}^2 \tag{3.21}$$

and consequently

$$\widehat{\mathrm{P}}_{jk}^{\mu} (\hat{x}_{0j}^2)^{-a_j} (\hat{x}_{0k}^2)^{-a_k} = 0. \tag{3.22}$$

Checking the two-point invariance of the integrand (3.7) under the remaining level-one and level-one extra generators is a straightforward exercise. For $y = 0$ and a dual-conformal integrand (2.1) there is of course no need for further checks as the algebra relations already guarantee the invariance under the remaining level-one generators. However, let us stress that level-one invariance as well as level-one extra invariance of the generic scalar $n$-point Feynman integrals (3.7) hold no matter whether the conformal constraint (2.1) is satisfied or not. This observation will later on allow us to also consider non-dual-conformal integrals which do not have full level-zero symmetry.

   In summary, we have found that one-loop integrals (and in fact already the integrands) are invariant under Yangian level-one generators,

$$\widehat{\mathrm{J}}^a I_n = 0, \tag{3.23}$$

irrespective of whether or not the conformal constraint (2.1) is satisfied. This finding also holds if we allow the internal summation in the Yangian level-one generators to include the mass dimension, i.e. we have

$$\widehat{\mathrm{J}}_{\mathrm{extra}}^a I_n = 0. \tag{3.24}$$

See (A.1) and (A.2) for explicit expressions of the generators.

---

[4]With regard to potential contact terms as they arise from the second order Laplace operator we note that $\widehat{\mathrm{P}}_{jk}^{\mu}$ is a product of first order differential operators each acting on a single leg $j$ or $k$ only. In the fully massive case, the propagators are completely regular even at the contact point.

**Internal Mass.** The last finding can be puzzling, since e.g. the generator $\widehat{P}^{\mu}_{\text{extra}}$ involves the densities of the generator $L_{\mu D+1}$, which mixes the mass with the other dimensions and is hence itself not a symmetry of the Feynman integral. We can illustrate the significance of the contribution $\widehat{P}^{\mu}_{\text{extra}}$ by allowing the internal mass $m_0$ to be non-vanishing. We are thus considering the Feynman integral

$$\tilde{I}_n = \int \frac{\mathrm{d}^D x_0}{\prod_{j=1}^{n}(x_{0j}^2 + (m_j - m_0)^2)^{a_j}}\,. \tag{3.25}$$

Note that this integral arises from the one with vanishing $m_0$ by shifting the external masses,

$$\tilde{I}_n = I_n(\{m_j - m_0\}) = e^{-im_0 P^{D+1}} I_n(\{m_j\})\,. \tag{3.26}$$

We thus find that the level-one generators act on the shifted Feynman integral as

$$\widehat{J}^a I_n(\{m_j - m_0\}) = e^{-im_0 P^{D+1}}\Big(e^{im_0 P^{D+1}}\widehat{J}^a e^{-im_0 P^{D+1}}\Big) I_n(\{m_j\})\,. \tag{3.27}$$

The above conjugation is evaluated as

$$e^{im_0 P^{D+1}}\widehat{J}^a e^{-im_0 P^{D+1}} = \sum_{n=0}^{\infty} \frac{i^n m_0^n}{n!}\big[P^{D+1},\widehat{J}^a\big]_{(n)}\,, \tag{3.28}$$

where, $[A,B]_{(n)}$ denotes the $n$-fold commutator

$$[A,B]_{(n)} = \big[A,[A,B]_{(n-1)}\big]\,, \qquad\qquad [A,B]_{(0)} = B\,. \tag{3.29}$$

These commutators are easy to evaluate by employing the commutation relations (2.15) and (3.1). Specifying to the level-one momentum generator, we note that $\widehat{P}^{\mu}$ only commutes with $P^{D+1}$ if we set $y = 1$ in (3.3) and thus include the contribution $\widehat{P}^{\mu}_{\text{extra}}$ summing over the mass component as well. We conclude that for non-vanishing $m_0$ the operator $\widehat{P}$ is only a symmetry if $y = 1$.

## 3.2 The Symmetry at Two Loops

The invariance of two-loop graphs can be derived from the invariance of the constituting one-loop graphs. Concretely, we consider a two-loop graph that arises from joining two one-loop graphs with $l + 1$ and $r + 1$ legs, respectively, thus having $n = l + r$ legs in total:

$$I_n^{(2)} = \quad\vcenter{\hbox{(diagram)}}\quad = \int \frac{\mathrm{d}^D x_0 \mathrm{d}^D x_{\bar{0}}}{x_{0\bar{0}}^{2b_0}\prod_{j=1}^{l}(x_{0j}^2 + m_j^2)^{a_j}\prod_{k=l+1}^{n}(x_{\bar{0}k}^2 + m_k^2)^{a_k}}\,. \tag{3.30}$$

We have noted in the above discussion that at one-loop order the diagram need not be invariant under the underlying dual conformal symmetry in order to have level-one invariance. For the two-loop discussion, however, level-zero invariance is more critical and we will set $y = 0$ in the following, since the Lorentz generator $L_{D+1\mu}$ is not a symmetry of the Feynman diagrams we consider, and thus the extra generators $\widehat{J}_{\text{extra}}$ will not generate separate symmetries at two loops, cf. (3.24).

**Dual Conformal Case.** We would like to show that there is a set of 2-loop evaluation parameters $s_j^{(2,n)}$ such that the above generator (3.9) becomes a symmetry of this diagram. Here we denote the $\ell$-loop evaluation parameters by $s_j^{(\ell,n)}$. To this end, note that we can split up a generic level-one generator as follows:

$$\widehat{J}^a_{(1,n)} = \widehat{J}^a_{(1,l)} + \widehat{J}^a_{(l+1,n)} + \tfrac{1}{2} f^a{}_{bc} J^c_{(1,l)} J^b_{(l+1,n)} + \sum_{k=1}^{l} \left( s_k^{(2,n)} - s_k^{(1,l)} \right) J_k^a + \sum_{k=l+1}^{n} \left( s_k^{(2,n)} - s_k^{(1,r)} \right) J_k^a . \tag{3.31}$$

The terms in the last line are due to the differences of the evaluation parameters for the generators acting on the diagrams containing 1 or 2 loops, respectively. When acting on the level-zero invariant $I_n^{(2)}$, we note that

$$\tfrac{1}{2} f^a{}_{bc} J^c_{(1,l)} J^b_{(l+1,n)} I_n^{(2)} = \tfrac{1}{2} f^a{}_{bc} J^c_{(1,l)} \left( J^b - J^b_{(1,l)} \right) I_n^{(2)} = -\tfrac{\mathfrak{c}}{2} J^a_{(1,l)} I_n^{(2)} , \tag{3.32}$$

where the dual Coxeter number $\mathfrak{c}$ arises from the contraction

$$f^a{}_{bc} f^{cb}{}_d = 2\mathfrak{c} \delta^a_d . \tag{3.33}$$

Consequently, we have

$$\widehat{J}^a_{(1,n)} I_n^{(2)} = \sum_{k=1}^{l} \left( s_k^{(2,n)} - s_k^{(1,l)} - \tfrac{\mathfrak{c}}{2} \right) J_k^a I_n^{(2)} + \sum_{k=l+1}^{n} \left( s_k^{(2,n)} - s_k^{(1,r)} \right) J_k^a I_n^{(2)} . \tag{3.34}$$

Here, we have used that $I_n^{(2)}$ is invariant under the partial level-one generators acting on the first $l$ and last $r$ legs, respectively, since already the *integrands* of the constituent one-loop graphs are invariant.

We can then take the above equation as a definition of the evaluation parameters for the Yangian level-one generators at the two-loop level. Concretely, this gives

$$s_k^{(2,n)} = \tfrac{1}{2} \left( a_{(k+1,l)} - a_{(1,k-1)} + \mathfrak{c} \right), \qquad \text{for} \quad k \leq l , \tag{3.35}$$

$$s_k^{(2,n)} = \tfrac{1}{2} \left( a_{(k+1,n)} - a_{(l+1,k-1)} \right), \qquad \text{for} \quad k \geq l+1 . \tag{3.36}$$

The dual Coxeter number can be inferred by commuting the constituents of the level-one momentum generator (3.4). This gives

$$2i([P^\mu, D] + [P_\nu, L^{\mu\nu}]) = 2 \left( P^\mu + \delta^\nu_\nu P^\mu - \delta^\mu_\nu P^\nu \right) = 2D P^\mu , \tag{3.37}$$

and consequently

$$\mathfrak{c} = D . \tag{3.38}$$

**Non-Dual-Conformal Case.** At the one-loop level, we noticed that all level-one generators are symmetries even if the conformal constraints (2.16) are not satisfied. This finding partially persists at the two-loop level, where we restrict ourselves to the level-one momentum generator, again setting $y = 0$. For this generator, many aspects of the above discussion are still valid. The only difference is that $I_n^{(2)}$ is no longer annihilated by the dilatation generator D. Instead, we note that

$$D I_n^{(2)} = -i \left( D - a_{(1,l)} - b_0 + D - a_{(l+1,n)} - b_0 \right) I_n^{(2)} = -i\alpha I_n^{(2)} , \tag{3.39}$$

which only vanishes if the conformal constraints are satisfied. We can then modify (3.34) to yield

$$\widehat{P}^{\mu} I_n^{(2)} = \sum_{k=1}^{l} \left( s_k^{(2,n)} - s_k^{(1,l)} - \frac{\mathfrak{c}}{2} + \frac{\alpha}{2} \right) P_k^{\mu} I_n^{(2)} + \sum_{k=l+1}^{n} \left( s_k^{(2,n)} - s_k^{(1,r)} \right) P_k^{\mu} I_n^{(2)}, \tag{3.40}$$

from which we read off the evaluation parameters

$$s_k^{(2,n)} = \tfrac{1}{2} \big( a_{(k+1,n)} - a_{(1,k-1)} - D + a_{(1,l)} + 2b_0 \big), \qquad \text{for} \quad k \leq l, \tag{3.41}$$

$$s_k^{(2,n)} = \tfrac{1}{2} \big( a_{(k+1,n)} - a_{(l+1,k-1)} \big), \qquad \text{for} \quad k \geq l+1. \tag{3.42}$$

We note that we can always adapt the evaluation parameters by employing a shift $s_k \to s_k + C$, which acts as

$$\widehat{P}^{\mu} \to \widehat{P}^{\mu} + C P^{\mu} \tag{3.43}$$

on the level-one generator. Since $P^{\mu}$ is a symmetry generator itself, we can omit the last term when discussing the respective level-one generator. In this way, we obtain the evaluation parameters

$$s_k^{(2,n)} = \tfrac{1}{2} \big( a_{(k+1,n)} - a_{(1,k-1)} \big), \qquad \text{for} \quad k \leq l, \tag{3.44}$$

$$s_k^{(2,n)} = \tfrac{1}{2} \big( a_{(k+1,n)} - a_{(1,k-1)} + D - 2b_0 \big), \qquad \text{for} \quad k \geq l+1. \tag{3.45}$$

These evaluation parameters can be stated in terms of the following rule: Pick an arbitrary leg as leg one and set

$$s_1^{(n)} = \tfrac{1}{2} a_{(2,n)}. \tag{3.46}$$

Then move clockwise around the diagram and update the next parameter as

$$s_{k+1}^{(n)} = s_k^{(n)} - \tfrac{1}{2} (a_k + a_{k+1}), \tag{3.47}$$

if legs $k$ and $k+1$ are attached to the same vertex, or as

$$s_{k+1}^{(n)} = s_k^{(n)} - \tfrac{1}{2} (a_k + a_{k+1} - D) + b, \tag{3.48}$$

if the vertices of legs $k$ and $k+1$ are connected by an internal propagator with weight $b$. This rule was also stated in [3]. We note that it applies to all cases we discussed above and we have hence omitted the explicit reference to the loop number.

**Summary and Higher Loops.** The above findings at one- and two-loop orders are summarized in Table 2. Here the statements at higher loop orders reflect the following conjecture formulated in [3]: Feynman graphs cut along a closed contour from one of the three regular tilings of the plane with massless internal propagators have full Yangian symmetry in the dual conformal case, or only $\widehat{P}$ symmetry in the non-dual-conformal case, respectively; external propagators can be massive or massless. This conjecture is motivated by the fact that in the massless limit, these are the classes of Feynman diagrams that are known to enjoy Yangian symmetry [2]. It is further supported by numerical evidence obtained as follows. The level-one momentum generator $\widehat{P}$ was applied to the Feynman parametrization of examples of Feynman graphs in the respective categories. The resulting expression was then integrated numerically in Mathematica and it was compared whether the given uncertainty neighborhood of the result includes zero. It would certainly be desirable to find an analytic proof of this conjecture on massive higher loop integrals similar to the one in the massless case [2], or to extend the numerical tests with more advanced numerical techniques. Notably, all integrals at higher loops that are expected to be Yangian invariant are related to planar diagrams. At this point there is no evidence that the Yangian symmetry of single Feynman diagrams can be generalized beyond planar integrals.

Table 2: Symmetries at different loop orders.

| Loops | Graphs | Dual Conformal | Not Dual Conformal | Status |
|-------|--------|----------------|--------------------|--------|
| 1 | $n$-gons | all $\widehat{J}$ and $\widehat{J}_{\text{extra}}$ | all $\widehat{J}$ and $\widehat{J}_{\text{extra}}$ | proved |
| 2 | $l$-$r$-gons | all $\widehat{J}$ | $\widehat{P}$ | proved |
| > 2 | tilings | all $\widehat{J}$ | $\widehat{P}$ | conjectural |

## 3.3 Two-Point Yangian Invariants

Let us discuss some subtleties specific to two points. Consider a two-point invariant under the level-one symmetry which obeys

$$\widehat{J}^a I_2 = 0\,, \tag{3.49}$$

where at two points we have

$$\widehat{J}^a = \widehat{J}^a_{12} = \tfrac{1}{2} f^a{}_{bc} J_1^c J_2^b + s_1^{(2)} J_1^c + s_2^{(2)} J_2^c\,. \tag{3.50}$$

The one-loop integrals

$$I_2 = \int \frac{\mathrm{d}^D x_0}{(x_{01}^2 + m_1^2)^{a_1}(x_{02}^2 + m_2^2)^{a_2}} = \quad \raisebox{-0.5ex}{\(\circ\!\!-\!\!\overset{a_1}{\underset{}{\bullet}}\!\!-\!\!\overset{a_2}{\underset{}{}}\!\!-\!\!\circ\)}\,, \tag{3.51}$$

are actually invariant under the level-zero symmetry if the dual conformal constraint $a_1 + a_2 = D$ holds, i.e. they obey (2.14)[5]

$$\left(J_1^b + J_2^b\right) I_2 = 0\,. \tag{3.52}$$

This implies that we can write

$$0 = \widehat{J}^a I_2 = \left[ -\tfrac{1}{2} f^a{}_{bc} J_1^c J_1^b + s_1^{(2)} J_1^a - s_2^{(2)} J_1^a \right] I_2\,. \tag{3.53}$$

Now we use that

$$-\tfrac{1}{2} f^a{}_{bc} J_1^c J_1^b = -\tfrac{1}{4} f^a{}_{bc} \left[ J_1^c, J_1^b \right] = -\tfrac{1}{2} \mathfrak{c} J_1^a\,, \tag{3.54}$$

with the dual Coxeter number $\mathfrak{c}$ defined in (3.33) via $f^a{}_{bc} f^{cb}{}_d = 2\mathfrak{c} \delta^a_d$. Hence, we have

$$0 = \widehat{J}^a I_2 = \left( s_1^{(2)} - s_2^{(2)} - \tfrac{1}{2}\mathfrak{c} \right) J_1^a I_2\,. \tag{3.55}$$

From (3.10) we can read off the one-loop evaluation parameters $s_1^{(2)} = a_2/2$ and $s_2^{(2)} = -a_1/2$, which yields

$$0 = \widehat{J}^a I_2 = \tfrac{1}{2}(a_1 + a_2 - \mathfrak{c}) J_1^a I_2\,. \tag{3.56}$$

Hence, for the dual Coxeter number $\mathfrak{c} = D$ as given in (3.38) this equation is trivial if the dual conformal constraint $a_1 + a_2 = D$ holds and thus yields no non-trival constraints that could help to determine the integral.

Nevertheless, there are two ways to obtain non-trivial constraints on a two-point invariant from the Yangian level-one generators:

---

[5]The following statements can also be adapted to two-point integrals that are covariant under the level-zero symmetry, cf. (3.39).

- Firstly, at one loop order we can employ the extra symmetry $\widehat{J}_{\text{extra}}$, see (3.24) and the two-point examples in Section 8.

- Secondly, we can give up on (parts of) the level-zero symmetry, e.g. the special conformal symmetry $K^\mu$, which amounts to relaxing the dual conformal condition $\sum_j a_j = D$. Examples of how the resulting constraints can be used are discussed in Section 7.

### 3.4   Recursions from $P^{D+1}$

As seen above, there is an extra contribution to the level-one generators containing the $D+1$-components of the generator densities, in particular the $D+1$ momentum density

$$P_j^{D+1} = -i\frac{\partial}{\partial m_j}. \tag{3.57}$$

**Recursions from $P^{D+1}$.**   While these generator densities feature in the level-one symmetry of one-loop diagrams, they do not combine to form level-zero symmetry generators. It is well known that the resulting differential equations in the mass variables are very helpful in solving Feynman integrals [45, 46]. As an example, acting on the one-loop integral (2.2),

$$I_n = \int \frac{\mathrm{d}^D x_0}{\prod_{j=1}^n (x_{0j}^2 + m_j^2)^{a_j}}, \tag{3.58}$$

results in

$$P_j^{D+1} I_n[a_1, ..., a_n] = 2i a_j m_j I_n[a_1, ..., a_j + 1, ..., a_n], \tag{3.59}$$

i.e. the set of all diagrams transforms covariantly under the action of the $D+1$ momentum generator density. Hence, after the kinematic dependence of a diagram has been reduced to a linear combination of basis functions using the level-zero and level-one symmetries, these covariance equations can be used to constrain the propagator weight dependence of the expansion coefficients. Since there is an independent covariance equation for every mass $m_j$, these constraints are most powerful for integrals with all propagators massive, where the dependence on all propagator powers $a_k$ is fixed. Otherwise, only the dependence on the propagator powers of massive propagators are constrained.

Below, we use these identities to fix the propagator weight dependence of the non-dual-conformal all-mass two-point one-loop integral, such that the only remaining freedom is given by numerical prefactors. This information can then be transported to all other two-point cases by taking massless and conformal limits.

**From Conformal to Non-Conformal.**   Another important application of the $D+1$ momentum densities is to generate non-dual-conformal integrals from dual-conformal ones. As we argued in Section 2, Feynman diagrams are dual conformal if for any vertex the sum of all propagator weights gives the spacetime dimension $D$. Acting on the respective Feynman integral with a $D+1$ momentum density raises the propagator weight of a leg and breaks the conformal condition at the corresponding vertex. Repeated action allows to extract the integral at an arbitrary integer distance to conformality from the knowledge of the dual-conformal integral.[6]

---

[6]This procedure should not be confused with another way to acquire a non-dual-conformal integral from a dual-conformal one: sending one of the external points to infinity, the corresponding propagators drop out of the adequately normalized integral, turning a conformal vertex into a non-conformal one.

As an example, consider the case of the two-point one-loop integral with one non-vanishing mass. In section 8.1 and section 7.3 we calculate both the conformal as well as the non-conformal version of this integral respectively. They are given by

$$I_2^{a_1,a_2,D=a_1+a_2} = \pi^{D/2}\frac{\Gamma_{a_1/2-a_2/2}}{\Gamma_{a_1}}\frac{m_1^{a_2-a_1}}{(x_{12}^2+m_1^2)^{a_2}},\tag{3.60}$$

and

$$I_2^{a_1,a_2,D} = \pi^{D/2}\frac{\Gamma_{a_1+a_2-D/2}\Gamma_{D/2-a_2}}{\Gamma_{a_1}\Gamma_{D/2}}m_1^{D-2(a_1+a_2)}\ {}_2F_1\left[\begin{matrix}a_2,a_1+a_2-D/2\\D/2\end{matrix};-\frac{x_{12}^2}{m_1^2}\right],\tag{3.61}$$

respectively. Here Gauß' hypergeometric function is defined as

$${}_2F_1\left[\begin{matrix}\alpha,\beta\\\gamma\end{matrix};u\right]=\sum_{k=0}^{\infty}\frac{(\alpha)_k(\beta)_k}{(\gamma_k)}\frac{u^k}{k!},\tag{3.62}$$

with the Pochhammer symbol $(a)_k = \Gamma_{\alpha+k}/\Gamma_\alpha$. The precise relation between the two above results is given by

$$\left(-\frac{1}{2a_1m_1}\frac{\partial}{\partial m_1}\right)^n I_2^{a_1,a_2,D=a_1+a_2} = I_2^{a_1+n,a_2,D=a_1+a_2}.\tag{3.63}$$

From the non-dual-conformal result, we find

$$I_2^{a_1+n,a_2,D=a_1+a_2} = \pi^{D/2}\frac{\Gamma_{1/2(a_1-a_2)}\Gamma_{n+1/2(a_1+a_2)}}{\Gamma_{a_1+n}\Gamma_{1/2(a_1+a_2)}}m_1^{-(a_1+a_2+2n)}\ {}_2F_1\left[\begin{matrix}a_2,n+1/2(a_1+a_2)\\1/2(a_1+a_2)\end{matrix};-\frac{x_{12}^2}{m_1^2}\right].\tag{3.64}$$

To be concrete, taking $n = 1$, we have

$$I_2^{a_1+1,a_2,D=a_1+a_2} = \pi^{D/2}\frac{\Gamma_{1/2(a_1-a_2)}}{2\Gamma_{1+a_1}}m_1^{a_2-a_1-2}\frac{(a_1-a_2)x_{12}^2+(a_1+a_2)m_1^2}{\left(x_{12}^2+m_1^2\right)^{a_2+1}},\tag{3.65}$$

in agreement with the derivative of the conformal integral.

Hence, knowledge of the dual-conformal case is actually enough to derive the results of many non-dual-conformal cases, at least the ones with integer deviations from conformality. These cases are also the ones important for phenomenological applications. Still, having the results of arbitrarily many non-dual-conformal integrals does not necessarily allow to derive the continuous dependence on propagator weights and spacetime-dimension.

# 4  Yangian Bootstrap in a Nutshell

In this section we discuss the Yangian bootstrap algorithm introduced in [19] and extend it to the massive situation. After a brief explanation of how to extract the Yangian PDEs from the invariance equations, we illustrate the algorithm by means of a simple example. Similar steps can be taken for the integrals discussed in the subsequent sections, but below we will refrain from giving all details and rather highlight some key elements.

## 4.1 Extracting Yangian PDEs

Above we have argued that certain classes of (massive) Feynman integrals are invariant under the Yangian algebra. In order to evaluate the constraints from Yangian symmetry on a given integral, it is useful to translate the Yangian invariance equations into differential equations for the function $\phi_n$ that depends only on a reduced set of variables which takes into account Poincaré, scaling or full (dual) conformal symmetry. Doing this requires two steps: First, we bring the PDEs to a canonical form, e.g. for the level-one momentum generator we have

$$\widehat{\mathrm{P}}^\mu I_n = \sum_{j<k=1}^n a_{jk}^\mu \, \mathrm{PDE}_{jk} \, \phi_n. \tag{4.1}$$

Here the form of the vectors $a_{jk}^\mu$ depends on the particular choice variables. As a second step, we may exploit the spacetime symmetry (e.g. conformal symmetry) of the integral to argue that the vectors $a_{jk}^\mu$ are in fact independent,[7] which implies the following set of partial differential equations:

$$\mathrm{PDE}_{jk} \, \phi_n = 0, \qquad 1 \le j < k \le n. \tag{4.2}$$

Similarly we can obtain a separate set of one-loop PDEs, which follow from the extra symmetries, see (3.24). Determining the respective integral then boils down to solving the resulting set of PDEs and to using additional input (e.g. permutation symmetry in the external legs) to identify a unique invariant that corresponds to the integral under study. It is instructive to discuss an example.

**Example: 3 Points, $m_1 m_2 m_3$ (Dual Conformal).** As an example consider the conformal three-point integral

$$I_{3\bullet}^{m_1 m_2 m_3} = \int \frac{\mathrm{d}^D x_0}{(x_{01}^2 + m_1^2)^{a_1}(x_{02}^2 + m_2^2)^{a_2}(x_{03}^2 + m_3^2)^{a_3}} = \quad \tag{4.3}$$

with three non-vanishing masses and the kinematic variables given in (2.31),

$$u = \frac{\hat{x}_{12}^2}{-4m_1 m_2}, \qquad v = \frac{\hat{x}_{13}^2}{-4m_1 m_3}, \qquad w = \frac{\hat{x}_{23}^2}{-4m_2 m_3}. \tag{4.4}$$

This integral is fully bootstrapped in Section 8.5. We set $I_3 = V_3 \, \phi_3(u, v, w)$ and choose the prefactor $V_3$ that carries the scaling weight according to

$$V_3 = m_1^{-a_1} m_2^{-a_2} m_3^{-a_3}. \tag{4.5}$$

Now we would like to evaluate the level-one momentum invariance which takes the form

$$0 = \widehat{\mathrm{P}}^\mu I_{3\bullet}^{m_1 m_2 m_3} = \sum_{j<k=1}^3 \frac{x_{jk}^\mu}{m_j m_k} \mathrm{PDE}_{jk} \, \phi_3(u, v, w). \tag{4.6}$$

Here the symbols $\mathrm{PDE}_{jk}$ represent differential operators in the conformal varibles $u$, $v$ and $w$. In the following we will argue that the vectors

$$a_{jk}^\mu = \frac{x_{jk}^\mu}{m_j m_k} \tag{4.7}$$

---

[7]Whether this is possible depends on the considered example, in particular on the number of external points.

are independent such that we can read off three partial differential equations in the cross ratios:[8]

$$\text{PDE}_{12}\,\phi_3 = 0\,, \qquad\qquad \text{PDE}_{13}\,\phi_3 = 0\,, \qquad\qquad \text{PDE}_{23}\,\phi_3 = 0\,. \qquad (4.8)$$

In order to see the independence of the above vectors, we employ the configurations derived in Section 2,

$$
\begin{aligned}
[X_1] &= [1:0:\ldots:0:1]\,,\\
[X_2] &= \left[1+\tilde{m}_2^2:1-\tilde{m}_2^2:0:0:0:0:2\tilde{m}_2\right]\,,\\
[X_3] &= \left[1+z_1^2+\tilde{m}_3^2:1-z_1^2-\tilde{m}_3^2:2z_1:0:0:0:2\tilde{m}_3\right]\,.
\end{aligned}
\qquad (4.9)
$$

Since for the case of equal masses the number of variables is reduced, we assume that the masses are different and ordered according to

$$m_1 = 1 < m_2 < m_3\,. \qquad (4.10)$$

Now we can investigate, whether the vectors $a_{jk}^{\mu}$ given in (4.7) are independent after conformal transformations. In the three-variable case, the invariance equation (4.6) takes the form

$$\widehat{\mathrm{P}}^{\mu}I_{3\bullet}^{m_1 m_2 m_3} = a_{12}^{\mu}\,\text{PDE}_{12}\phi_3 + a_{13}^{\mu}\,\text{PDE}_{13}\phi_3 + a_{23}^{\mu}\,\text{PDE}_{23}\phi_3\,. \qquad (4.11)$$

Setting the parameter $c^{\mu}$ of the special conformal transformation to zero, yields

$$\text{PDE}_{13}\phi_3 + m_2^{-1}\text{PDE}_{23}\phi_3 = 0\,. \qquad (4.12)$$

On the other hand, taking the $c_1, c_4$-derivative and then setting $c^{\mu}$ to zero gives

$$\text{PDE}_{13}\phi_3 + m_2\text{PDE}_{23}\phi_3 = 0\,. \qquad (4.13)$$

Hence, for $m_2 > 1$, we can conclude that $\text{PDE}_{13}$ and $\text{PDE}_{23}$ independently annihilate $\phi_3$. Finally, taking the $c_4$-derivative and then setting $c^{\mu}$ to 0 yields (after using that $\text{PDE}_{13}$ and $\text{PDE}_{23}$ annihilate $\phi_3$)

$$\left(m_2 - m_2^{-1}\right)\text{PDE}_{12}\phi_3 = 0\,. \qquad (4.14)$$

Hence, also the differential operator $\text{PDE}_{12}$ annihilates $\phi_3$.

## 4.2 Algorithmic Solution of PDEs

In [19] an algorithmic solution of the Yangian PDEs for massless Feynman integrals was presented. The same steps can also be applied to massive Feynman integrals:

1. Translate the symmetry PDEs (4.2) into recurrence equations.

2. Find a fundamental solution of the recurrence equations.

3. Find all zeros of the fundamental solution yielding a solution basis.

4. Classify these zeros by their kinematic region.

5. In a given kinematic region, use further constraints to fix the linear combination of basis functions. These constraints can come from e.g. permutation symmetry or kinematic limits that simplify the form of the integral, such as sending a mass to zero or an external point to infinity.

For conciseness we will not go through all these steps for all of the examples discussed in the subsequent sections. We explicitly illustrate the above algorithm on the following pedagogical example.

---

[8]For explicit PDEs for the considered integral see Section 8.5.

### 4.3  2 Points, $m_1 0$: Gauß $_2F_1$ (Non-Dual-Conformal Example)

Consider the two-point integral

$$I_2^{m_1 0} = \int \frac{\mathrm{d}^D x_0}{(x_{01}^2 + m_1^2)^{a_1} (x_{02}^2)^{a_2}} = \quad \raisebox{-0.5em}{\includegraphics{diagram}} \;, \tag{4.15}$$

where we do not impose any (dual conformal) constraint on the parameters $a_1$, $a_2$ and $D$. We choose to write

$$I_2^{m_1 0} = (x_{12}^2)^{D/2 - a_1 - a_2} \phi(u), \qquad u = -\frac{m_1^2}{x_{12}^2}. \tag{4.16}$$

The integral is trivially annihilated by $\widehat{P}$ and $\widehat{L}$ but not by $\widehat{K}$ and $\widehat{D}$ since we do not have level-zero symmetry under K and D, cf. Section 3.3 . Hence, $\widehat{P}$ and $\widehat{P}_{\mathrm{extra}}$ do not yield any non-trivial PDEs. However, invariance under $\widehat{K}$ and $\widehat{D}$ give rise to the same equation, namely to Euler's hypergeometric differential equation

$$u(1 - u)\phi'' + [\gamma - (\alpha + \beta + 1)u]\phi' - \alpha \beta \phi = 0, \tag{4.17}$$

where we have introduced the parameters

$$\alpha = 1 + a_1 + a_2 - D, \qquad \beta = a_1 + a_2 - \tfrac{D}{2}, \qquad \gamma = 1 - \tfrac{D}{2} + a_1. \tag{4.18}$$

To convert this Yangian PDE into a recurrence equation we make the series ansatz

$$\phi(u) = \sum_{n \in x + \mathbb{Z}} g_n u^n. \tag{4.19}$$

Here, a priori the sum runs over all integer numbers and we even allow for a non-integer shift $x \in [0, 1)$ of the (here one-dimensional) summation lattice. Then the hypergeometric PDE translates into the following recurrence equation for the coefficient functions $g_n$:

$$0 = (n + \alpha)(n + \beta) g_n - (n + 1)(n + \gamma) g_{n+1}. \tag{4.20}$$

Modulo an overall constant this equation has a unique *fundamental solution*

$$g_n^{\alpha \beta \gamma} = \frac{(\alpha)_n (\beta)_n}{\Gamma_{n+1} (\gamma)_n}, \tag{4.21}$$

where we make use of the Pochhammer symbol

$$(a)_k = \frac{\Gamma_{a+k}}{\Gamma_a}. \tag{4.22}$$

While the above representation $g_n^{\alpha \beta \gamma}$ of the fundamental solution in terms of Pochhammer symbols may be better behaved in certain limits, we can alternatively express the above series via

$$f_n^{\alpha \beta \gamma} = \frac{1}{\Gamma_{n+1} \Gamma_{1-n-\alpha} \Gamma_{1-n-\beta} \Gamma_{n+\gamma}}, \tag{4.23}$$

where for integer $n$ we can write

$$f_{n+x}^{\alpha \beta \gamma} = C(\alpha, \beta, \gamma, x) g_{n+x}^{\alpha \beta \gamma}, \qquad C(\alpha, \beta, \gamma, x) = \frac{\Gamma_\alpha \Gamma_\beta \sin \pi(\alpha + x) \sin \pi(\beta + x)}{\pi^2 \Gamma_\gamma}. \tag{4.24}$$

Here the overall constant $C$ depends on the parameters $\alpha$, $\beta$, $\gamma$ and the base point $x$. Hence, for every $x \in [0, 1)$ the following series represents a formal solution of the hypergeometric PDE:[9]

$$G_x^{\alpha\beta\gamma}(u) = \sum_{n \in x+\mathbb{Z}} u^n f_n^{\alpha\beta\gamma} = u^x \sum_{n \in \mathbb{Z}} u^n f_{n+x}^{\alpha\beta\gamma} . \tag{4.25}$$

The above series $G_x^{\alpha\beta\gamma}(u)$ terminates for the following four choices of $x$, which we take as a condition for convergence:

|   | Region I | Region II |
|---|----------|-----------|
| 1 | 0 | $-\alpha$ |
| 2 | $1-\gamma$ | $-\beta$ |

(4.26)

These four choices for $x$ correspond to the zeros of the fundamental solution (4.23), and anticipating their interpretation we have classified them according to two different kinematic regions. Moreover, the fundamental solution (4.23), which is symmetric in the first two parameters, obeys the following shift identities (plus the identities with $\alpha$ and $\beta$ exchanged):

$$f_{n+1-\gamma}^{\alpha\beta\gamma} = f_{+n}^{\alpha-\gamma+1, \beta-\gamma+1, 2-\gamma} , \tag{4.27}$$

$$f_{n-\alpha}^{\alpha\beta\gamma} = f_{-n}^{\alpha, \alpha-\gamma+1, \alpha-\beta+1} . \tag{4.28}$$

Note that these identities and similar identities for the more complicated functions considered in the rest of the paper get additional prefactors when formulated in terms of the $g_n^{\alpha\beta\gamma}$ of (4.23).

Evaluating the series defined by (4.25) explicitly we find the following expressions in terms of Gauß' hypergeometric function $_2F_1$:

$$G_{I,1} \equiv G_0^{\alpha\beta\gamma}(u) = \frac{1}{\Gamma_{1-\alpha}\Gamma_{1-\beta}\Gamma_\gamma} \, _2F_1\left[\begin{matrix} \alpha, \beta \\ \gamma \end{matrix}; u\right], \tag{4.29}$$

$$G_{I,2} \equiv G_{1-\gamma}^{\alpha\beta\gamma}(u) = u^{1-\gamma} \frac{1}{\Gamma_{\gamma-\alpha}\Gamma_{\gamma-\beta}\Gamma_{2-\gamma}} \, _2F_1\left[\begin{matrix} \alpha-\gamma+1, \beta-\gamma+1 \\ 2-\gamma \end{matrix}; u\right], \tag{4.30}$$

$$G_{II,1} \equiv G_{-\alpha}^{\alpha\beta\gamma}(u) = u^{-\alpha} \frac{1}{\Gamma_{1-\alpha}\Gamma_{\alpha-\beta+1}\Gamma_{\gamma-\alpha}} \, _2F_1\left[\begin{matrix} \alpha, \alpha-\gamma+1 \\ \alpha-\beta+1 \end{matrix}; \frac{1}{u}\right], \tag{4.31}$$

$$G_{II,2} \equiv G_{-\beta}^{\alpha\beta\gamma}(u) = u^{-\beta} \frac{1}{\Gamma_{1-\beta}\Gamma_{\beta-\alpha+1}\Gamma_{\gamma-\beta}} \, _2F_1\left[\begin{matrix} \beta, \beta-\gamma+1 \\ \beta-\alpha+1 \end{matrix}; \frac{1}{u}\right]. \tag{4.32}$$

The above algorithmic procedure generalizes to the more involved examples discussed in the subsequent sections. In this simple case, however, the two functions $G_{I,1}$ and $G_{I,2}$ in Region I are also found when solving the Yangian PDE (4.17) directly in Mathematica:

$$\phi(u) = + c_1 \, _2F_1\left[\begin{matrix} \alpha, \beta \\ \gamma \end{matrix}; u\right] + c_2 \, u^{1-\gamma} \, _2F_1\left[\begin{matrix} 1+\alpha-\gamma, 1+\beta-\gamma \\ 2-\gamma \end{matrix}; u\right]. \tag{4.33}$$

Finally, we can use the $\mathrm{P}^{D+1}$ recursion from Section 3.4 to determine the $a_1$-dependence of the prefactors $c_j$. Here, (3.59) reads

$$\frac{1}{a_1}\phi'(u) = \phi(u)\big|_{a_1 \to a_1+1} . \tag{4.34}$$

In this particular case, this constraint could be solved using identities of $_2F_1$, but in anticipation of the more complicated cases in the following sections, we instead expand both sides of (4.34)

---

[9]Note that we could alternatively define the series by summing over $g_n^{\alpha\beta\gamma}$ in (4.25). This, however, leads to divergent factors in the expressions (4.31,4.32) that we would have to regulate, cf. (4.24) for basepoints $x = -\alpha$ or $x = -\beta$.

in a series and compare term by term,

$$0 = \left( \frac{\alpha\beta c_1(a_1)}{a_1 \gamma} - c_1(a_1 + 1) + \mathcal{O}(u^2) \right) + u^{-\gamma} \left( \frac{(1-\gamma)c_2(a_1)}{a_1} - c_2(a_1 + 1) + \mathcal{O}(u^2) \right). \quad (4.35)$$

Solving the resulting difference equations for $c_1$ and $c_2$, we find

$$c_1 = e_1 \frac{\Gamma_\beta \Gamma_{1-\gamma}}{\Gamma_{1-\alpha} \Gamma_{\beta-\alpha+\gamma}}, \qquad\qquad c_2 = e_2 (-1)^{1+\gamma} \frac{\Gamma_{\gamma-1}}{\Gamma_{\beta-\alpha+\gamma}}, \quad (4.36)$$

where $e_1$ and $e_2$ are constant in $a_1$ but may still depend on $a_2$ and $D$.

To fix the full dependence of the prefactors, one needs to start from the more symmetric two-mass integral given in (7.45), which we bootstrap in Section 7.5. Taking the $m_2 \to 0$ limit, the answer agrees with the $a_1$ dependence we derived here and the full result given in [47] modulo a conventional overall factor:

$$c_1 = \pi^{D/2} \frac{\Gamma_\beta \Gamma_{\gamma-\alpha} \Gamma_{1-\gamma}}{\Gamma_{1-\alpha} \Gamma_{1+\beta-\gamma} \Gamma_{\beta-\alpha+\gamma}}, \qquad\qquad c_2 = \pi^{D/2} (-1)^{1+\gamma} \frac{\Gamma_{\gamma-1}}{\Gamma_{\beta-\alpha+\gamma}}. \quad (4.37)$$

**Conformal Limit.** Due to distinguished role of integrals with dual conformal symmetry, let us now consider the dual conformal limit in which we expect to find the below result (8.2). Setting $D = a_1 + a_2 + 2\epsilon$, we have

$$\alpha = 1 - 2\epsilon, \qquad \beta = \frac{a_1 + a_2}{2} - \epsilon, \qquad \gamma = 1 + \frac{a_1 - a_2}{2} - \epsilon. \quad (4.38)$$

Accordingly, in the limit $\epsilon \to 0$ (4.33) takes the form

$$\phi(u) = +c_1 \, {}_2F_1 \left[ {}^{1, A_+ - 1}_{\quad A_-}; u \right] + c_2 \, u^{1-A_-} \, {}_2F_1 \left[ {}^{A_+, a_2}_{\quad A_+}; u \right], \quad (4.39)$$

where $A_\pm = (a_1 \pm a_2)/2 + 1$. Since $c_1$ given in (4.37) is of order $\epsilon$, we conclude

$$\lim_{D \to a_1 + a_2} I_2^{m_1 0} = \pi^{D/2} \frac{\Gamma_{a_1/2 - a_2/2}}{\Gamma_{a_1}} \frac{m_1^{a_2 - a_1}}{(x_{12}^2 + m_1^2)^{a_2}}, \quad (4.40)$$

in agreement with (8.2).

# 5 A Change of Perspective: Massive Conformal Symmetry in Momentum Space

In this section we look at Yangian symmetry in region momentum space from the momentum space point of view. We begin by deriving the momentum space representation of the dual level-one momentum generator, thereby introducing a novel massive generalization of momentum space conformal symmetry, cf. [3]. After verifying the algebraic consistency of this representation, we explore the idea to bootstrap Feynman integrals in momentum space by utilizing the newly gained insights.

**Translating Generators.** A natural question is whether the massive dual conformal Yangian symmetry can be understood as the closure of the massive dual conformal Lie algebra symmetry and an ordinary (massive) conformal symmetry, similar to the situation in the massless

case [1]. To address this question, we focus on the level-one momentum generator in dual space

$$\widehat{\mathrm{P}}^\mu = \frac{i}{2}\sum_{i=2}^{n}\sum_{j=1}^{i-1}\big((\mathrm{L}_i^{\mu\nu} + \eta^{\mu\nu}\mathrm{D}_i)\mathrm{P}_{j,\nu} - (i\leftrightarrow j)\big) + \sum_{i=1}^{n} s_i \mathrm{P}_i^\mu + y\widehat{\mathrm{P}}_{\mathrm{extra}}^\mu\,, \tag{5.1}$$

where

$$\widehat{\mathrm{P}}_{\mathrm{extra}}^\mu = \frac{i}{2}\sum_{i=2}^{n}\sum_{j=1}^{i-1}\big(\mathrm{L}_i^{\mu D+1}\mathrm{P}_{j,D+1} - (i\leftrightarrow j)\big)\,, \tag{5.2}$$

and rewrite it in terms of momentum variables. The latter are related to the dual variables in the following way:

$$p_i^\mu = x_i^\mu - x_{i+1}^\mu\,. \tag{5.3}$$

The mass variables, on the contrary, stay untouched. Note that momentum conservation implies the identification $x_{n+1}^\mu = x_1^\mu$ which will be implicitly assumed henceforth. Inverting the above relation yields

$$x_i^\mu = x_1^\mu - \sum_{j<i} p_j^\mu\,, \tag{5.4}$$

which shows that the dual variables are in fact region momenta, thus explaining the arbitrary reference point. In order to rewrite the generator (5.1) as a generator in momentum space, we furthermore need to express the derivatives in equation (5.1) in terms of derivatives in momentum space. To this end, we apply the chain rule and obtain

$$\partial_{x_i}^\mu = \partial_{p_i}^\mu - \partial_{p_{i-1}}^\mu\,. \tag{5.5}$$

Substituting the expressions (5.4) and (5.5) into equation (5.1) and simplifying the result yields

$$\begin{aligned}
\widehat{\mathrm{P}}_{y=0}^\mu &= \frac{i}{2}\left\{\sum_{i=1}^{n}\big(\bar{\mathrm{K}}_{i,\bar\Delta=0}^\mu - (\Delta_i + \Delta_{i+1} + 2s_i - 2s_{i+1})\partial_{p_i}^\mu - (m_i\partial_{m_i} + m_{i+1}\partial_{m_{i+1}})\partial_{p_i}^\mu\big)\right.\\
&\qquad\left.+2\sum_{i=1}^{n}\big((\bar{\mathrm{D}}_{i,\bar\Delta=0} + m_i\partial_{m_i} + \Delta_i)\eta^{\mu\nu} - \bar{\mathrm{L}}_i^{\mu\nu}\big)\partial_{p_n,\nu}\right\}\,,\\
\widehat{\mathrm{P}}_{\mathrm{extra}}^\mu &= -\frac{i}{2}\left\{\sum_{i=2}^{n}\sum_{j=1}^{i-1}\big((p_j^\mu + \ldots + p_{i-1}^\mu)\partial_{m_i}\partial_{m_j} + (m_i - m_{i+1})\partial_{m_j}\partial_{p_i}^\mu - (m_j - m_{j+1})\partial_{m_i}\partial_{p_j}^\mu\big)\right.\\
&\qquad\left.-\sum_{i=1}^{n} m_{i+1}(\partial_{m_i} + \partial_{m_{i+1}})\partial_{p_i}^\mu + \sum_{i=1}^{n} m_1(\partial_{m_i} + \partial_{m_{i+1}})\partial_{p_n}^\mu\right\}\,,
\end{aligned} \tag{5.6}$$

where

$$\begin{aligned}
\bar{\mathrm{P}}_i^\mu &= p_i^\mu\,, & \bar{\mathrm{K}}_i^\mu &= p_i^\mu\,\partial_{p_i}\cdot\partial_{p_i} - 2p_i\cdot\partial_{p_i}\partial_{p_i}^\mu - 2\bar\Delta_i\partial_{p_i}^\mu\,,\\
\bar{\mathrm{D}}_i &= p_i\cdot\partial_{p_i} + \bar\Delta_i\,, & \bar{\mathrm{L}}_i^{\mu\nu} &= p_i^\mu\partial_{p_i}^\nu - p_i^\nu\partial_{p_i}^\mu\,.
\end{aligned} \tag{5.7}$$

Note that momentum conservation always allows us to eliminate one momentum in favor of the remaining $n-1$ momenta. The massive $n$-gon integrals for example can be expressed as

$$I_n = \int \frac{\mathrm{d}^D k}{\prod_{i=1}^{n}((k - \sum_{j<i} p_j)^2 + m_i^2)^{a_i}}\,, \tag{5.8}$$

so that the $n$-th momentum does not appear. This choice turns out to be very convenient since it allows us to drop all terms containing a derivative with respect to $p_n$ in equation (5.6). Finally, we recall that level-one momentum invariance requires the scaling dimensions $\Delta_i$ to be equal to $a_i$ while the evaluation parameters $s_j$ have to be chosen according to (3.10). For the combination appearing in equation (5.6) this implies

$$\Delta_i + \Delta_{i+1} + 2s_i - 2s_{i+1} = 2(a_i + a_{i+1}). \tag{5.9}$$

**Algebra.** With the above results at our disposal, we can now define the massive representation of the conformal algebra in momentum space as

$$\bar{P}^\mu_{m,i} = p^\mu_i, \qquad \bar{L}^{\mu\nu}_{m,i} = p^\mu_i \partial^\nu_{p_i} - p^\nu_i \partial^\mu_{p_i}, \qquad \bar{D}_{m,i} = p_i \cdot \partial_{p_i} + \tfrac{1}{2}(m_i \partial_{m_i} + m_{i+1} \partial_{m_{i+1}}) + \bar{\Delta}_i,$$

$$\bar{K}^\mu_{m,i} = p^\mu_i \, \partial_{p_i} \cdot \partial_{p_i} - 2\, p_i \cdot \partial_{p_i} \partial^\mu_{p_i} - (m_i \partial_{m_i} + m_{i+1} \partial_{m_{i+1}})\partial^\mu_{p_i} - 2\bar{\Delta}_i \partial^\mu_{p_i}. \tag{5.10}$$

A straightforward computation of the commutation relations yields that these generators indeed satisfy the same algebra relations as the massless generators, i.e.

$$[\bar{L}^{\mu\nu}_m, \bar{L}^{\rho\sigma}_m] = \eta^{\mu\sigma}\bar{L}^{\nu\rho}_m + \eta^{\nu\rho}\bar{L}^{\mu\sigma}_m - \eta^{\mu\rho}\bar{L}^{\nu\sigma}_m - \eta^{\nu\sigma}\bar{L}^{\mu\rho}_m,$$

$$[\bar{L}^{\mu\nu}_m, \bar{P}^\rho_m] = \eta^{\nu\rho}\bar{P}^\mu_m - \eta^{\mu\rho}\bar{P}^\nu_m, \qquad [\bar{D}_m, \bar{K}^\mu_m] = -\bar{K}^\mu_m,$$

$$[\bar{P}^\mu_m, \bar{K}^\nu_m] = 2\eta^{\mu\nu}\bar{D}_m + 2\bar{L}^{\mu\nu}_m, \qquad [\bar{D}_m, \bar{P}^\mu_m] = \bar{P}^\mu_m,$$

$$[\bar{L}^{\mu\nu}_m, \bar{K}^\rho_m] = \eta^{\nu\rho}\bar{K}^\mu_m - \eta^{\mu\rho}\bar{K}^\nu_m, \tag{5.11}$$

where the action on an $n$-fold tensor product space reads

$$\bar{J}_m = \sum_{i=1}^n \bar{J}_{m,i}. \tag{5.12}$$

This confirms the statement that (5.10) is merely a different representation.

**Consistency Check: One Loop Invariance.** In order to check the consistency of equation (5.6), let us verify that the right-hand side indeed annihilates the massive $n$-gons (5.8). For simplicity, we set $y = 0$. Denoting the integrand of (5.8) as $i_n$ we find

$$\bar{K}^\mu_m i_n = \bar{K}^\mu_{k,\bar{\Delta}=0} i_n - \partial_k(D + 2a_1 + m_1 \partial_{m_1}) i_n, \tag{5.13}$$

where

$$\bar{K}^\mu_m = \sum_{i=1}^{n-1} \left( \bar{K}^\mu_{i,\bar{\Delta}=0} - 2(a_i + a_{i+1})\partial^\mu_{p_i} - (m_i \partial_{m_i} + m_{i+1}\partial_{m_{i+1}})\partial^\mu_{p_i} \right), \tag{5.14}$$

with $\bar{K}^\mu_k$ denoting the massless special conformal generator acting on the loop momentum. Due to the fact that $\bar{K}^\mu_k$ is itself a total derivative, the above expression vanishes when integrated over.

# 6 Momentum Space Conformal Bootstrap

In this section we explore the idea to bootstrap Feynman integrals in momentum space rather than in dual space. The motivation to do so is twofold. First, this idea seems natural as the ubiquitous (dual) level-one momentum generator has been identified as the special conformal generator in momentum space which is local and thus easier to handle. Second, such an analysis bridges the gap between the Yangian bootstrap approach and the study of conformal constraints in momentum space pursued e.g. in [24, 25, 29].



## 6.1   3 Points, 000: Appell $F_4$

In order to start with an example that is in fact completely fixed by momentum space conformal symmetry, we begin by considering the non-dual-conformal massless star integral with three external points

$$I_3^{000} = \int \frac{\mathrm{d}^D x_0}{x_{10}^{2a_1} x_{20}^{2a_2} x_{30}^{2a_3}} = \quad 1 \quad . \tag{6.1}$$

While its dual conformal cousin with $a_1 + a_2 + a_3 = D$ is uniquely fixed by the star-triangle relation, see e.g. [48],

$$I_{3\bullet}^{000} = \int \frac{\mathrm{d}^D x_0}{x_{10}^{2a_1} x_{20}^{2a_2} x_{30}^{2a_3}} = \frac{\Gamma_{a_1'} \Gamma_{a_2'} \Gamma_{a_3'}}{\Gamma_{a_1} \Gamma_{a_2} \Gamma_{a_3}} \frac{\pi^{D/2}}{x_{12}^{2a_3'} x_{23}^{2a_1'} x_{31}^{2a_2'}} \,, \qquad a_i' = D/2 - a_i \,, \tag{6.2}$$

no such statement holds for the non-dual-conformal version of the integral. We therefore employ the conformal momentum space symmetry from above to constrain the function. To do so, we first express the star integral in terms of momenta by using equation (5.3) and obtain

$$\int \frac{\mathrm{d}^D k}{k^{2a_1} (k - p_1)^{2a_2} (k - p_1 - p_2)^{2a_3}} \,. \tag{6.3}$$

Next, we employ the scaling equation

$$\bar{\mathrm{D}}_{\bar{\Delta}=0} I_3^{000} = (D - 2a_1 - 2a_2 - 2a_3) I_3^{000} \,, \tag{6.4}$$

to justify the following ansatz:

$$I_3^{000} = (p_3^2)^{D/2 - a_1 - a_2 - a_3} \phi(u, v) \,, \qquad \text{where} \quad u = \frac{p_1^2}{p_3^2}, \quad v = \frac{p_2^2}{p_3^2} \,. \tag{6.5}$$

Eliminating $p_3$ from the ansatz by using momentum conservation and acting on it with $\bar{\mathrm{K}}^\mu_{m=0}$ and $\bar{\Delta}_i$ as specified in (5.9),i.e. $\bar{\Delta}_i = a_i + a_{i+1}$, yields

$$\bar{\mathrm{K}}^\mu I_3^{000} = 4 (p_3^2)^{D/2 - a_1 - a_2 - a_3 - 1} \left( p_1^\mu \,\mathrm{PDE}_{p_1} + p_2^\mu \,\mathrm{PDE}_{p_2} \right) \phi \,, \tag{6.6}$$

where

$$\mathrm{PDE}_{p_1} = \left( \alpha\beta + (\alpha + \beta + 1)v\partial_v + ((\alpha + \beta + 1)u - \gamma)\partial_u + v^2\partial_v^2 + (u - 1)u\partial_u^2 + 2vu\partial_v\partial_u \right),$$
$$\mathrm{PDE}_{p_2} = \left( \alpha\beta + (\alpha + \beta + 1)u\partial_u + ((\alpha + \beta + 1)v - \gamma')\partial_v + u^2\partial_u^2 + (v - 1)v\partial_v^2 + 2vu\partial_u\partial_v \right).$$
$$\tag{6.7}$$

Here, the Greek parameters are related to the propagator powers and dimension in the following way:

$$\alpha = a_2, \quad \beta = a_1 + a_2 + a_3 - \frac{D}{2}, \quad \gamma = 1 + a_1 + a_2 - \frac{D}{2}, \quad \gamma' = 1 + a_2 + a_3 - \frac{D}{2}. \tag{6.8}$$

Since $p_1$ and $p_2$ can be freely varied, both equations have to be fulfilled independently. We recognize these partial differential equations as the system defining the Appell function $F_4$, see also [24, 25, 29] for similar discussions of the conformal momentum space constraints. This comes as no surprise since the triangle integral has been shown to evaluate to a linear combination of four $F_4$ functions more than 30 years ago [47]. Furthermore, the triangle can

be obained from the box integral by sending one of the external points to infinity [6]. The latter was recently computed in [19] by utilizing the Yangian bootstrap approach and we can use the exact same techniques to reproduce the result stated in [47]. Here, we only give a brief summary of the necessary steps. For more details the reader is referred to [19].

In order to solve the partial differential equations from above, we make a power series ansatz

$$G_{xy}^{\alpha\beta\gamma\gamma'}(u,v) = \sum_{\substack{k \in x+\mathbb{Z} \\ n \in y+\mathbb{Z}}} f_{kn}^{\alpha\beta\gamma\gamma'} u^m v^n \,. \tag{6.9}$$

Acting with the PDEs (6.6) on this ansatz yields recurrence relations for the coefficients $f_{kn}^{\alpha\beta\gamma\gamma'}$ which can straightforwardly be solved, for example, by using Mathematica

$$f_{kn}^{\alpha\beta\gamma\gamma'} = \frac{1}{\Gamma_{k+1}\Gamma_{n+1}\Gamma_{k+\gamma}\Gamma_{n+\gamma'}\Gamma_{1-k-n-\alpha}\Gamma_{1-k-n-\beta}} \,. \tag{6.10}$$

Note that this expression leads to a (formal) solution of the PDEs for any value of $x$ and $y$ in (6.9). However, for generic values the series will most likely be divergent for any value of $u$ and $v$ because the sum extends over a shifted $\mathbb{Z}$-lattice. Only if $x$ and $y$ are chosen in such a way that the series terminates at a lower or upper bound for both $k$ and $n$ the series has a chance of being convergent. A careful investigation of the zeros of fundamental solution (6.10) shows that there are 12 choices for $(x, y)$ for which the power series terminates and converges. However, only four of them converge in a region around the origin in the $u$-$v$-plane which is the region that we want to focus on here. These are

$$G_{00}^{\alpha\beta\gamma\gamma'},$$
$$G_{1-\gamma,0}^{\alpha\beta\gamma\gamma'} = u^{1-\gamma} G_{00}^{\alpha+1-\gamma,\beta+1-\gamma,2-\gamma,\gamma'},$$
$$G_{0,1-\gamma'}^{\alpha\beta\gamma\gamma'} = v^{1-\gamma'} G_{00}^{\alpha+1-\gamma',\beta+1-\gamma',\gamma,2-\gamma'},$$
$$G_{1-\gamma,1-\gamma'}^{\alpha\beta\gamma\gamma'} = u^{1-\gamma} v^{1-\gamma'} G_{00}^{\alpha+2-\gamma-\gamma',\beta+2-\gamma-\gamma',2-\gamma,2-\gamma'}, \tag{6.11}$$

where

$$G_{00}^{\alpha\beta\gamma\gamma'}(u,v) = \frac{F_4\left[\begin{matrix} \alpha,\beta \\ \gamma,\gamma' \end{matrix}; u, v\right]}{\Gamma_{1-\alpha}\Gamma_{1-\beta}\Gamma_\gamma\Gamma_{\gamma'}} \,. \tag{6.12}$$

Here the Appell hypergeometric function $F_4$ is defined as

$$F_4\left[\begin{matrix} \alpha,\beta \\ \gamma,\gamma' \end{matrix}; u, v\right] = \sum_{j,k=0}^{\infty} \frac{(\alpha)_{j+k}(\beta)_{j+k}}{(\gamma)_j(\gamma')_k} \frac{u^j}{j!} \frac{v^k}{k!} \,. \tag{6.13}$$

In the final step, we employ the permutation symmetries of the triangle integral to completely fix the solution up to an overall constant $N_4$:

$$\phi(u,v) = N_4\Big(\Gamma_\alpha\Gamma_\beta\Gamma_{1-\gamma'}\Gamma_{1-\gamma} F_4\left[\begin{matrix} \alpha,\beta \\ \gamma,\gamma' \end{matrix}; u, v\right] \tag{6.14}$$
$$+ \Gamma_{1+\alpha-\gamma}\Gamma_{1+\beta-\gamma}\Gamma_{\gamma-1}\Gamma_{1-\gamma'} u^{1-\gamma} F_4\left[\begin{matrix} \alpha+1-\gamma,\beta+1-\gamma \\ 2-\gamma,\gamma' \end{matrix}; u, v\right]$$
$$+ \Gamma_{1+\alpha-\gamma'}\Gamma_{1+\beta-\gamma'}\Gamma_{1-\gamma}\Gamma_{\gamma'-1} v^{1-\gamma'} F_4\left[\begin{matrix} \alpha+1-\gamma',\beta+1-\gamma' \\ \gamma,2-\gamma' \end{matrix}; u, v\right]$$
$$+ \Gamma_{2+\beta-\gamma-\gamma'}\Gamma_{2+\alpha-\gamma-\gamma'}\Gamma_{\gamma'-1}\Gamma_{\gamma-1} u^{1-\gamma} v^{1-\gamma'} F_4\left[\begin{matrix} \alpha+2-\gamma-\gamma',\beta+2-\gamma-\gamma' \\ 2-\gamma,2-\gamma' \end{matrix}; u, v\right]\Big).$$

The overall constant can be fixed by comparison with the star-triangle integral relation (6.2):

$$N_4 = \frac{\pi^{2+\alpha+\beta-\gamma-\gamma'}}{\Gamma_\alpha \Gamma_{1+\beta-\gamma}\Gamma_{1+\beta-\gamma'}\Gamma_{2+\alpha-\gamma-\gamma'}}\,. \tag{6.15}$$

The result (6.14) can also be shown to be in full agreement with the Feynman parametrization of the function $\phi(u,v)$ reading

$$\phi(u,v) = \frac{\pi^{D/2}}{\Gamma_{a_1}\Gamma_{a_2}\Gamma_{a_3}} \int\limits_0^\infty \frac{\mathrm{d}\alpha_2\,\mathrm{d}\alpha_3\,\alpha_2^{a_2-1}\alpha_3^{a_3-1}\Gamma_{a_1+a_2+a_3-D/2}}{(1+\alpha_2+\alpha_3)^{D-a_1-a_2-a_3}(\alpha_3+\alpha_2 u+\alpha_2\alpha_3 v)^{a_1-a_2-a_3-D/2}}\,. \tag{6.16}$$

## 6.2  3 Points, $m_1 00$

Let us now consider the same integral with one massive leg:

$$I_3^{m_1 00} = \int \frac{\mathrm{d}^D x_0}{(x_{10}^2+m_1^2)^{a_1}\,x_{20}^{2a_2}\,x_{30}^{2a_3}}$$

$$= \int \frac{\mathrm{d}^D k}{(k^2+m_1^2)^{a_1}(k-p_1)^{2a_2}(k-p_1-p_2)^{2a_3}} = \quad \begin{array}{c}\includegraphics\end{array} \quad . \tag{6.17}$$

Again, we utilize the scaling equation to justify an ansatz of the following form:

$$I_3^{m_1 00} = m_1^{D-2a_1-2a_2-2a_3}\phi\,(u,v,w)\,, \tag{6.18}$$

where

$$u = -\frac{p_1^2}{m_1^2}\,, \qquad v = -\frac{p_3^2}{m_1^2}\,, \qquad w = +\frac{p_2^2}{m_1^2}\,. \tag{6.19}$$

Here, the signs have been chosen for later convenience. Eliminating $p_3$ from the ansatz using momentum conservation and acting on the integral with $\bar{\mathrm{K}}_m^\mu$ for $\bar{\Delta}_i = a_i + a_{i+1}$ yields

$$\bar{\mathrm{K}}_m^\mu I_3^{m_1 00} = -4\,m_1^{D-2a_1-2a_2-2a_3-2}\left(p_1^\mu\,\mathrm{PDE}_{p_1}+p_2^\mu\,\mathrm{PDE}_{p_2}\right)\phi\,, \tag{6.20}$$

where

$$\mathrm{PDE}_{p_1} = (a_3\partial_u - a_2\partial_v + w\partial_u\partial_w - w\partial_v\partial_w + (v-u)\partial_u\partial_v)\,,$$
$$\mathrm{PDE}_{p_2} = \left((1+a_2+a_3-\tfrac{D}{2})\partial_w - a_2\partial_v + w\partial_w\partial_w - u\partial_u\partial_v - w\partial_v\partial_w\right)\,.$$

Since the number of independent momenta has not increased compared to the massless case, we again find two partial differential equations which need to be satisfied independently. However, as the mass introduces an additional degree of freedom, the function (6.18) now depends on three scale invariant variables instead of two. Hence, the number of PDEs is not sufficient to fully constrain the function. To make matters worse, the $\widehat{\mathrm{P}}_{\mathrm{extra}}^{\hat\mu}$ symmetry equation is trivially fulfilled and does therefore not yield any additional constraints.

To be more explicit, we make the series ansatz

$$G_{xyz}(u,v,w) = \sum_{k,l,n} f_{kln}\,\frac{u^k}{k!}\frac{v^l}{l!}\frac{w^n}{n!}\,. \tag{6.21}$$

The function $G_{xyz}(u, v, w)$ solves the above differential equations for

$$f_{kln} = c_{k+l+n} \frac{(a_2)_{k+n}(a_3)_{l+n}}{(a_2 + a_3 - \frac{D}{2} + 1)_n}, \tag{6.22}$$

with an unfixed function $c_{k+l+n}$ that depends on the sum of the three summation indices. This function is not fixed by momentum space conformal symmetry only.

A natural course of action to generate further PDEs is to consider the full set of level-one generators in dual region momentum space instead of only the level-one momentum generator, cf. Table 2. In fact, since the level-zero algebra in dual space is partially broken, it is clear that considering just the level-one momentum generator is no longer sufficient. Considering the full set of dual Yangian level-one generators can of course also be done in momentum space. However, since the level-one generators are scattered among different levels in momentum space, for example, the generator $\widehat{K}^\mu$ in $x$-space corresponds to the trilocal level-two momentum generator in $p$-space, we prefer to work in region momentum space which puts all generators on an equal footing. We will come back to the above integral $I_3^{m_1 00}$ in Section 7.6 where the function $c_{k+l+n}$ is constrained using the remaining Yangian level-one generators, cf. (7.69):

$$c_{k+l+n} = \frac{(a_1 + a_2 + a_3 - D/2)_{k+l+n}}{(D/2)_{k+l+n}}. \tag{6.23}$$

## 6.3   3 Points, 2 Loops, $m_1 00$

Consider now the two-loop integral

$$I_3^{(2)m_1 00} = \int \frac{d^D x_0 d^D x_{\bar{0}}}{(x_{10}^2 + m_1^2)^{a_1} x_{0\bar{0}}^{2b_0} x_{2\bar{0}}^{2a_2} x_{3\bar{0}}^{2a_3}}$$

$$= \int \frac{d^D q \, d^D k}{(q^2 + m_1^2)^{a_1} (k + q + p_1 + p_2)^{2b_0} (k + p_2)^{2a_2} k^{2a_3}} = \quad \tag{6.24}$$

We write this integral as

$$I_3^{(2)m_1 00} = m_1^{2D - 2b_0 - 2a_1 - 2a_2 - 2a_3} \phi(u, v, w), \tag{6.25}$$

where

$$u = -\frac{x_{12}^2}{m_1^2}, \qquad v = -\frac{x_{13}^2}{m_1^2}, \qquad w = +\frac{x_{23}^2}{m_1^2}. \tag{6.26}$$

Acting on the above ansatz with $\bar{K}_m^\mu$ and

$$\bar{\Delta}_1 = a_1 + a_2 + b_0 - D/2, \tag{6.27}$$

$$\bar{\Delta}_2 = a_2 + a_3, \tag{6.28}$$

$$\bar{\Delta}_3 = D/2 - a_2 - b_0, \tag{6.29}$$

as follows from the general rule $\bar{\Delta}_i = \Delta_i + \Delta_{i+1} + 2s_i - 2s_{i+1}$, we find exactly the same system of partial differential equations as in the one-loop case (6.20). In fact, as in the one-loop case, those equations do only depend on $a_2$, $a_3$ and $D$ and not on $a_1$. The $\bar{K}_m^\mu$ equations are solved by the fundamental solution (6.22), with the yet to be determined function $c_{k+l+n}$. For the one-loop integral this function is fixed by the $\widehat{K}$ equations. Hence, the one- and two-loop integrals (6.17) and (6.24) only differ in the function $c_{k+l+n}$.

# 7 Changing Back: Non-Dual-Conformal Integrals

While we will study dual conformal integrals in the following Section 8, here we consider integrals without imposing any constraints on the propagator powers $a_j$. In particular, this means that these integrals are not invariant under the dual conformal generator $\mathrm{K}^\mu$ at level zero. Despite the absence of dual conformal symmetry, we will see that in comparison with the momentum space conformal symmetry discussed in the previous Section 6, the Yangian level-one generators yield additional constraints for one-loop integrals as discussed in Section 3.1. In particular, we focus on the interplay between the Yangian differential equations and their solutions via hypergeometric functions. We also discuss relations between different cases. Even if integrals with more masses and parameters can in principle be reduced to simpler examples, it is instructive to explicitly discuss different cases with increasing complexity. A useful variable in a more complex example with two masses may be given by $u = x_{12}^2/m_1 m_2$ which in the limit $m_2 \to 0$ diverges, thus obscuring the reduction of the integral to a simpler case.

## 7.1 1 Point, $m_1$: Rational

As the simplest example consider the tadpole integral

$$I_1^{m_1} = \int \frac{\mathrm{d}^D x_{01}}{(x_{01}^2 - m_1^2)^{a_1}} = \quad \overset{a_1}{\multimap\!\!\bullet} \,. \tag{7.1}$$

Using a single spacetime point one cannot form a translationally and scaling invariant variable. Hence, the integral is pure weight, i.e.

$$I_1^{m_1} = c_{a_1} m_1^{D - 2a_1} \,. \tag{7.2}$$

To fix the propagator weight dependence of the constant $c_{a_1}$, we act with $\mathrm{P}^{D+1}$ as described in 3.4, implying

$$c_{a_1} = c \, \frac{\Gamma_{a_1 - D/2}}{\Gamma_{a_1}} \,. \tag{7.3}$$

Finally, we evaluate the integral numerically at a single point to fix the overall constant and find

$$I_1^{m_1} = \pi^{D/2} m_1^{D - 2a_1} \frac{\Gamma_{a_1 - D/2}}{\Gamma_{a_1}} \,. \tag{7.4}$$

## 7.2 2 Points, 00: Rational

Also for two points and two vanishing masses there is no scaling-invariant variable and the one-loop integral collapses into a trivial propagator. This is also known as the *group relation*, see e.g. [49]:

$$I_2^{00} = \int \frac{\mathrm{d}^D x_0}{x_{01}^{2a_1} x_{02}^{2a_2}} = \quad 1 \overset{}{\multimap}\!\!-\!\!-\!\!\overset{a_1}{\bullet}\!\!-\!\!-\!\!\overset{a_2}{\multimap} 2 = B \quad 1 \overset{}{\multimap}\!\!-\!\!-\!\!-\!\!-\!\!\underset{a_1 + a_2 - \frac{D}{2}}{-}\!\!-\!\!-\!\!\multimap 2 = B \frac{1}{x_{12}^{2(a_1 + a_2 - D/2)}} \,, \tag{7.5}$$

with the constant

$$B = \frac{\pi^{D/2} \Gamma_{a_1 + a_2 - D/2} \Gamma_{D/2 - a_1} \Gamma_{D/2 - a_2}}{\Gamma_{a_1} \Gamma_{a_2} \Gamma_{D - a_1 - a_2}} \,. \tag{7.6}$$

In Section 8.1 we discuss a similar situation in the dual conformal case with $a_1 + a_2 = D$, where a massless and a massive propagator are fused into a massive propagator.

### 7.3  2 Points, $m_1 0$: Gauß $_2F_1$

Note that the two-point integral with one mass was explicitly discussed in Section 4.3 for the variable $-m_1^2/x_{12}^2$ as an example to illustrate the bootstrap algorithm. When comparing to limiting cases of other integrals, it can be useful to consider different choices of variables, e.g. we can invert the variable used in Section 4.3 and write

$$I_2^{m_1 0} = m_1^{D-2a_1-2a_2} \phi(u) = \quad , \qquad u = -\frac{x_{12}^2}{m_1^2} . \tag{7.7}$$

Setting

$$\alpha = a_2 , \qquad \beta = a_1 + a_2 - \frac{D}{2} , \qquad \gamma = \frac{D}{2} , \tag{7.8}$$

the Yangian PDE obtained from invariance under the level-one special conformal generator $\widehat{\mathrm{K}}$ reads

$$u(1-u)\phi'' + [\gamma - (1+\alpha+\beta)u]\phi' - \alpha\beta\phi = 0 , \tag{7.9}$$

and is solved by

$$\phi = c_1\,{}_2F_1\left[{\alpha,\beta \atop \gamma};u\right] + c_2 u^{1-\gamma}\,{}_2F_1\left[{1+\alpha-\gamma,1+\beta-\gamma \atop 2-\gamma};u\right] . \tag{7.10}$$

This result can be compared to the below three-point result (7.75) in the coincidence limit of points 2 and 3 in Section 4.3 which yields

$$c_1 = \pi^{D/2}\frac{\Gamma_{a_1+a_2-D/2}\Gamma_{D/2-a_2}}{\Gamma_{a_1}\Gamma_{D/2}} , \qquad c_2 = 0 . \tag{7.11}$$

### 7.4  2 Points, $m_1 m_2$: Kampé de Fériet

Also for two non-vanishing masses different choices of variables lead to different types of functions in which the considered two-point integral can be expressed. For a nice solution in terms of a single Appell $F_1$ series see [50].[10]

As a first example we choose our variabels to be

$$u = -\frac{x_{12}^2}{m_1^2} , \qquad v = 1 - \frac{m_2^2}{m_1^2} , \tag{7.12}$$

such that we can write

$$I_2^{m_1 m_2} = m_1^{D-2a_1-2a_2} \phi(u,v) = \quad . \tag{7.13}$$

For convenience we set

$$\alpha = a_1 , \qquad \beta = a_2 , \qquad \gamma = a_1 + a_2 - \frac{D}{2} . \tag{7.14}$$

Making the series ansatz

$$G_{xy}(u,v) = \sum_{\substack{k\in x+\mathbb{Z} \\ n\in y+\mathbb{Z}}} g_{kn} u^k v^n = u^x v^y \sum_{\substack{k\in\mathbb{Z} \\ n\in\mathbb{Z}}} g_{k+x,n+y} u^k v^n , \tag{7.15}$$

we can solve the level-one $\widehat{\mathrm{P}}_{\mathrm{extra}}$ and $\widehat{\mathrm{K}}_{y=0}$ equations to find the fundamental solution

$$g_{kn} = \frac{(\alpha)_k(\beta)_{k+n}(\gamma)_{k+n}}{\Gamma_{k+1}\Gamma_{n+1}(\alpha+\beta)_{2k+n}} . \tag{7.16}$$

---

[10]This solution requires an inspired choice of variables including square roots.

We can alternatively express the above series via

$$f_{kn} = \frac{(-1)^k}{\Gamma_{k+1}\Gamma_{n+1}\Gamma_{1-k-\alpha}\Gamma_{1-k-n-\beta}\Gamma_{1-k-n-\gamma}\Gamma_{2k+n+\alpha+\beta}}, \tag{7.17}$$

where for a certain constant $C = C(\alpha, \beta, \gamma, x, y)$ and for integer $k$ and $n$ we have

$$f_{k+x,n+y} = C\, g_{k+x,n+y}. \tag{7.18}$$

We find 13 doublets $(x, y)$ that correspond to zeros of the fundamental solution $f_{kn}$ in $k$ and $n$, which make the above series terminate at an upper or lower bound. Only the following two of these doublets give rise to effective variables $u$ and $v$:

$$(x, y) = (0, 0), \qquad (x, y) = (0, -\alpha - \beta). \tag{7.19}$$

In terms of the two basis functions

$$G_{00} = \sum_{\substack{k \in \mathbb{Z} \\ n \in \mathbb{Z}}} g_{kn} u^k v^n, \qquad G_{0,-\alpha-\beta} = v^{-\alpha-\beta} \sum_{\substack{k \in \mathbb{Z} \\ n \in \mathbb{Z}}} g_{k,n-\alpha-\beta}\, u^k v^n, \tag{7.20}$$

we can thus make the ansatz

$$\phi = c_1\, G_{00} + c_2\, G_{0,-\alpha-\beta}. \tag{7.21}$$

Using the covariance under the action of $\mathrm{P}^{D+1}$, see Section 3.4, the prefactors are determined to be

$$c_1 = e_1 \frac{\Gamma_\gamma}{\Gamma_{\alpha+\beta}}, \qquad\qquad c_2 = e_2 \frac{\Gamma_\gamma}{\Gamma_{\alpha+\beta}}, \tag{7.22}$$

where $e_1$ and $e_2$ are fixed by two random numerical configurations to

$$e_1 = \pi^{D/2} \qquad\qquad e_2 = 0. \tag{7.23}$$

This reproduces the generalized Kampé de Fériet hypergeometric function in [42] (modulo conventions).

**Equal-Mass Limit.** Consider the limit $m_2 \to m_1$ where $v \to 0$. Hence, for $\lim_{v\to 0} v^n = \delta_{0n}$ and assuming $\alpha + \beta < 0$, we find

$$\lim_{m_2 \to m_1} \phi = c_1\, G_{00}|_{v\to 0} = c_1\, {}_3F_2\left[\begin{matrix}\alpha,\beta,\gamma\\\alpha/2+\beta/2,\alpha/2+\beta/2+1/2\end{matrix}; \frac{u}{4}\right]. \tag{7.24}$$

The coefficient $c_1$ is fixed by the below limit

$$c_1 = \pi^{D/2} \frac{\Gamma_\gamma}{\Gamma_{\alpha+\beta}}. \tag{7.25}$$

The given result agrees with the expression of [47] for the equal-mass two-point integral.

**One-Point, One-Mass Limit.** Consider now the combined limit $m_2 \to m_1, x_2 \to x_1$ where $u, v \to 0$. Here for $\alpha + \beta < 0$ we find

$$\lim_{x_2 \to x_1, m_2 \to m_1} \phi = c_1\, G_{00}|_{u,v\to 0} = c_1. \tag{7.26}$$

Comparing to the tadpole result of Section 7.1 for $a_1 \to a_1 + a_2$ which reads

$$I_1^{m_1}\big|_{a_1 \to a_1+a_2} \int \frac{\mathrm{d}^D x_0}{(x_0^2 - m_1^2)^{a_1}} = \pi^{D/2} m_1^{D-2a_1-2a_2} \frac{\Gamma_{a_1+a_2-D/2}}{\Gamma_{a_1+a_2}}, \tag{7.27}$$

we can read off (7.25).

## 7.5  2 Points, $m_1 m_2$: Appell $F_4$

Let us discuss a second choice of kinematic variables which is more symmetric than (7.12), and which will lead to a solution expressed in terms of Appell hypergeometric functions $F_4$. This result will be useful to compare to various limiting cases that were expressed in terms of Gauß' $_2F_1$. In fact, this example will show that more symmetry in the choice of variables does not necessarily lead to a simpler solution in the sense that the resulting expression will be a linear combination of hypergeometric functions rather than a single hypergeometric series as in the previous Section 7.4. We now write

$$I_2^{m_1 m_2} = \left(x_{12}^2\right)^{D/2-a_1-a_2} \phi(u,v) = \quad \text{(7.28)}$$

where we choose the symmetric variables

$$u = -\frac{m_1^2}{x_{12}^2}, \qquad v = -\frac{m_2^2}{x_{12}^2}. \qquad (7.29)$$

Moreover, we define the following abbreviations

$$\alpha = a_1 + a_2 + 1 - D, \qquad\qquad \gamma = a_1 + 1 - \tfrac{D}{2}, \qquad (7.30)$$
$$\beta = a_1 + a_2 - \tfrac{D}{2}, \qquad\qquad \gamma' = a_2 + 1 - \tfrac{D}{2}. \qquad (7.31)$$

Linear combinations of the Yangian PDEs obtained from $\widehat{P}$ and $\widehat{K}$ invariance yield the system of two differential equations defining the Appell hypergeometric function $F_4$, cf. (6.7):

$$0 = \left(\alpha\beta + (\alpha+\beta+1)u\partial_u + ((\alpha+\beta+1)v - \gamma')\partial_v + u^2\partial_u^2 + (v-1)v\partial_v^2 + 2vu\partial_u\partial_v\right)\phi(u,v),$$
$$0 = \left(\alpha\beta + (\alpha+\beta+1)v\partial_v + ((\alpha+\beta+1)u - \gamma)\partial_u + v^2\partial_v^2 + (u-1)u\partial_u^2 + 2vu\partial_v\partial_u\right)\phi(u,v).$$
$$(7.32)$$

Hence, for small $u, v$ the solution is a linear combination, cf. (6.11),

$$\phi(u,v) = c_1^{\alpha\beta\gamma\gamma'} g_1 + c_2^{\alpha\beta\gamma\gamma'} g_2 + c_3^{\alpha\beta\gamma\gamma'} g_3 + c_4^{\alpha\beta\gamma\gamma'} g_4, \qquad (7.33)$$

of the four functions

$$g_1 = F_4\left[\begin{smallmatrix}\alpha,\beta\\\gamma,\gamma'\end{smallmatrix}; u,v\right], \qquad (7.34)$$

$$g_2 = u^{1-\gamma} F_4\left[\begin{smallmatrix}\alpha+1-\gamma,\beta+1-\gamma\\2-\gamma,\gamma'\end{smallmatrix}; u,v\right], \qquad (7.35)$$

$$g_3 = v^{1-\gamma'} F_4\left[\begin{smallmatrix}\alpha+1-\gamma',\beta+1-\gamma'\\\gamma,2-\gamma'\end{smallmatrix}; u,v\right], \qquad (7.36)$$

$$g_4 = u^{1-\gamma} v^{1-\gamma'} F_4\left[\begin{smallmatrix}\alpha+2-\gamma-\gamma',\beta+2-\gamma-\gamma'\\2-\gamma,2-\gamma'\end{smallmatrix}; u,v\right]. \qquad (7.37)$$

**Permutation Symmetry.**  Let us now use further input to constrain the coefficients $c_j^{\alpha\beta\gamma\gamma'}$. We note that the considered two-point integral is invariant under the permutation $(x_1, m_1, a_1) \leftrightarrow (x_2, m_2, a_2)$, which translates into $(u, \gamma) \leftrightarrow (v, \gamma')$. Under this map we have

$$g_1 \leftrightarrow g_1, \qquad g_2 \leftrightarrow g_3, \qquad g_4 \leftrightarrow g_4, \qquad (7.38)$$

such that we conclude that permutation symmetry implies the following constraints

$$c_1^{\alpha\beta\gamma\gamma'} = c_1^{\alpha\beta\gamma'\gamma}, \qquad c_2^{\alpha\beta\gamma\gamma'} = c_3^{\alpha\beta\gamma'\gamma}, \qquad c_4^{\alpha\beta\gamma\gamma'} = c_4^{\alpha\beta\gamma'\gamma}. \qquad (7.39)$$

**Recursions from** $P^{D+1}$. As a further input we can employ the shift-covariance discussed in Section 3.4. The coefficients are functions of the space-time dimension and the propagator weights according to

$$c_i^{\alpha\beta\gamma\gamma'} = c_i(a_1, a_2, D).$$ (7.40)

Acting with $P^{D+1}$ on the general solution (7.33) and using the linear independence of the $g_i$ leads to a set of recurrence relations for the $c_i$ in the propagator weights which are solved by

$$c_1(a_1, a_2, D) = e_1 \frac{\Gamma_{D/2-a_1} \Gamma_{D/2-a_2} \Gamma_{a_1+a_2-D/2}}{\Gamma_{a_1} \Gamma_{a_2} \Gamma_{D-a_1-a_2}},$$

$$c_2(a_1, a_2, D) = e_2(-1)^{a_1} \frac{\Gamma_{a_1-D/2}}{\Gamma_{a_1}},$$

$$c_3(a_1, a_2, D) = e_3(-1)^{a_2} \frac{\Gamma_{a_2-D/2}}{\Gamma_{a_2}},$$

$$c_4(a_1, a_2, D) = e_4(-1)^{a_1+a_2} \frac{\Gamma_{a_1-D/2} \Gamma_{a_2-D/2}}{\Gamma_{a_1} \Gamma_{a_2}}.$$ (7.41)

Here the $e_i$ are constant complex numbers independent of $a_1$, $a_2$ and $D$ and the above contraints (7.39) from permutation invariance imply $e_2 = e_3$. Using random points of numerical data we finally fix

$$e_1 = \pi^{D/2}, \qquad e_2 = e_3 = (-1)^{-D/2} \pi^{D/2}, \qquad e_4 = 0.$$ (7.42)

Hence, in total we obtain the full expression for the two-mass integral

$$I_2^{m_1 m_2} = \pi^{D/2} \left(x_{12}^2\right)^{D/2-a_1-a_2} \left( \frac{\Gamma_{D/2-a_1} \Gamma_{D/2-a_2} \Gamma_{a_1+a_2-D/2}}{\Gamma_{a_1} \Gamma_{a_2} \Gamma_{D-a_1-a_2}} F_4 \left[ \begin{matrix} \alpha,\beta \\ \gamma,\gamma' \end{matrix}; u, v \right] \right.$$

$$+ (-1)^{a_1-D/2} u^{1-\gamma} \frac{\Gamma_{a_1-D/2}}{\Gamma_{a_1}} F_4 \left[ \begin{matrix} \alpha+1-\gamma,\beta+1-\gamma \\ 2-\gamma,\gamma' \end{matrix}; u, v \right]$$

$$\left. + (-1)^{a_2-D/2} v^{1-\gamma'} \frac{\Gamma_{a_2-D/2}}{\Gamma_{a_2}} F_4 \left[ \begin{matrix} \alpha+1-\gamma',\beta+1-\gamma' \\ \gamma,2-\gamma' \end{matrix}; u, v \right] \right).$$ (7.43)

Indeed, the result given in [47] agrees with the above expression obtained from bootstrap (modulo phases due to a different sign convention in the propagator).

**One-Mass Limit.** For $m_2 \to 0$ we have $v \to 0$, such that due to the reduction formula

$$F_4 \left[ \begin{matrix} \alpha,\beta \\ \gamma,\gamma' \end{matrix}; u, 0 \right] = {}_2F_1 \left[ \begin{matrix} \alpha,\beta \\ \gamma \end{matrix}; u \right],$$ (7.44)

we end up with a linear combination of two Gauß hypergeometric functions (see also Section 4.3 and Section 7.3):

$$I_2^{m_1 0} = \pi^{D/2} \left(x_{12}^2\right)^{D/2-a_1-a_2} \left( \frac{\Gamma_{D/2-a_1} \Gamma_{D/2-a_2} \Gamma_{a_1+a_2-D/2}}{\Gamma_{a_1} \Gamma_{a_2} \Gamma_{D-a_1-a_2}} {}_2F_1 \left[ \begin{matrix} \alpha,\beta \\ \gamma \end{matrix}; u \right] \right.$$

$$\left. + (-1)^{a_1-D/2} u^{D/2-a_1} \frac{\Gamma_{a_1-D/2}}{\Gamma_{a_1}} {}_2F_1 \left[ \begin{matrix} \alpha+1-\gamma,\beta+1-\gamma \\ 2-\gamma \end{matrix}; u \right] \right).$$ (7.45)

Taking in addition the conformal limit $D \to a_1 + a_2$ of this expression we have $c_1 \to 0$ and

$$I_{2\bullet}^{m_1 0} = \pi^{D/2} \frac{\Gamma_{a_1/2-a_2/2}}{\Gamma_{a_1}} \frac{m_1^{a_2-a_1}}{(m_1^2 + x_{12}^2)^{a_2}}.$$ (7.46)

This agrees with the below expression (8.2).

**Conformal Limit.** We would like to take the limit $D \to a_1 + a_2$ of the above full two-mass expression in order to arrive at the dual conformal case presented in terms of associated Legendre functions in (8.12) or in terms of hypergeometric functions in (8.31). In this limit we have

$$\alpha \to 1, \qquad\qquad \gamma \to 1 + \frac{a_1}{2} - \frac{a_2}{2} = 1 + a_-, \qquad (7.47)$$

$$\beta \to \frac{a_1 + a_2}{2} = a_+, \qquad\qquad \gamma' \to 1 - \frac{a_1}{2} + \frac{a_2}{2} = 1 - a_-, \qquad (7.48)$$

where we define $a_\pm = \frac{1}{2}(a_1 \pm a_2)$. The four basis solutions become

$$g_1 = F_4\left[\begin{smallmatrix} 1,a_+ \\ 1+a_-,1-a_- \end{smallmatrix}; u, v\right], \qquad\qquad g_2 = u^{-a_-} F_4\left[\begin{smallmatrix} 1-a_-,a_+-a_- \\ 1-a_-,1-a_- \end{smallmatrix}; u, v\right],$$

$$g_3 = v^{a_-} F_4\left[\begin{smallmatrix} 1+a_-,a_++a_- \\ 1+a_-,1+a_- \end{smallmatrix}; u, v\right], \qquad\qquad g_4 = u^{-a_-} v^{a_-} F_4\left[\begin{smallmatrix} 1,a_+ \\ 1-a_-,1+a_- \end{smallmatrix}; u, v\right], \qquad (7.49)$$

whereas the coefficients turn into

$$c_1 = 0, \qquad\qquad c_2 = (-1)^{a_+ + a_-} e_2 \frac{\Gamma_{a_-}}{\Gamma_{a_+ + a_-}},$$

$$c_4 = 0, \qquad\qquad c_3 = (-1)^{a_+ - a_-} e_3 \frac{\Gamma_{-a_-}}{\Gamma_{a_+ - a_-}}. \qquad (7.50)$$

Numerically, we find the interesting relation[11]

$$F_4\left[\begin{smallmatrix} a_+ + a_-, 1 + a_- \\ 1 + a_-, 1 + a_- \end{smallmatrix}; u, v\right] = \frac{1}{[1 + (\sqrt{-u} - \sqrt{-v})^2]^{a_+ + a_-}} \, {}_2F_1\left[\begin{smallmatrix} a_- + 1/2, a_- + a_+ \\ 2a_- + 1 \end{smallmatrix}; \frac{-4\sqrt{uv}}{1 + (\sqrt{-u} - \sqrt{-v})^2}\right], \quad (7.51)$$

which implies that the conformal two-point function becomes

$$I_{2\bullet}^{m_1 m_2} = \pi^{D/2} \frac{1}{m_1^{a_1} m_2^{a_2}} \left(\frac{\Gamma_{a_2/2 - a_1/2}}{\Gamma_{a_2}} \left(\frac{\tilde{u}}{-4}\right)^{a_1} {}_2F_1\left[\begin{smallmatrix} a_1/2 - a_2/2 + 1/2, a_1 \\ a_1 - a_2 + 1 \end{smallmatrix}; \tilde{u}\right] + (a_1 \leftrightarrow a_2)\right), \qquad (7.52)$$

with

$$\tilde{u} = \frac{-4 m_1 m_2}{x_{12}^2 + (m_1 - m_2)^2}. \qquad (7.53)$$

To convert this result into the expression (8.13) in terms of associated Legendre functions $P$ and $Q$ that we obtain from bootstrap in the subsequent Section 8, we use the identities [51]

$$_2F_1\left[\begin{smallmatrix} 1+v, 1+\mu+v \\ 2+2v \end{smallmatrix}; u\right] = 4^{v+1/2} e^{-\frac{1}{2} i\pi(\mu-1)} \sin(\pi\mu)(1-u)^{-\mu/2}(-u)^{-1-v} \frac{\Gamma(-\mu-v)}{\sqrt{\pi}\Gamma(-1/2-v)}$$
$$\times (1 + \cot(\pi\mu)\tan(\pi v))\left[\pi P_v^\mu\left(1 - \tfrac{2}{u}\right) - 2i Q_v^\mu\left(1 - \tfrac{2}{u}\right)\right], \qquad (7.54)$$

as well as

$$_2F_1\left[\begin{smallmatrix} -v, \mu-v \\ -2v \end{smallmatrix}; u\right] = 4^{-v-\frac{1}{2}}\left(-1 + \frac{1}{u}\right)^{-\mu}(-1 + u)^{\frac{\mu}{2}}(-u)^{-\mu+v} \frac{\Gamma(1-\mu+v)}{\sqrt{\pi}\Gamma(1/2+v)}(1 + \cot(\pi\mu)\tan(\pi v))$$
$$\times \left[-2\sin(\pi\mu)Q_v^\mu\left(1 - \tfrac{2}{u}\right) + \pi(\cos(\pi\mu) + (-1)^{1+\mu}\csc(\pi(\mu+v))\sin(\pi(v-\mu)))P_v^\mu\left(1 - \tfrac{2}{u}\right)\right], \qquad (7.55)$$

which are valid for $u < 0$ only. Remarkably, these identities imply that the expression (7.52) for the dual conformal two-point integral finally collapses to (see (8.13)):

$$I_{2\bullet}^{m_1 m_2} = \pi^{D/2+1/2} \frac{(1 - \tilde{v}^2)^{1/4 - a_1/4 - a_2/4}}{m_1^{a_1} m_2^{a_2}} P_{a_1/2 - a_2/2 - 1/2}^{1/2 - a_1/2 - a_2/2}(\tilde{v}), \qquad \tilde{v} = \frac{m_1^2 + m_2^2 + x_{12}^2}{2 m_1 m_2}. \qquad (7.56)$$

---

[11]Different representations of the same Feynman integrals have led to other relationships for hypergeometric functions [50].

### 7.6  3 Points, $m_1 00$: Generalized Lauricella

Next we study the one-mass triangle integral

$$I_3^{m_1 00} = \int \frac{\mathrm{d}^D x_0}{(x_{01}^2 - m_1^2)^{a_1}(x_{02}^2)^{a_2}(x_{03}^2)^{a_3}} = 1 \quad \tag{7.57}$$

We write the three-point integral in terms of a scale invariant function of three arguments:

$$I_3^{m_1 00} = m_1^{D - 2a_1 - 2a_2 - 2a_3} \phi(u, v, w), \qquad u = -\frac{x_{12}^2}{m_1^2}, \quad v = -\frac{x_{13}^2}{m_1^2}, \quad w = +\frac{x_{23}^2}{m_1^2}. \tag{7.58}$$

This integral has the full level-one symmetry but no special conformal level-zero symmetry since we do not impose the dual conformal constraint on the propagator powers. Imposing level-one momentum and special conformal symmetry on the above ansatz, we can read off the coefficients of the vectors $x_j^\mu$. Note that in the case of the special conformal level-one generator $\widehat{\mathrm{K}}$, in order to turn these coeffients into functions of $u, v, w$ modulo overall coefficients, we need to impose the $\widehat{\mathrm{P}}$ equations which makes the coefficients of $x_j^2$ with $j = 1, 2, 3$ vanish.

The resulting PDEs can be turned into recurrence equations for the coefficients $g_{kln}$ in the series ansatz

$$G_{xyz}^{\alpha_1 \alpha_2 \alpha_3 \gamma_1 \gamma_2} = \sum_{\substack{k \in x + \mathbb{Z} \\ l \in y + \mathbb{Z} \\ n \in z + \mathbb{Z}}} g_{kln} u^k v^l w^n = u^x v^y w^z \sum_{\substack{k \in \mathbb{Z} \\ l \in \mathbb{Z} \\ n \in \mathbb{Z}}} g_{k+x, l+y, n+z} u^k v^l w^n. \tag{7.59}$$

For convenience we introduce the parameters

$$\alpha_1 = a_1 + a_2 + a_3 - \frac{D}{2}, \qquad \alpha_2 = a_2, \qquad \alpha_3 = a_3, \qquad \gamma_1 = \frac{D}{2}, \tag{7.60}$$

as well as the depend parameter

$$\gamma_2 = 1 - \gamma_1 + \alpha_2 + \alpha_3 = a_2 + a_3 + 1 - D/2. \tag{7.61}$$

Modulo an unconstrained overall constant, the recurrence equations are solved by the unique fundamental solution

$$g_{kln} = \frac{(\alpha_1)_{k+l+n}(\alpha_2)_{k+n}(\alpha_3)_{l+n}}{\Gamma_{k+1}\Gamma_{l+1}\Gamma_{n+1}(\gamma_1)_{k+l+n}(\gamma_2)_n}. \tag{7.62}$$

For a fixed constant $C = C(\alpha_1, \alpha_2, \alpha_3, \gamma_1, \gamma_2, x, y, z)$ we can alternatively express the series (7.59) in terms of

$$f_{k+x, l+y, n+z} = C g_{k+x, l+y, n+z}, \tag{7.63}$$

where

$$f_{kln} = \frac{(-1)^n}{\Gamma_{k+1}\Gamma_{l+1}\Gamma_{n+1}\Gamma_{1-k-l-n-\alpha_1}\Gamma_{1-k-n-\alpha_2}\Gamma_{1-l-n-\alpha_3}\Gamma_{k+l+n+\gamma_1}\Gamma_{n+\gamma_2}}. \tag{7.64}$$

Hence, for any triplet $(x, y, z)$ the series (7.59) furnishes a formal solution of the Yangian PDEs. We find 36 zeros of the fundamental solutions, i.e. combinations of $x, y, z$ for which the series terminates. However, only for 2 of these 36 possibilities, $u, v$ and $w$ are the effective variables of the series:

$$(x, y, z) = (0, 0, 0), \qquad (x, y, z) = (0, 0, 1 - \gamma_2). \tag{7.65}$$

We note the shift identity

$$f_{kl,n+1-\gamma_2}^{\alpha_1\alpha_2\alpha_3\gamma_1\gamma_2} = (-1)^{\gamma_2-1} f_{kln}^{\alpha_1-\gamma_2+1,\alpha_2-\gamma_2+1,\alpha_3-\gamma_2+1,\gamma_1-\gamma_2+1,2-\gamma_2}, \tag{7.66}$$

which alternatively can be expressed as

$$g_{kl,n+1-\gamma_2}^{\alpha_1\alpha_2\alpha_3\gamma_1\gamma_2} = \frac{\Gamma_{\gamma_1}\Gamma_{\gamma_2}\Gamma_{1+\alpha_1-\gamma_2}\Gamma_{1+\alpha_2-\gamma_2}\Gamma_{1+\alpha_3-\gamma_2}}{\Gamma_{\alpha_1}\Gamma_{\alpha_2}\Gamma_{\alpha_3}\Gamma_{2-\gamma_2}\Gamma_{1+\gamma_1-\gamma_2}} g_{kln}^{\alpha_1-\gamma_2+1,\alpha_2-\gamma_2+1,\alpha_3-\gamma_2+1,\gamma_1-\gamma_2+1,2-\gamma_2}. \tag{7.67}$$

Hence, we may relate the second series solution specified by (7.65) to a shifted version of the first:

$$G_{001-\gamma_2}^{\alpha_1\alpha_2\alpha_3\gamma_1\gamma_2} = w^{1-\gamma_2} \frac{\Gamma_{\gamma_1}\Gamma_{\gamma_2}\Gamma_{1+\alpha_1-\gamma_2}\Gamma_{1+\alpha_2-\gamma_2}\Gamma_{1+\alpha_3-\gamma_2}}{\Gamma_{\alpha_1}\Gamma_{\alpha_2}\Gamma_{\alpha_3}\Gamma_{2-\gamma_2}\Gamma_{1+\gamma_1-\gamma_2}} G_{000}^{\alpha_1-\gamma_2+1,\alpha_2-\gamma_2+1,\alpha_3-\gamma_2+1,\gamma_1-\gamma_2+1,2-\gamma_2}. \tag{7.68}$$

Making the ansatz

$$\begin{aligned}\phi &= c_1 G_{000}^{\alpha_1\alpha_2\alpha_3\gamma_1\gamma_2} + \tilde{c}_2 G_{001-\gamma_2}^{\alpha_1\alpha_2\alpha_3\gamma_1\gamma_2}\\ &= c_1 G_{000}^{\alpha_1\alpha_2\alpha_3\gamma_1\gamma_2} + c_2 w^{1-\gamma_2} G_{000}^{\alpha_1-\gamma_2+1,\alpha_2-\gamma_2+1,\alpha_3-\gamma_2+1,\gamma_1-\gamma_2+1,2-\gamma_2},\end{aligned} \tag{7.69}$$

we can fix the coefficients $c_1$ and $c_2$ by the two limits discussed in the following:

$$c_1 = \pi^{D/2} \frac{\Gamma_{\alpha_1}\Gamma_{1-\gamma_2}}{\Gamma_{\alpha_1-\gamma_2+1}\Gamma_{\gamma_1}}, \qquad c_2 = \pi^{D/2} \frac{\Gamma_{\gamma_1-\alpha_2}\Gamma_{\gamma_1-\alpha_3}\Gamma_{\gamma_2-1}}{\Gamma_{\alpha_2}\Gamma_{\alpha_3}\Gamma_{\gamma_1-\gamma_2+1}}, \tag{7.70}$$

or alternatively

$$\tilde{c}_2 = \pi^{D/2} \frac{\Gamma_{\alpha_1}\Gamma_{\gamma_1-\alpha_2}\Gamma_{\gamma_1-\alpha_3}\Gamma_{\gamma_2-1}\Gamma_{2-\gamma_2}}{\Gamma_{\gamma_1}\Gamma_{\gamma_2}\Gamma_{1+\alpha_1-\gamma_2}\Gamma_{1+\alpha_2-\gamma_2}\Gamma_{1+\alpha_3-\gamma_2}}. \tag{7.71}$$

This result agrees with the expression given in [47].

**Two-Point One-Mass Limit.**  Consider the coincindence limit $x_3 \to x_2$ which implies

$$v \to u, \qquad w \to 0, \tag{7.72}$$

and thus with $\lim_{w\to 0} w^n = \delta_{n0}$ yields

$$\lim_{3\to 2} G_{000}^{\alpha_1\alpha_2\alpha_3\gamma_1\gamma_2}(u,v,w) = \sum_{k,l=0}^{\infty} f_{kl0} u^{k+l} = {}_2F_1\left[\begin{matrix}\alpha_1,\alpha_2+\alpha_3\\\gamma_1\end{matrix};u\right], \tag{7.73}$$

$$\lim_{3\to 2} G_{00,1-\gamma_2}^{\alpha_1\alpha_2\alpha_3\gamma_1\gamma_2}(u,v,w) \overset{\gamma_2<1}{=} 0. \tag{7.74}$$

We may thus conclude

$$\begin{aligned}\phi|_{3\to 2} &= c_1 m_1^{D-2a_1-2a_2-2a_3} {}_2F_1\left[\begin{matrix}\alpha_1,\alpha_2+\alpha_3\\\gamma_1\end{matrix};u\right]\\ &= \pi^{\frac{D}{2}} m_1^{D-2a_1-2a_2-2a_3} \frac{\Gamma_{a_1+a_2+a_3-D/2}\Gamma_{D/2-a_2-a_3}}{\Gamma_{a_1}\Gamma_{D/2}} {}_2F_1\left[\begin{matrix}a_1+a_2+a_3-D/2,a_2+a_3\\D/2\end{matrix};u\right].\end{aligned} \tag{7.75}$$

This can be compared with the two-point integral (7.10) for $a_2 + a_3 \to a_2$:

$$\phi|_{3\to 2}^{a_2+a_3\to a_2} = \pi^{\frac{D}{2}} m_1^{D-2a_1-2a_2} \frac{\Gamma_{a_1+a_2-D/2}\Gamma_{D/2-a_2}}{\Gamma_{a_1}\Gamma_{D/2}} {}_2F_1\left[\begin{matrix}a_1+a_2-D/2,a_2\\D/2\end{matrix};u\right], \tag{7.76}$$

which fixes

$$c_1 = \pi^{D/2} \frac{\Gamma_{\alpha_1}\Gamma_{1-\gamma_2}}{\Gamma_{\alpha_1-\gamma_2+1}\Gamma_{\gamma_1}}. \tag{7.77}$$

**Two-Point Zero-Mass Limit.** Note that in the limit of a vanishing propagator power $a_1 \to 0$, we have $c_1 = 0$ and the $c_2$ term yields the expected propagator type contribution with an overall factor of Gamma functions given in (7.5):

$$I_3^{m_1 00}\big|_{a_1 \to 0} = \pi^{D/2} (x_{23}^2)^{D-2a_2-2a_3} \frac{\Gamma_{D/2-a_2} \Gamma_{D/2-a_3} \Gamma_{a_2+a_3-D/2}}{\Gamma_{a_2} \Gamma_{a_3} \Gamma_{D-a_2-a_3}}. \tag{7.78}$$

This fixes the coefficient $c_2$ for $a_1 = 0$ to

$$c_2\big|_{a_1 \to 0} = \pi^{D/2} \frac{\Gamma_{\gamma_1-a_2} \Gamma_{\gamma_1-a_3} \Gamma_{\gamma_2-1}}{\Gamma_{a_2} \Gamma_{a_3} \Gamma_{\gamma_1-\gamma_2+1}}. \tag{7.79}$$

Note that using the recursions from acting with $\mathrm{P}^{D+1}$ as discussed in Section 3.4 fixes the $a_1$ dependence of the two coefficients to be

$$c_1 = f_1(a_2, a_3) \frac{\Gamma_{a_1}}{\Gamma_{a_1-\gamma_2+1}}, \qquad\qquad c_2 = f_2(a_2, a_3). \tag{7.80}$$

This shows that in fact even for $a_1 \neq 0$ we have

$$c_2 = \pi^{D/2} \frac{\Gamma_{\gamma_1-a_2} \Gamma_{\gamma_1-a_3} \Gamma_{\gamma_2-1}}{\Gamma_{a_2} \Gamma_{a_3} \Gamma_{\gamma_1-\gamma_2+1}}. \tag{7.81}$$

# 8 Dual Conformal Integrals

In this section we systematically apply the Yangian symmetry discussed above to constrain one-loop Feynman integrals with massless and massive propagators. In order to have the full Yangian symmetry, we consider the case of dual conformal integrals, i.e. the Yangian constraints on integrals for which the condition

$$D = \sum_{j=1}^{n} a_j, \tag{8.1}$$

is satisfied by the propagator powers $a_j$ entering an $n$-point vertex in the (region momentum) Feynman graph. Again we start from the simplest examples and increase the complexity step by step. For the non-dual-conformal examples we considered in the previous section we could in principle simply take the dual conformal limit to obtain the solution. However, since the dual conformal integrals are invariant under the whole Yangian and depend on less variables, the resulting constraints allow us to bootstrap more examples than above. Moreover, in the dual conformal case it is natural to employ a different set of (constrained) variables.

## 8.1 2 Points, $m_1 0$: Rational

For a single massive propagator, the two-point integral has no independent variable. Conformal symmetry fixes it to take the form

$$I_{2\bullet}^{m_1 0} = c_1 \frac{m_1^{a_2-a_1}}{(x_{12}^2 + m_1^2)^{a_2}} = \quad \raisebox{-0.5ex}{\includegraphics{placeholder}} , \tag{8.2}$$

with an undetermined constant $c_1$. This combination is also invariant under the level-one generators. To fix the normalization we can e.g. straightforwardly compute the integral in the

incident point limit (or compare with the result given in [52]). From the conformal case we can then read off the coefficient in (8.2) to be

$$c_1 = \pi^{D/2} \frac{\Gamma_{a_1/2 - a_2/2}}{\Gamma_{a_1}} . \tag{8.3}$$

We can understand this solution as a *fusion rule* for a massless and a massive propagator in the dual conformal case $a_1 + a_2 = D$:

$$I_{2\bullet}^{m_1 0} = \int \frac{d^D x_0}{(x_{01}^2 + m_1^2)^{a_1}(x_{02}^2)^{a_2}} = \; 1 \underset{m_1, a_1}{\overset{}{\bullet}} \text{---} \underset{a_2}{\circ} 2 \; = A \; 1 \underset{m_1, a_2}{\circ\text{---}\circ} 2 \; = A \frac{1}{(x_{12}^2 + m_1^2)^{a_2}} . \tag{8.4}$$

Here we have defined

$$A = \frac{\pi^{D/2} m_1^{a_2 - a_1} \Gamma_{a_1/2 - a_2/2}}{\Gamma_{a_1}} . \tag{8.5}$$

This rule is similar to the so-called *group relation* for two massless propagators in the non-conformal situation, cf. Section 7.2. These relations imply that chains of propagators connected by dual conformal vertices can be reduced according to

$$1 \circ\!\!-\!\!-\!\!\bullet\text{---}\bullet\text{---}\bullet \cdots \bullet\text{---}\circ 2 \; \simeq \; 1 \circ\!\!-\!\!-\!\!\circ 2 \; . \tag{8.6}$$

## 8.2 2 Points, $m_1 m_2$: Associated Legendre $P$

For the two-point integral with $m_1 \neq 0$, $m_2 \neq 0$ there is a single conformal variable and one can consider different choices for this variable that lead to different types of well known differential equations. In this subsection we consider a choice that results in two-parameter Legendre functions. This choice is natural since the number of parameters of the class of functions matches the number of free propagator weights of the integral. We choose the conformal variable to be (cf. [3])

$$v = \frac{x_{12}^2 + (m_1 - m_2)^2}{2m_1 m_2} + 1 = \frac{x_{12}^2 + m_1^2 + m_2^2}{2m_1 m_2} . \tag{8.7}$$

**Direct Solution of PDEs.** For compactness we set

$$\alpha = \tfrac{1}{2}(a_1 - a_2 - 1), \qquad \beta = \tfrac{1}{2}(-a_1 - a_2 + 1), \tag{8.8}$$

and make the conformal ansatz

$$I_{2\bullet}^{m_1 m_2} = m_1^{-a_1} m_2^{-a_2}(1 - v^2)^{\beta/2} \phi(v) = \; \circ\!\!-\!\!\overset{a_1}{\underset{}{\bullet}}\!\!\overset{a_2}{-}\!\!\circ . \tag{8.9}$$

Note that while it may seem unatural at first sight, pulling out the factor $(1 - v^2)^{\beta/2}$ leads to the canonical form of the below PDE. The systematics behind this prefactor is understood by writing

$$1 - v^2 = \det v_{jk} , \tag{8.10}$$

for the matrix $v$ with elements $v_{jk} = (x_{jk}^2 + m_j^2 + m_k^2)/2m_j m_k$, see also Section 8.5. Acting on this function with the level-one momentum generator $\widehat{P}_{\text{extra}}$ leads to the following associated Legendre differential equation:

$$\left(\alpha(\alpha + 1) + \frac{\beta^2}{v^2 - 1}\right)\phi - 2v\phi' + (1 - v^2)\phi'' = 0 . \tag{8.11}$$

This equation is solved by

$$\phi(v) = c_1 P_\alpha^\beta(v) + c_2 Q_\alpha^\beta(v), \tag{8.12}$$

with $P_\alpha^\beta$ and $Q_\alpha^\beta$ being the associated Legendre function of the first and second kind, respectively. The coefficients can be fixed using numerical data points to find

$$\phi(v) = 2^\beta \pi^{1-\beta} P_\alpha^\beta(v). \tag{8.13}$$

**Solution of Recurrence Relations.** As an alternative to the above solution, we can apply the steps of the formal boostrap outlined in Section 4.2. We define a function $\bar\phi$ by

$$I_{2\bullet}^{m_1 m_2} = m_1^{-a_1} m_2^{-a_2} \bar\phi(v), \tag{8.14}$$

and we make the series ansatz

$$\bar\phi(v) = \sum_k f_k v^k, \tag{8.15}$$

such that the level-one momentum constraints turn into the recurrence relation

$$(k + a_1)(k + a_2) f_k - (k+1)(k+2) f_{k+2} = 0. \tag{8.16}$$

This relation is straightforwardly solved and yields the two fundamental solutions

$$f_k^P = \frac{(-2)^k (\hat a_1)_{\hat k} (\hat a_2)_{\hat k}}{\Gamma_{k+1}}, \qquad f_k^Q = \frac{2^k (\hat a_1)_{\hat k} (\hat a_2)_{\hat k}}{\Gamma_{k+1}}, \tag{8.17}$$

where we abbreviate $\hat a = a/2$. Indeed, numerically we find the interesting identity

$$(1 - v^2)^{\beta/2} \phi(v) = \pi^{(D+2)/2} \frac{2^{1-a_1-a_2}}{\Gamma_{\hat a_1 + 1/2} \Gamma_{\hat a_2 + 1/2}} \sum_{k=0}^\infty v^k \frac{(-2)^k (\hat a_1)_{\hat k} (\hat a_2)_{\hat k}}{\Gamma_{k+1}}$$

$$= \frac{\pi^{D/2}}{2} \sum_{k=0}^\infty v^k (-2)^k \frac{\Gamma_{a_1/2 + k/2} \Gamma_{a_2/2 + k/2}}{\Gamma_{k+1} \Gamma_{a_1} \Gamma_{a_2}}. \tag{8.18}$$

**From Legendre to Gauß.** We note the relation for $v > 1$,

$$P_\alpha^\beta(v) = \frac{1}{\Gamma_{1-\beta}} (1+v)^{\beta/2} (1-v)^{-\beta/2} \, {}_2F_1 \left[ {-\alpha, \alpha+1 \atop 1-\beta} ; \frac{1-v}{2} \right], \tag{8.19}$$

which implies the alternative representation in terms the Gauß hypergeometric function ${}_2F_1$:

$$I_{2\bullet}^{m_1 m_2} = \frac{2^\beta \pi^{1-\beta}}{\Gamma_{1-\beta}} (1+v)^\beta m_1^{-a_1} m_2^{-a_2} \, {}_2F_1 \left[ {-\alpha, \alpha+1 \atop 1-\beta} ; \frac{1-v}{2} \right]. \tag{8.20}$$

This suggests to introduce the variable

$$u = \frac{x_{12}^2 + (m_1 - m_2)^2}{-4 m_1 m_2} = \frac{1-v}{2}. \tag{8.21}$$

Numerically we find for $v > 1$ that

$$\frac{2^\beta \pi^{1-\beta}}{\Gamma_{1-\beta}} (1+v)^\beta \, {}_2F_1 \left[ {-\alpha, \alpha+1 \atop 1-\beta} ; \frac{1-v}{2} \right] = \pi^{1/2 - \beta} \frac{\Gamma_{1/2-\beta}}{\Gamma_{1-2\beta}} \, {}_2F_1 \left[ {-\alpha-\beta, \alpha+1-\beta \atop 1-\beta} ; \frac{1-v}{2} \right], \tag{8.22}$$

which implies the representation

$$I_{2\bullet}^{m_1 m_2} = \frac{\pi^{1/2-\beta} \Gamma_{1/2-\beta}}{\Gamma_{1-2\beta} m_1^{a_1} m_2^{a_2}} \, {}_2F_1 \left[ {-\alpha-\beta, \alpha+1-\beta \atop 1-\beta} ; u \right] = \frac{\pi^{D/2} \Gamma_{D/2}}{\Gamma_D m_1^{a_1} m_2^{a_2}} \, {}_2F_1 \left[ {a_1, a_2 \atop (D+1)/2} ; u \right]. \tag{8.23}$$

In Section 9 we conjecture a generalization of this expression in $u$-type variables to higher point integrals. In the subsequent Section 8.3 we will bootstrap the same integral in the variable $w = 1/u$.

**Unit Propagator Powers.** In order to compare with the discussion in [35] we can take the limit $a_j \to 1$ or $\alpha, \beta \to -1/2$ in $D = 2$ which implies

$$\phi(v) = 2\pi(1-v^2)^{-1/4}\arcsin\left(\sqrt{(1-v)/2}\right). \tag{8.24}$$

Alternatively, we can solve the above PDE (8.11) directly for $a_j = 1$ which yields

$$\phi(v) = (1-v^2)^{-1/4}\left(c_1 + c_2 \log\left[\sqrt{v^2-1}+v\right]\right). \tag{8.25}$$

Comparing to (8.24) for $v > 1$, the constants are fixed to

$$c_1 = 0, \qquad c_2 = \pi i. \tag{8.26}$$

## 8.3  2 Points, $m_1 m_2$: Gauß $_2F_1$

As another alternative variable to the above we set

$$w = \frac{-4m_1 m_2}{x_{12}^2 + (m_1 - m_2)^2} = \frac{2}{1-v} = \frac{1}{u}, \tag{8.27}$$

and we define the conformal function $\phi$ via

$$I_{2\bullet}^{m_1 m_2} = m_1^{-a_1} m_2^{-a_2} u^{a_1} \phi(w). \tag{8.28}$$

Here it is convenient to introduce three shorthands

$$\alpha = \tfrac{1}{2}(a_1 - a_2 + 1), \qquad \beta = a_1, \qquad \gamma = 2\alpha. \tag{8.29}$$

Then acting on the above function with the level-one momentum generator produces the Gauß hypergeometric differential equation

$$w(1-w)\phi'' + [\gamma - (\alpha + \beta + 1)w]\phi' - \alpha\beta\phi = 0, \tag{8.30}$$

which is solved by

$$\phi(w) = c_1 \, _2F_1\left[\begin{matrix} \alpha, \beta \\ \gamma \end{matrix}; w\right] + c_2 w^{1-\gamma} \, _2F_1\left[\begin{matrix} 1-\alpha, 1-2\alpha+\beta \\ 2-\gamma \end{matrix}; w\right]. \tag{8.31}$$

The coefficients for this dual conformal integral can be fixed from a limit of the non-dual-conformal two-point integral (see (7.52)) to find

$$I_{2\bullet}^{m_1 m_2} = \pi^{D/2} m_1^{-a_1} m_2^{-a_2} \left[\frac{\Gamma_{a_2/2-a_1/2}}{\Gamma_{a_2}}\left(\frac{w}{-4}\right)^{a_1} \, _2F_1\left[\begin{matrix} \alpha, \beta \\ \gamma \end{matrix}; w\right] + (a_1 \leftrightarrow a_2)\right]. \tag{8.32}$$

**Unit Propagator Powers.** It is interesting to evaluate the limit $a_1, a_2 \to 1$ for $D = 2$, which corresponds to $\alpha \to 1/2$ and $\beta \to 1$ and yields

$$I_{2\bullet}^{m_1 m_2}\big|_{a_1=1, a_2=1} = \frac{\pi w \arccsc(\sqrt{w})}{m_1 m_2 \sqrt{w-1}}. \tag{8.33}$$

**One-mass Limit.** In the limit where $m_2 \to 0$ we have $w \to 0$ such that $_2F_1 \to 1$. We are therefore left with

$$\lim_{m_2\to 0} I_{2\bullet}^{m_1 m_2} = \lim_{m_2\to 0}\left(c_1 \frac{m_2^{a_1-a_2}}{(x_{12}^2 + (m_1-m_2)^2)^{a_1}} + c_2 \frac{m_1^{a_2-a_1}}{(x_{12}^2 + (m_1-m_2)^2)^{a_2}}\right) = c_2 \frac{m_1^{a_2-a_1}}{(x_{12}^2 + m_1^2)^{a_2}}, \tag{8.34}$$

where we assumed $a_1 > a_2$. This matches the result from Section 8.1 for

$$c_2 = \pi^{D/2} \frac{\Gamma_{a_1/2-a_2/2}}{\Gamma_{a_1}}. \tag{8.35}$$

## 8.4   3 Points, $m_1 00$: Gauß $_2F_1$

In the case of the three-point integral with two vanishing masses and evaluated at the dual conformal point $D = a_1 + a_2 + a_3$, the integral depends on a single variable which we choose as

$$u = \frac{m_1^2 x_{23}^2}{(x_{12}^2 + m_1^2)(x_{13}^2 + m_1^2)}. \tag{8.36}$$

Taking the scaling weight of the integral into account, we write

$$I_{3\bullet}^{m_1 00} = \frac{(m_1^2)^{\gamma-1}}{(x_{12}^2 + m_1^2)^\alpha (x_{13}^2 + m_1^2)^\beta} \, \phi(u) = \quad \text{,} \tag{8.37}$$



where we abbreviate

$$\alpha = a_2, \qquad \beta = a_3, \qquad \gamma = 1 + \tfrac{1}{2}(-a_1 + a_2 + a_3). \tag{8.38}$$

Then level-one momentum invariance of the integral directly implies Gauß' hypergeometric differential equation

$$u(1-u)\phi'' + [\gamma - (\alpha + \beta + 1)u]\phi' - \alpha\beta\phi = 0, \tag{8.39}$$

which is solved by

$$\phi(u) = c_1 \, {}_2F_1\left[\begin{smallmatrix} \alpha,\beta \\ \gamma \end{smallmatrix}; u\right] + c_2 u^{1-\gamma} \, {}_2F_1\left[\begin{smallmatrix} 1+\alpha-\gamma,1+\beta-\gamma \\ 2-\gamma \end{smallmatrix}; u\right]. \tag{8.40}$$

The coefficients are fixed by the below limits to read

$$c_1 = \pi^{D/2} \frac{\Gamma_{a_1/2 - a_2/2 - a_3/2}}{\Gamma_{a_1}}, \qquad c_2 = \pi^{D/2} \frac{\Gamma_{D/2-a_1} \Gamma_{D/2-a_2} \Gamma_{D/2-a_3}}{\Gamma_{a_1} \Gamma_{a_2} \Gamma_{a_3}}. \tag{8.41}$$

**Coefficients from Star-Triangle Relation.**   To determine the coefficients $c_1$ and $c_2$, we take the limit $m_1 \to 0$ which implies $u \to 0$ and thus $_2F_1 \to 1$. Hence, for $\gamma > 1$ we have

$$\lim_{m_1 \to 0} I_{3\bullet}^{m_1 00} = c_2 \frac{1}{x_{12}^{2(1+\alpha-\gamma)} x_{13}^{2(1+\beta-\gamma)} x_{23}^{2(\gamma-1)}} = c_2 \frac{1}{x_{12}^{D-2a_3} x_{23}^{D-2a_1} x_{13}^{D-2a_2}}. \tag{8.42}$$

This should be compared with the star-triangle relation for the massless three-point integral given in (6.2):

$$I_{3\bullet}^{000} = \int \frac{d^D x_0}{x_{10}^{2a_1} x_{20}^{2a_2} x_{30}^{2a_3}} = \frac{\Gamma_{a_1'} \Gamma_{a_2'} \Gamma_{a_3'}}{\Gamma_{a_1} \Gamma_{a_2} \Gamma_{a_3}} \frac{\pi^{D/2}}{x_{12}^{2a_3'} x_{23}^{2a_1'} x_{31}^{2a_2'}}, \qquad a_i' = D/2 - a_i. \tag{8.43}$$

We conclude that

$$c_2 = \pi^{D/2} \frac{\Gamma_{D/2-a_1} \Gamma_{D/2-a_2} \Gamma_{D/2-a_3}}{\Gamma_{a_1} \Gamma_{a_2} \Gamma_{a_3}}. \tag{8.44}$$

**Two-point limit** Another way to relate this integral to a simpler object is the two-point limit in which $x_3 \to x_2$. In this case, the integral should be given by the conformal one-mass two-point integral from Section 8.1, i.e. we expect

$$\lim_{x_3 \to x_2} I_{3\bullet}^{m_1 00} \overset{!}{=} \pi^{D/2} \frac{\Gamma_{a_1/2-a_2/2-a_3/2}}{\Gamma_{a_1}} \frac{m_1^{a_2+a_3-a_1}}{(x_{12}^2 + m_1^2)^{a_2+a_3}} . \tag{8.45}$$

On the other hand, performing this limit using the general solution (8.40), we find for $\gamma < 1$ that

$$\lim_{x_3 \to x_2} I_{3\bullet}^{m_1 00} = c_1 \frac{m_1^{a_2+a_3-a_1}}{(x_{12}^2 + m_1^2)^{a_2+a_3}} . \tag{8.46}$$

This allows us to immediately fix

$$c_1 = \pi^{D/2} \frac{\Gamma_{a_1/2-a_2/2-a_3/2}}{\Gamma_{a_1}} . \tag{8.47}$$

## 8.5   3 Points, $m_1 m_2 m_3$: Srivastava $H_C$, Region A

We introduce the conformal variables

$$u = \frac{x_{12}^2 + (m_1 - m_2)^2}{-4m_1 m_2} , \qquad v = \frac{x_{13}^2 + (m_1 - m_3)^2}{-4m_1 m_3} , \qquad w = \frac{x_{23}^2 + (m_2 - m_3)^2}{-4m_2 m_3} , \tag{8.48}$$

and write the integral as

$$I_{3\bullet}^{m_1 m_2 m_3} = \frac{\phi(u, v, w)}{m_1^{a_1} m_2^{a_2} m_3^{a_3}} = \quad \text{(diagram)} . \tag{8.49}$$

In Section 4.1 this integral was discussed as an example for how to extract the explicit PDEs in the conformal variables. The $\widehat{P}$-equations split into two contributions coming from $\widehat{P}_{y=0}$ and $\widehat{P}_{\text{extra}}$ that annihilate the integral separately. We will show that we can find the fundamental solution for the integral by working only with the constraints arising from $\widehat{P}_{y=0}$. Reading off the coefficients of $x_{jk}^\mu/m_j m_k$ with $j, k = 1, 2, 3$ yields the following three differential operators that annihilate $\phi(u, v, w)$:

$$\text{PDE}_{12}^{\widehat{P}_{y=0}} = 2a_3 \partial_u - \partial_v \partial_w + (2w - 1)\partial_u \partial_w + (2v - 1)\partial_u \partial_v , \tag{8.50}$$

$$\text{PDE}_{13}^{\widehat{P}_{y=0}} = 2a_2 \partial_v - \partial_u \partial_w + (2w - 1)\partial_v \partial_w + (2u - 1)\partial_u \partial_v , \tag{8.51}$$

$$\text{PDE}_{23}^{\widehat{P}_{y=0}} = 2a_1 \partial_w - \partial_u \partial_v + (2v - 1)\partial_v \partial_w + (2u - 1)\partial_u \partial_w . \tag{8.52}$$

We make the series ansatz

$$\phi(u, v, w) = \sum_{k,l,n} g_{kln} u^k v^l w^n , \tag{8.53}$$

such that (8.50,8.51,8.52) translate into three recurrence equations for the coefficients $g_{kln}$, e.g. from (8.50) we obtain

$$\begin{aligned} 0 = &2a_3(k+1)g_{k+1,l,n} + 2(k+1)l g_{k+1,ln} + 2(k+1)n g_{k+1,ln} \\ &- (l+1)(n+1)g_{k,l+1,n+1} - (k+1)(n+1)g_{k+1,l,n+1} - (k+1)(l+1)g_{k+1,l+1,n} . \end{aligned} \tag{8.54}$$

These reccurence equations can be solved in Mathematica, which yields the following fundamental solution that is unique up to overall constants:

$$g_{kln} = \frac{(a_1)_{k+l}(a_2)_{k+n}(a_3)_{l+n}}{\Gamma_{k+1}\Gamma_{l+1}\Gamma_{n+1}(\gamma)_{k+l+n}}, \qquad \gamma = \frac{D+1}{2}. \tag{8.55}$$

Remember that here we have $D = a_1 + a_2 + a_3$. For a fixed constant $C$ and for integer $k,l,n$ the fundamental solution $g_{kln}$ can be written as

$$f_{k+x,l+y,n+z} = C(a_1,a_2,a_3,x,y,z)\, g_{k+x,l+y,n+z}, \tag{8.56}$$

where

$$f_{kln} = \frac{1}{\Gamma_{1+k}\Gamma_{1+l}\Gamma_{1+n}\Gamma_{k+l+n+D/2+1/2}\Gamma_{1-k-l-a_1}\Gamma_{1-k-n-a_2}\Gamma_{1-l-n-a_3}}. \tag{8.57}$$

The transpositions of the three external legs of the integral translate into $(a_1 \leftrightarrow a_2, l \leftrightarrow n)$, $(a_2 \leftrightarrow a_3, k \leftrightarrow l)$, and $(a_1 \leftrightarrow a_3, k \leftrightarrow n)$, which are manifest symmetries of the above fundamental solution. The fundamental solution $f_{kln}$ has 29 zeros which corresponds to 29 possible choices for the set $(x,y,z)$ such that the following series terminates at an upper or lower bound

$$G_{xyz}(u,v,w) = \sum_{\substack{k \in x+\mathbb{Z} \\ l \in y+\mathbb{Z} \\ n \in z+\mathbb{Z}}} g_{kln} u^k v^l w^n. \tag{8.58}$$

Numerical analysis shows that only the choice $(x,y,z) = (0,0,0)$ of the 29 possible zeros $(x,y,z)$ of the fundamental solution leads to a series $G_{xyz}$ with effective variables $u,v,w$. This leads us to an ansatz

$$\phi = c_1\, G_{000}(u,v,w), \tag{8.59}$$

in terms of Srivastava's triple hypergeometric function $H_C$, cf. e.g. [53,54]:

$$G_{000}(u,v,w) = H_C\left[{a_1,a_2,a_3 \atop \gamma}; u,v,w\right] = \sum_{k,l,n=0}^{\infty} \frac{(a_1)_{k+l}(a_2)_{k+n}(a_3)_{l+n}}{(\gamma)_{k+l+n}} \frac{u^k}{k!}\frac{v^l}{l!}\frac{w^n}{n!}. \tag{8.60}$$

This series is known to converge for

$$|u| + |v| + |w| - 2\sqrt{(1-|u|)(1-|v|)(1-|w|)} < 2. \tag{8.61}$$

Comparing the above ansatz to numerical data from the Feynman parametrization of the integral $I_{3\bullet}^{m_1 m_2 m_3}$ we can fix the overall coefficient to find (for $D = a_1 + a_2 + a_3$):

$$I_{3\bullet}^{m_1 m_2 m_3} = \frac{\pi^{D/2}\Gamma_{D/2}}{\Gamma_D m_1^{a_1} m_2^{a_2} m_3^{a_3}} H_C\left[{a_1,a_2,a_3 \atop \frac{D+1}{2}}; u,v,w\right]. \tag{8.62}$$

In Section 9.3 we compare this series to a result for unit propagator powers in $D = 3$ and in Section 9.1 we conjecture an $n$-point generalization of this representation.

## 8.6 3 Points, $m_1 m_2 m_3$: Region B

Consider now the alternative conformal variables (these generalize the single variable for the two-point Legendre solution (8.7))

$$u = \frac{x_{12}^2 + m_1^2 + m_2^2}{2m_1 m_2}, \qquad v = \frac{x_{13}^2 + m_1^2 + m_3^2}{2m_1 m_3}, \qquad w = \frac{x_{23}^2 + m_2^2 + m_3^2}{2m_2 m_3}. \tag{8.63}$$

We refer to the kinematic region covered by a series representation in these variabels as region B. We note that for Euclidean $x_{jk}^2$ and $m_j$ these variables are never small (as opposed to (8.48)), while at the same time we expect the corresponding series solution to converge only for small $u, v, w$.

We define the function $\phi$ according to

$$I_{3\bullet}^{m_1 m_2 m_3} = \frac{\phi(u,v,w)}{m_1^{a_1} m_2^{a_2} m_3^{a_3}} = \quad \text{} \tag{8.64}$$

For the series ansatz

$$G_{xyz}(u,v,w) = \sum_{\substack{k \in x+\mathbb{Z} \\ l \in y+\mathbb{Z} \\ n \in z+\mathbb{Z}}} g_{kln} u^k v^l w^n, \tag{8.65}$$

the recurrence equations arising from $\widehat{P}_{(y=0)}$ read

$$0 = (l+1)(n+1)g_{k,l+1,n+1} + (k+1)(l+n+a_3)g_{k+1,l,n}, \tag{8.66}$$

$$0 = (k+1)(n+1)g_{k+1,l,n+1} + (l+1)(k+n+a_2)g_{k,l+1,n}, \tag{8.67}$$

$$0 = (k+1)(l+1)g_{k+1,l+1,n} + (n+1)(k+l+a_1)g_{kl,n+1}. \tag{8.68}$$

On the support of these equations, the recurrences following from the invariance under $\widehat{P}_{\text{extra}}$ take the form

$$0 = (k+l+a_1)(k+n+a_2)g_{kln} - (k+1)(k+2)g_{k+2,l,n}, \tag{8.69}$$

$$0 = (k+l+a_1)(l+n+a_3)g_{kln} - (l+1)(l+2)g_{k,l+2,n}, \tag{8.70}$$

$$0 = (k+n+a_2)(l+b+a_3)g_{kln} - (n+1)(n+2)g_{kl,n+2}. \tag{8.71}$$

While we have not determined the general solution to these equations, we note that they are solved by

$$g_{kln} = \frac{(-2)^{k+l+n} (\hat{a}_1)_{\hat{k}+\hat{l}} (\hat{a}_2)_{\hat{k}+\hat{n}} (\hat{a}_3)_{\hat{l}+\hat{n}}}{\Gamma_{k+1} \Gamma_{l+1} \Gamma_{n+1}}, \tag{8.72}$$

with the shorthand $\hat{k} = k/2$. To motivate this expression we note that in (8.18) we have found that the two-point two-mass integral can be expressed as

$$I_{2\bullet}^{m_1 m_2} = \frac{\pi^{(D+2)/2}}{m_1^{a_1} m_2^{a_2}} \frac{2^{1-D}}{\Gamma_{\hat{a}_1+1/2} \Gamma_{\hat{a}_2+1/2}} \sum_{k=0}^{\infty} v^k \frac{(-2)^k (\hat{a}_1)_{\hat{k}} (\hat{a}_2)_{\hat{k}}}{\Gamma_{k+1}}, \tag{8.73}$$

which shows that (8.72) is the natural three-point generalization of the two-point summand in (8.73).

Based on (8.72) we can now investigate the possible basis series for our ansatz. For a constant $C = C(a_1, a_2, a_3, x, y, z)$ we can write

$$f_{k+x,l+y,n+z} = C g_{k+x,l+y,n+z}, \tag{8.74}$$

where

$$f_{kln} = \frac{2^{k+l+n}}{\Gamma_{1-\hat{a}_1-\hat{k}-\hat{l}} \Gamma_{1-\hat{a}_2-\hat{k}-\hat{n}} \Gamma_{1-\hat{a}_3-\hat{l}-\hat{n}} \Gamma_{k+1} \Gamma_{l+1} \Gamma_{n+1}}. \tag{8.75}$$

This function has 17 zeros $(x, y, z)$ in $k, l, n$ but only the triplet $(x, y, z) = (0, 0, 0)$ leads to a series in the effective variables $u, v$ and $w$. We thus conclude that for small $u, v, w$ the correct Yangian invariant is proportional to this series:

$$\phi(u,v,w) = c_1 G_{000}. \tag{8.76}$$

The coefficient can be fixed using numerical data points such that we find

$$I_{3\bullet}^{m_1 m_2 m_3} = \frac{\pi^{(D+3)/2}}{m_1^{a_1} m_2^{a_2} m_3^{a_3}} \frac{2^{1-D}}{\Gamma_{\hat{a}_1+1/2} \Gamma_{\hat{a}_2+1/2} \Gamma_{\hat{a}_3+1/2}} \sum_{k,l,n=0}^{\infty} (-2)^{k+l+n} (\hat{a}_1)_{\hat{k}+\hat{l}} (\hat{a}_2)_{\hat{k}+\hat{n}} (\hat{a}_3)_{\hat{l}+\hat{n}} \frac{u^k}{k!} \frac{v^l}{l!} \frac{w^n}{n!} \,.$$

(8.77)

In the following section we will conjecture an $n$-point generalization of this expression.

# 9 All-Mass Yangian Invariant $n$-Gon Integrals

Based on the evidence from the previous Section 8, we propose the below conjectural series representations for the dual conformal, i.e. Yangian invariant $n$-point one-loop integrals with all propagators massive:

$$I_{n\bullet}^{m_1 \dots m_n} = \int \frac{d^D x_0}{\prod_{j=1}^{n} (x_{0j}^2 + m_j^2)^{a_j}} = \quad , \qquad \sum_{j=1}^{n} a_j = D \,.$$

(9.1)

Here the masses $m_j$ take generic non-zero values.

## 9.1 $n$ Points, $m_1 \dots m_n$: Conjecture in Region A

We first choose the kinematic variables $u_\alpha$ according to

$$u_{ij} = \frac{x_{ij}^2 + (m_i - m_j)^2}{-4 m_i m_j} \,.$$

(9.2)

Note that these variables can become small for real Euclidean $x_{ij}$ and $m_j$, e.g. for large masses. This is relevant since we believe that the below series merely converges for small $u_{ij}$. We refer to this series representation as the series in region A as opposed to the B-series presented in the following subsection.

When expressed in the above variables we conjecture the dual conformal $n$-point all-mass integral to be given by the expression

$$I_{n\bullet}^{m_1 \dots m_n} = \frac{\pi^{D/2} \Gamma_{D/2}}{\Gamma_D \prod_{j=1}^{n} m_j^{a_j}} \sum_{k_{12},k_{13},\dots,k_{n-1,n}=0}^{\infty} \frac{\prod_{j=1}^{n} (a_j)_{\sum_{\alpha \in B_{n|j}} k_\alpha}}{(\frac{D+1}{2})_{\sum_{\alpha \in B_n} k_\alpha}} \prod_{\alpha \in B_n} \frac{u_\alpha^{k_\alpha}}{k_\alpha!} \,,$$

(9.3)

where $B_n = \{12, 13, 23, \dots, (n-1, n)\}$ is the set of all ordered pairs of distinct numbers between 1 and $n$, whereas $B_{n|j}$ is the subset of $B_n$ which is comprised of pairs containing $j$.

We note that the Feynman parametrization of the corresponding dual conformal all-mass $n$-point integrals in terms of the variables (9.2) is given by

$$I_{n\bullet}^{m_1 \dots m_n} = \pi^{\frac{D}{2}} \Gamma_{D/2} \left( \prod_{i=2}^{n} \int_0^\infty d\alpha_i \right) \left( \prod_{i=1}^{n} \frac{\alpha_i^{a_i-1}}{\Gamma_{a_i} m_i^{a_i}} \right) \left( \sum_{i<j=1}^{n} 2\alpha_i \alpha_j (1 - 2u_{ij}) + \sum_{i=1}^{n} \alpha_i^2 \right)^{-D/2} \Bigg|_{\alpha_1=1} \,.$$

(9.4)

Let us present our evidence for the correctness of the above series representation at lower points.

**Evidence for the Conjecture**

$n = 1$.  For $n = 1$, the entire sum collapses, leaving only

$$I_{1\bullet}^{m_1} = \frac{\pi^{D/2}\Gamma_{D/2}}{\Gamma_D m_1^{a_1}},$$
(9.5)

in agreement with (7.4).

$n = 2$.  In the two-point case, the expression reads

$$I_{2\bullet}^{m_1 m_2} = \frac{\pi^{D/2}\Gamma_{D/2}}{\Gamma_D m_1^{a_1} m_2^{a_2}} \sum_{k_{12}=0}^{\infty} \frac{(a_1)_{k_{12}}(a_2)_{k_{12}}}{(\frac{D+1}{2})_{k_{12}}} \frac{u_{12}^{k_{12}}}{k_{12}!} = \frac{\pi^{D/2}\Gamma_{D/2}}{\Gamma_D m_1^{a_1} m_2^{a_2}} \, {}_2F_1\left[\begin{matrix} a_1, a_2 \\ (D+1)/2 \end{matrix}; u_{12}\right],$$
(9.6)

which agrees with (8.23).

$n = 3$.  At three points, the proposed formula coincides with the expression using Srivastava's triple hypergeometric function $H_C$ given in (8.62):

$$\begin{aligned} I_{3\bullet}^{m_1 m_2 m_3} &= \frac{\pi^{D/2}\Gamma_{D/2}}{\Gamma_D m_1^{a_1} m_2^{a_2} m_3^{a_3}} \sum_{k_{12},k_{13},k_{23}=0}^{\infty} \frac{(a_1)_{k_{12}+k_{13}}(a_2)_{k_{12}+k_{23}}(a_3)_{k_{13}+k_{23}}}{(\frac{D+1}{2})_{k_{12}+k_{13}+k_{23}}} \frac{u_{12}^{k_{12}} u_{13}^{k_{13}} u_{23}^{k_{23}}}{k_{12}!k_{13}!k_{23}!} \\ &= \frac{\pi^{D/2}\Gamma_{D/2}}{\Gamma_D m_1^{a_1} m_2^{a_2} m_3^{a_3}} H_C\left[\begin{matrix} a_1, a_2, a_3 \\ D/2 \end{matrix}; u_{12}, u_{13}, u_{23}\right]. \end{aligned}$$
(9.7)

$n = 4$.  At four points the series representation is supported by numerical comparison with the Feynman parametrization. We have also checked that the conjecture provides a solution to the $\widehat{P}$ Yangian PDEs, cf. Appendix B.

$n = 5$.  The conjectured series agrees with the numerical evaluation of the above Feynman parametrization.

## 9.2  $n$ Points, $m_1 \ldots m_n$: Conjecture in Region B

The above results suggest a second series representation, which is closely related to a representation given in [34]. This series does not converge in the Euclidean region of the Feynman integrals that we have been focussing on so far, but its analytical continuation is conjectured to agree with the integral. The B-series is expressed in terms of the variables

$$v_{ij} = \frac{x_{ij}^2 + m_i^2 + m_j^2}{2m_i m_j}.$$
(9.8)

For Euclidean choices of $x_{ij}$ and $m_j$, these variables do not become small, which would correspond to the kinematic region where we expect the below series to converge. The conjectured series representation reads

$$I_{n\bullet}^{m_1 \ldots m_n} = \frac{\pi^{D/2}}{2^{n-1}} \sum_{k_{12},k_{13},\ldots,k_{n-1,n}=0}^{\infty} \prod_{j=1}^{n} \frac{\Gamma_{\frac{1}{2}}(a_j + \sum_{\alpha \in B_{n|j}} k_\alpha)}{m_j^{a_j} \Gamma_{a_j}} \prod_{\alpha \in B_n} (-2)^{k_\alpha} \frac{v_\alpha^{k_\alpha}}{k_\alpha!}.$$
(9.9)

Here again $B_n = \{12, 13, 23, \ldots, (n-1, n)\}$ represents the set of all ordered pairs of distinct numbers between 1 and $n$ and $B_{n|j}$ is the subset of $B_n$ which contains all pairs containing $j$.

Again we note that the Feynman parameter representation of the conformal massive $n$-gon integrals in terms of the variables (9.8) takes the form[12]

$$I_{n\bullet}^{m_1...m_n} = \pi^{D/2}\Gamma_{D/2}\left(\prod_{i=2}^n \int_0^\infty d\alpha_i\right)\left(\prod_{i=1}^n \frac{\alpha_i^{a_i-1}}{\Gamma_{a_i}m_i^{a_i}}\right)\left(\sum_{i<j=1}^n 2\alpha_i\alpha_j v_{ij} + \sum_{i=1}^n \alpha_i^2\right)^{-D/2}\Bigg|_{\alpha_1=1}. \quad (9.10)$$

For the series (9.9) this Feynman representation is particularly useful to numerically verify examples of the conjectured representation, since for real values of the $m_j$ and $x_{jk}$ the variables $v_{ij}$ in the Euclidean signature do not become small and the series does not converge.

**Evidence for the Conjecture**

$n = 1$.  As in the case of the series representation in region A, for $n = 1$ the sum collapses and agrees with (7.4).

$n = 2$.  The result agrees with(8.18) that we found from Yangian symmetry and with the numerical Feynman parameter integration.

$n = 3$.  The result agrees with (8.76) that we found from Yangian symmetry and with the numerical Feynman parameter integration.

$n = 4$.  The conjectured series agrees with the numerical evaluation of the above Feynman parametrization. Moreover, it provides a solution to the $\widehat{P}$ Yangian PDEs, cf. Appendix B.

$n = 5$.  The conjectured series agrees with the numerical evaluation of the above Feynman parametrization.

## 9.3   Unit Propagator Powers for 2,3 and 4 Points

In this section we compare the above A- and B-series to some lower point expressions for the Yangian invariant all-mass integrals with unit propagator powers.

**A-Series for 2 Points.**   The unit propagator power limit of the result in Section 8.2 in $D = 2$ reads

$$I_{2\bullet}^{m_1m_2} = \frac{2\pi}{m_1^{-a_1}m_2^{-a_2}}(1-v_{12}^2)^{-1/2}\arcsin\left(\sqrt{(1-v_{12})/2}\right). \quad (9.11)$$

On the other hand, we can evaluate the A-series for $a_1 = a_2 = 1$ with $D = 2$ and using $v_{12} = -2u_{12} + 1$. This yields the relation

$${}_2F_1\left[\begin{smallmatrix}1,1\\3/2\end{smallmatrix};\frac{1-v_{12}}{2}\right] = 2(1-v_{12}^2)^{-1/2}\arcsin\left(\sqrt{(1-v_{12})/2}\right). \quad (9.12)$$

**B-Series for 2 Points.**   For the B-series the relation in (8.18) implies that

$$\frac{1}{2}\sum_{k=0}^\infty (-2)^k\Gamma_{k/2+1/2}^2\frac{v_{12}^k}{k!} = 2(1-v_{12}^2)^{-1/2}\arcsin\left(\sqrt{(1-v_{12})/2}\right). \quad (9.13)$$

---

[12]The representation in terms of the variables $u_{ij}$ given in (9.2) is obtained by replacing $v_{ij} = 1 - 2u_{ij}$.

**A-Series for 3 Points: Comparison with Nickel.** In [55] Nickel has computed the all-mass three-point integral in the $v$-type variables for propagator powers $a_j = 1$ in $D = 3$ dimensions and found

$$I_{3\bullet a_j=1}^{m_1 m_2 m_3} = \pi^2 \frac{\phi_N(v_{12}, v_{13}, v_{23})}{m_1^{a_1} m_2^{a_2} m_3^{a_3}}, \tag{9.14}$$

where[13]

$$\phi_N(v_{12}, v_{13}, v_{23}) = \frac{1}{\sqrt{\det \mathcal{G}}} \arctan\left(\frac{\sqrt{\det \mathcal{G}}}{1 + v_{12} + v_{13} + v_{23}}\right). \tag{9.15}$$

Here the matrix $\mathcal{G}$ is defined to have matrix elements $\mathcal{G}_{jk} = v_{jk}$. We can compare this result to the above 3-point result in $u$-type variables (9.7). Here we note that $v_{jk} = -2u_{jk} + 1$. and for the prefactor of the series expression in $D = 3$ have $\Gamma_{3/2}/\Gamma_3 = \sqrt{\pi}/4$. We thus conclude that in the region of convergence (8.61) for $H_C$ we have

$$\phi_N(v_{12}, v_{13}, v_{23}) = \tfrac{1}{4} H_C\left[\begin{smallmatrix} 1,1,1 \\ 2 \end{smallmatrix}; u_{12}, u_{13}, u_{23}\right]. \tag{9.16}$$

Expanding the left hand side in Mathematica assuming $0 < u_{jk} < 1$ indeed shows that at least up to and including order 8 in the variables $u_{12}, u_{13}, u_{23}$, the expansions of both sides of this equation coincide.

**B-Series for 3 Points: Comparison with Nickel.** Similarly, we can compare the above result (9.15) by Nickel with the B-series (9.10), which is actually formulated in the same $v$-type variables. Also here we find agreement at leading orders when expanding the two expressions, i.e.

$$\phi_N(v_{12}, v_{13}, v_{23}) = \frac{1}{\sqrt{\pi}} \sum_{k,l,n=0}^{\infty} (-2)^{k+l+n} \Gamma_{k/2+l/2+1/2} \Gamma_{k/2+n/2+1/2} \Gamma_{l/2+n/2+1/2} \frac{v_{12}^k}{k!} \frac{v_{13}^l}{l!} \frac{v_{23}^n}{n!}. \tag{9.17}$$

**B-Series at 4 Points: Comparison with Murakami–Yano.** In [35] the Murakami–Yano formula [57, 58], which gives a compact expression for the volume of a hyperbolic/spherical tetrahedron, has been leveraged to give a concise dilogarithmic expression for the all-mass box integral in four dimensions with unit propagator powers. This result provides a valuable cross check of our series representation (9.9), which we deem worth detailing.

The volume of a spherical tetrahedron is most elegantly phrased in terms of its dihedral angles. We therefore begin by making explicit the relation between these angles and our variables $v_{ij}$. Let $\mathcal{G}$ be the matrix whose elements are given by the variables $v_{ij}$, i.e.

$$\mathcal{G}_{ij} = v_{ij}. \tag{9.18}$$

The matrix $\mathcal{G}$ encodes the distances between all pairs of points forming the tetrahedron. The dihedral angles are readily obtained by employing the formula

$$\theta_{ij} = \arccos\left(-\frac{(\mathcal{G}^{-1})_{ij}}{\sqrt{(\mathcal{G}^{-1})_{ii}} \sqrt{(\mathcal{G}^{-1})_{jj}}}\right), \tag{9.19}$$

see [35] for a detailed discussion of the underlying geometry. Given these angular variables, we define

$$c_1 = e^{i\theta_{12}}, \quad c_2 = e^{i\theta_{13}}, \quad c_3 = e^{i\theta_{23}}, \quad c_4 = e^{i\theta_{34}}, \quad c_5 = e^{i\theta_{24}}, \quad c_6 = e^{i\theta_{14}}. \tag{9.20}$$

---

[13]For general $D$ and propagator powers 1, the integral can be written in terms of Appell functions $F_1$ and arctan's [56].

The expression for the volume of a spherical tetrahedron makes use of the positive root

$$z_+ = \frac{-q_1 + \sqrt{q_1^2 - 4q_0 q_2}}{2q_2}, \tag{9.21}$$

of the quadratic equation $q_2 z^2 + q_1 z + q_0 = 0$, where

$$q_0 = c_1 c_2 c_3 c_4 c_5 c_6 + c_1 c_2 c_6 + c_1 c_3 c_5 + c_2 c_3 c_4 + c_4 c_5 c_6 + \sum_{i=1}^{3} c_i c_{i+3}, \tag{9.22}$$

$$q_1 = -\sum_{i=1}^{3} \left(c_i - c_i^{-1}\right)\left(c_{i+3} - c_{i+3}^{-1}\right),$$

$$q_1 = (c_1 c_2 c_3 c_4 c_5 c_6)^{-1} + (c_1 c_2 c_6)^{-1} + (c_1 c_3 c_5)^{-1} + (c_2 c_3 c_4)^{-1} + (c_4 c_5 c_6)^{-1} + \sum_{i=1}^{3}(c_i c_{i+3})^{-1}.$$

Furthermore, we require the function

$$L(z) = \frac{1}{2}\left[ \mathrm{Li}_2(z) + \mathrm{Li}_2\left(\frac{z}{c_1 c_2 c_4 c_5}\right) + \mathrm{Li}_2\left(\frac{z}{c_1 c_3 c_4 c_6}\right) + \mathrm{Li}_2\left(\frac{z}{c_2 c_3 c_5 c_6}\right) - \mathrm{Li}_2\left(-\frac{z}{c_1 c_2 c_3}\right) \right.$$

$$\left. - \mathrm{Li}_2\left(-\frac{z}{c_1 c_5 c_6}\right) - \mathrm{Li}_2\left(-\frac{z}{c_2 c_4 c_6}\right) - \mathrm{Li}_2\left(-\frac{z}{c_3 c_4 c_5}\right) + \sum_{i=1}^{3} \log(c_i)\log(c_{i+3}) \right]. \tag{9.23}$$

In terms of the function $L(z_+)$, the volume of the tetrahedron described by the matrix $\mathcal{G}$ is given by

$$V_4(\mathcal{G}) = -\mathrm{Re}(L(z_+)) + \pi\left(\arg(-q_2) + \tfrac{1}{2}\sum_{i<j}\theta_{ij}\right) - \frac{3\pi^2}{2} \quad (\mathrm{mod}\ 2\pi^2). \tag{9.24}$$

Utilizing this expression, the all-mass box integral can be expressed as

$$I_{4\bullet a_j=1}^{m_1 m_2 m_3 m_4} = \frac{\pi^2 V_4(\mathcal{G})}{\sqrt{\det \mathcal{G}^0}}, \tag{9.25}$$

where $\mathcal{G}_{ij}^0 = \mathcal{G}_{ij} m_i m_j$. We find this expression to be in perfect numerical agreement with the series representation (9.9) in the Euclidean domain.

# 10 Outlook

The results presented in this paper suggest plenty of different directions for further investigation. Let us detail a few.

In the case of one-loop Feynman integrals we have demonstrated that Yangian symmetry is in fact highly constraining. For the simple examples studied in Section 7 and Section 8, this results in a small basis of hypergeometric series whose linear combination yields the integral under study. Here the number of basis elements depends on the chosen variables as can for instance be seen in Section 7.4 and Section 7.5, where the same integral is studied for two different choices of variables. In particular, the solution basis is generically expected to grow with the number of variables, as becomes apparent in the case of the 9-variable massless double box and hexagon integrals considered in [19]. Here the close connection to the Mellin–Barnes approach deserves further study. In particular, it would be desirable to find a symmetry principle that selects the specific subsets of formal Yangian invariants that span the solution, see

[20]. Eventually it seems plausible that to fix these linear combinations using only conformal Yangian symmetry, kinematical configurations with on-shell external particles have to be taken into account. These may result in deformations of the symmetry generators similar to the case of scattering amplitudes in $\mathcal{N} = 4$ SYM and ABJM theory and thus in additional relations, cf. [59–64].

In certain cases and for particular choices of the conformal variables, the Yangian bootstrap selects a single series solution to the symmetry constraints. In these cases merely an overall coefficient remains to be fixed in order to determine a representation of the integral. In particular, this is the case for the all-mass $n$-gon integrals subject of Section 9 and allowed us to conjecture two different single series representations (9.3) and (9.9) for generic one-loop integrals in $D$ spacetime dimensions. While the outstanding properties of these integrals have been studied in the past for unit propagator powers (cf. e.g. [35]), the novel Yangian symmetry sheds new light on their distinguished role. It would be very interesting to further explore the space of Yangian invariant integrals in order to identify families with similarly beautiful properties. Here the next step is to proceed to two loops. While the analysis of the massless double box in [19] shows the increase of complexity for a higher number of external points, the present paper suggests that it may be beneficial to first consider situations with massive propagators. With regard to the one-loop integrals, it would be interesting to better understand the mathematical properties of the two series representations (9.3) and (9.9). While we have not found a name for the B-series that closely resembles that given in [34], at least for $n = 2$ and $n = 3$ points the A-series coincides with Gauß' hypergeometric function $_2F_1$ and Srivastava's triple hypergeometric function $H_C$, respectively, and thus can be assumed to represent a useful generalization.

When proceeding to more complicated examples, we note that at loop orders beyond two, the statements about level-one Yangian symmetry are still conjectural, cf. Table 2 and [3]. It would be important to make progress on understanding these cases in detail. Ideally one could find an analytic proof similar to the one in the massless case using the Lax operator formalism [2]. Alternatively, it would be interesting to systematically map out the space of higher loop integrals using advanced numerical integration techniques. Into this direction it would also be interesting to study massive Feynman integrals with particles different from scalars which appear in non-scalar fishnet theories [65–67]. In fact, in the massless case certain brick wall Feynman graphs including fermionic lines were found to be Yangian invariant [11] and thus represent a natural starting point.

Physical application of our results asks for an extension into two further directions. Firstly, while Feynman integrals with generic propagator powers are clearly of interest, it would be desirable to better understand the considered Yangian approach for integer (in particular for unit) propagator powers, which at one loop order results in polylogarithmic expressions. Here the situation of the massless box integral was explicitly discussed in [19], which underlines the importance of the choice of kinematic variables. Secondly, while we have focussed on the case of Euclidean spacetime signature, there is an obvious demand to explicitly discuss the results in Minkowski signature. Though this step should not modify the Yangian constraints (and thus the solution basis), identifying the correct linear combination needs more care. For the case of the massless box integral, the results of [68] indeed show that also in Minkowski spacetime the integral is spanned by the Yangian invariant building blocks given in [19].

The four-point Basso–Dixon diagrams of [69] represent another example of Feynman in-

tegrals with an intriguing connection to integrability, see also [70–72]. Using sophisticated integrability techniques from AdS/CFT and the Steinmann relations, a conjecture for the polylogarithmic result for these integrals was given at generic loop order. While a Yangian symmetry has not been formulated in this case, these integrals can be understood as coincident point limits of the more generic Yangian invariant fishnet integrals discussed in [2,11]. If one assumes a connection between this Yangian symmetry and the simplicity of the expressions given by Basso and Dixon, one may wonder whether massive propagators can be introduced into their formula.

As mentioned in the introduction, the conformal Yangian and its massive generalization studied here can be considered as the closure of two distinct conformal algebras. As such, it would be very interesting to identify its place within the large landscape of results on the conformal and momentum space conformal bootstrap. Clearly, the Yangian constraints can be studied independently of Feynman integrals and it would be interesting to search for further applications. These might correspond to lifting a conformal setup to an integrable one. Even if one does not consider the full Yangian, it seems natural to generalize the various applications of the momentum space conformal bootstrap (see e.g. [31] and references therein) to the massive extension of the algebra given in (5.10).

## Acknowledgements

We are grateful to Cristian Vergu for illuminating discussions and for providing us with an implementation of the Murakami–Yano formula. The work of FL is funded by the Deutsche Forschungsgemeinschaft (DFG, German Research Foundation)–Projektnummer 363895012. JM is supported by the International Max Planck Research School for Mathematical and Physical Aspects of Gravitation, Cosmology and Quantum Field Theory. DM was supported by DFF-FNU through grant number DFF-FNU 4002-00037. The work of HM was supported by the grant no. 615203 of the European Research Council under the FP7 and by the Swiss National Science Foundation by the NCCR SwissMAP.

## A  Yangian Level-One Generators

We provide the expressions for all Yangian level-one generators over the (dual) conformal algebra:

$$\widehat{\mathrm{P}}^{\mu} = \tfrac{i}{2}\sum_{j<k}\Big(\mathrm{P}_j^{\mu}\mathrm{D}_k + \mathrm{P}_{j\nu}\mathrm{L}_k^{\mu\nu} - (j\leftrightarrow k)\Big) + \sum_{j=1}^{n}s_j\mathrm{P}_j^{\mu},$$

$$\widehat{\mathrm{L}}^{\mu\nu} = \tfrac{i}{2}\sum_{j<k}\Big(\tfrac{1}{2}(\mathrm{P}_j^{\mu}\mathrm{K}_k^{\nu} - \mathrm{P}_j^{\nu}\mathrm{K}_k^{\mu}) + \mathrm{L}_k^{\mu\rho}\mathrm{L}_{j\rho}{}^{\nu} - (j\leftrightarrow k)\Big) + \sum_{j=1}^{n}s_j\mathrm{L}_j^{\mu\nu},$$

$$\widehat{\mathrm{D}} = \tfrac{i}{4}\sum_{j<k}\Big(\mathrm{P}_{j\mu}\mathrm{K}_k^{\mu} - (j\leftrightarrow k)\Big) + \sum_{j=1}^{n}s_j\mathrm{D}_j,$$

$$\widehat{\mathrm{K}}^{\mu} = \tfrac{i}{2}\sum_{j,k=1}^{n}\Big(\mathrm{D}_j\mathrm{K}_k^{\mu} + \mathrm{K}_{j\nu}\mathrm{L}_k^{\mu\nu} - (j\leftrightarrow k)\Big) + \sum_{j=1}^{n}s_j\mathrm{K}_j^{\mu}. \tag{A.1}$$

The extra generators are given by

$$\widehat{P}^\mu_{\text{extra}} = \frac{i}{2}\sum_{j<k}\left(P_{jD+1}L^{\mu D+1}_k - (j\leftrightarrow k)\right), \qquad \widehat{L}^{\mu\nu}_{\text{extra}} = \frac{i}{2}\sum_{j<k}\left(L^{\mu D+1}_k L_{jD+1}{}^\nu - (j\leftrightarrow k)\right),$$

$$\widehat{K}^\mu_{\text{extra}} = \frac{i}{2}\sum_{j,k=1}^n\left(K_{jD+1}L^{\mu D+1}_k - (j\leftrightarrow k)\right), \qquad \widehat{D}_{\text{extra}} = \frac{i}{4}\sum_{j<k}\left(P_{jD+1}K^{D+1}_k - (j\leftrightarrow k)\right). \qquad (A.2)$$

## B  All-Mass Yangian PDEs for 4 Points

Here we give some details on the all-mass Yangian constraints for four points.

**Region A.**  We write the conformal four-point integral as

$$I^{m_1 m_2 m_3 m_4}_{4\bullet} = \frac{\phi(u_{12},u_{13},u_{23},u_{14},u_{24},u_{34})}{m^{a_1}_1 m^{a_2}_2 m^{a_3}_3 m^{a_4}_4}, \qquad (B.1)$$

with $u_{jk}$ as defined in (9.2). From the $\widehat{P}$ invariance equation we can read off the coefficients of the vectors $x^\mu_{jk}/m_j m_k$ to find the annihilators of the function $\phi$, e.g.

$$\begin{aligned}
\text{PDE}^{\widehat{P}_{(y=0)}}_{12} =\,& \partial_{u_{14}}\partial_{u_{24}} + \partial_{u_{13}}\partial_{u_{23}} - 2(a_3+a_4)\partial_{u_{12}} + (2-4u_{34})\partial_{u_{13}}\partial_{u_{24}} \\
&+ (1-2u_{24})\partial_{u_{12}}\partial_{u_{24}} + (1-2u_{14})\partial_{u_{12}}\partial_{u_{14}} + (1-2u_{23})\partial_{u_{12}}\partial_{u_{23}} \\
&+ (1-2u_{13})\partial_{u_{12}}\partial_{u_{13}}.
\end{aligned} \qquad (B.2)$$

When applied to the series ansatz

$$\phi = \sum_{\{k_{ij}\}} f_{k_{12}k_{13}k_{23}k_{14}k_{24}k_{34}}\, u^{k_{12}}_{12} u^{k_{13}}_{13} u^{k_{23}}_{23} u^{k_{14}}_{14} u^{k_{24}}_{24} u^{k_{34}}_{34}, \qquad (B.3)$$

the partial differential equations $\text{PDE}_{jk}\phi = 0$ translate into recurrence equations, e.g. for $\text{PDE}^{\widehat{P}_{(y=0)}}_{12}$

$$\begin{aligned}
0 =\,& -2(k_{12}+1)(a_3+a_4+k_{13}+k_{14}+k_{23}+k_{24})f_{k_{12}+1,k_{13}k_{23}k_{14}k_{24},k_{34}} \qquad (B.4)\\
& -2(k_{13}+1)(k_{24}+1)[2f_{k_{12},k_{13}+1,k_{23},k_{14},k_{24}+1,k_{34}-1} - f_{k_{12},k_{13}+1,k_{23},k_{14},k_{24}+1,k_{34}}] \\
& +(k_{13}+1)(k_{23}+1)f_{k_{12},k_{13}+1,k_{23}+1,k_{14}k_{24},k_{34}} + (k_{14}+1)(k_{24}+1)f_{k_{12},k_{13},k_{23},k_{14}+1,k_{24}+1,k_{34}} \\
& +(k_{12}+1)(k_{24}+1)f_{k_{12}+1,k_{13}k_{23}k_{14},k_{24}+1,k_{34}} + (k_{12}+1)(k_{14}+1)f_{k_{12}+1,k_{13}k_{23},k_{14}+1,k_{24},k_{34}} \\
& +(k_{12}+1)(k_{23}+1)f_{k_{12}+1,k_{13},k_{23}+1,k_{14}k_{24}k_{34}} + (k_{12}+1)(k_{13}+1)f_{k_{12}+1,k_{13}+1,k_{23}k_{14}k_{24}k_{34}}.
\end{aligned}$$

It seems not straightforward to solve these recurrences directly, but based on the previous experience we find that they are solved by the fundmantal solution corresponding to our conjectural A-series for $n=4$:

$$f_{k_{12}k_{13}k_{23}k_{14}k_{24}k_{34}} = \frac{(a_1)_{k_{12}+k_{13}+k_{14}}(a_2)_{\hat{k}_{12}+k_{23}+k_{24}}(a_3)_{k_{13}+k_{23}+k_{34}}(a_4)_{k_{14}+k_{24}+k_{34}}}{\Gamma_{k_{12}+1}\Gamma_{k_{13}+1}\Gamma_{k_{23}+1}\Gamma_{k_{14}+1}\Gamma_{k_{24}+1}\Gamma_{k_{34}+1}(\gamma)_{\Sigma_k}}. \qquad (B.5)$$

Here we abbreviate $\gamma = (D+1)/2$ and $\Sigma_k = k_{12}+k_{13}+k_{14}+k_{24}+k_{34}$.

**Region B.** We write

$$I_{4\bullet}^{m_1 m_2 m_3 m_4} = \frac{\phi(v_{12}, v_{13}, v_{23}, v_{14}, v_{24}, v_{34})}{m_1^{a_1} m_2^{a_2} m_3^{a_3} m_4^{a_4}}, \tag{B.6}$$

with the $v_{jk}$ as given in (9.8). Acting with the level-one momentum generator, we can read off the annihilators $\text{PDE}_{jk}$ of the function $\phi$ as the coefficients of the vectors $x_{jk}^\mu/m_j m_k$ to find for instance

$$\begin{aligned}
\text{PDE}_{12}^{\widehat{P}_{(y=0)}} =& (a_3 + a_4)\partial_{u_{12}} + \partial_{u_{14}}\partial_{u_{24}} + \partial_{u_{13}}\partial_{u_{23}} + 2u_{34}\partial_{u_{13}}\partial_{u_{24}} \\
&+ u_{24}\partial_{u_{12}}\partial_{u_{24}} + u_{14}\partial_{u_{12}}\partial_{u_{14}} + u_{23}\partial_{u_{12}}\partial_{u_{23}} + u_{23}\partial_{u_{12}}\partial_{u_{13}}.
\end{aligned} \tag{B.7}$$

Making the series ansatz

$$\phi = \sum_{\{k_{ij}\}} f_{k_{12}k_{13}k_{23}k_{14}k_{24}k_{34}} u_{12}^{k_{12}} u_{13}^{k_{13}} u_{23}^{k_{23}} u_{14}^{k_{14}} u_{24}^{k_{24}} u_{34}^{k_{34}}, \tag{B.8}$$

the invariance equation $\text{PDE}_{12}^{\widehat{P}_{(y=0)}}\phi = 0$ for instance is transformed into the recurrence equation

$$\begin{aligned}
0 =& + (k_{12}+1)(a_3 + a_4 + k_{13} + k_{14} + k_{23} + k_{24}) f_{k_{12}+1,k_{13}k_{23}k_{14}k_{24}k_{34}} \\
&+ (k_{14}+1)(k_{24}+1) f_{k_{12}k_{13}k_{23},k_{14}+1,k_{24}+1,k_{34}} + 2(k_{13}+1)(k_{24}+1) f_{k_{12},k_{13}+1,k_{23}k_{14},k_{24}+1,k_{34}-1} \\
&+ (k_{13}+1)(k_{23}+1) f_{k_{12},k_{13}+1,k_{23}+1,k_{14}k_{24}k_{34}}.
\end{aligned} \tag{B.9}$$

This is relation is indeed solved by the four-point version of (9.9) given by

$$f_{k_{12}k_{13}k_{23}k_{14}k_{24}k_{34}} = (-2)^{\Sigma_k} \frac{(\hat{a}_1)_{\hat{k}_{12}+\hat{k}_{13}+\hat{k}_{14}} (\hat{a}_2)_{\hat{k}_{12}+\hat{k}_{23}+\hat{k}_{24}} (\hat{a}_3)_{\hat{k}_{13}+\hat{k}_{23}+\hat{k}_{34}} (\hat{a}_4)_{\hat{k}_{14}+\hat{k}_{24}+\hat{k}_{34}}}{\Gamma_{k_{12}+1}\Gamma_{k_{13}+1}\Gamma_{k_{23}+1}\Gamma_{k_{14}+1}\Gamma_{k_{24}+1}\Gamma_{k_{34}+1}}, \tag{B.10}$$

where we abbreviate $\hat{k} = k/2$ and $\Sigma_k = k_{12} + k_{13} + k_{14} + k_{24} + k_{34}$.

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
