# Peer review of "Yangian Bootstrap for Massive Feynman Integrals"

_SciPost Physics, doi:SciPost Phys. 11, 010 (2021)_

## Round 1 · Referee Report · Anonymous (Referee 1) · 2021-4-13

Report

This paper is an exploration of Yangian extension of dual conformal symmetry of certain Feynman loop integrals and the differential equations which follow from it. The symmetry is verified on the integral form and then exploited to derive constraints for series representations of the resulting integrals.

Many examples are discussed from different perspectives and some potentially useful general series representations are conjectured, which are verified in a number of cases by a variety of analytic and numerical means.

The paper represents a useful addition to the study of Feynman integrals and the relation to Yangian symmetry and I recommend publication after the following minor points are clarified.

  1. Below (3.1) it is stated that the commutators of level one generators are constrained by Serre relations. It is not clear if these relations have been checked for the representations introduced just below in eqs. (3.2) and below. In particular for (3.2) to hold it requires a certain relation cubic in the level zero generators on a single site.

  2. In the discussion of invariance at one loop, the generators are applied to the integrand. In fact only two propagator factors are relevant here. In eq. (3.20) the action is given to obtain a vanishing after applying the level one generators on two sires. Strictly speaking this zero is only for non-coincident points. In principle there could be contact contributions which could spoil invariance, as happens for example on applying the Laplace operator (a different second order operator) to the massless propagator. Some argument why such contributions vanish is missing.

  3. Both at one loop and two loops the cases of non-dual conformal invariant integrals are discussed. One way to relate conformally invariant integrals and non-invariant ones is simply to send a point to infinity. This operation can relate many non-invariant integrals to invariant ones. It seems the relations discussed in the paper are unrelated to this mechanism but some comment on this point might clarify the situation.

  • validity: -
  • significance: -
  • originality: -
  • clarity: -
  • formatting: -
  • grammar: -

Author:  Julian Miczajka  on 2021-05-12  [id 1421]

(in reply to Report 1 on 2021-04-13)

We thank the Referee for the useful comments and the positive feedback.

Regarding the requested changes we made the following modifications:

  1. Below (3.1) we added the following footnote: "We have not verified whether the generators in (2.13) satisfy the Serre relations. For our bootstrap purposes below we have solely used the level-zero and level-one symmetries without an appeal to the infinite tower of Yangian generators."

  2. We thank the referee for pointing out the possibility of contact terms. We note that in the completely massive case, the propagators are completely regular even at the contact point. At this point we don't have a rigorous argument that excludes contact terms in the massless case. We can rule out the appearance of operators like the Laplace-operator, since the Yangian generator contains first order derivatives with respect to the respective external points only. Certainly, for the cases we have bootstrapped in section 4, contact terms don't seem to play any role, since our results agree with earlier direct calculations. Similar statements hold for lower point massless diagrams analysed from the Yangian perspective in earlier papers. Above (3.20) we added the following footnote: "With regard to potential contact terms as they arise from the second order Laplace operator we note that P_jk is a product of first order differential operators each acting on a single leg j or k only. In the fully massive case, the propagators are completely regular even at the contact point."

  3. Above (3.60) we added the following footnote: "This procedure should not be confused with another way to acquire a nondual-conformal integral from a dual-conformal one: sending one of the external points to infinity, the corresponding propagators drop out of the adequately normalized integral, turning a conformal vertex into a non-conformal one."

For clarification we also added the following sentence to point 5. in section 4.2: "These constraints can come from e.g. permutation symmetry or kinematic limits that simplify the form of the integral, such as sending a mass to zero or an external point to infinity."

---

## Round 1 · Referee Report · Anonymous (Referee 2) · 2021-4-26

Strengths

  1. The presentation in the paper is very clear and self-contained. The paper provides detailed proofs of the statements from the Letter arXiv:2005.01735 by the same authors.
  2. The volume of the paper is adequate for the number of results it contains.
  3. The study implemented by the authors is thorough. Numerous aspects of the massive Yangian symmetry of the scalar Feynman integrals are elucidated and the connection between the massive Yangian symmetry formulations in momentum and dual-momentum variables is exposed.
  4. The paper contains numerous examples. They help to demonstrate different aspects of the Yangian bootstrap in the momentum and dual-momentum representations.
  5. The authors show how their Yangian bootstrap approach allows to uniformly rederive numerous analytic results available in the literature on the Feynman integrals with arbitrary propagator. They provide relevant references and compare different analytic expressions for the Feynman integrals.

Weaknesses

  1. The majority of the provided examples of massive Feynman integrals are limited to the one-loop approximation, and they have been previously evaluated (in a different form and using different approaches). Nevertheless, the aim of the authors was not to obtain state-of-the-art results on multi-loop and multi-scale Feynman integrals, but to investigate a new bootstrap approach to their calculation.

Report

This paper elaborates on the Letter arXiv:2005.01735 by the same authors and sets up, in great detail, a formalism for using a generalization of the Yangian symmetry to obtain analytic expressions for certain D-dimensional planar off-shell Feynman integrals with massive propagators. The authors provide a detailed exposition of the massive Yangian algebra introduced in the Letter, and illustrate by numerous examples how the symmetry constraints can be efficiently used to obtain concise expressions for a number of 2-,3-, and n-point one-loop Feynman integrals involving several internal masses and having generic propagator powers (subjected to the Yangian symmetry constraint). The latter calculation technique is dubbed the Yangian bootstrap and it generalizes the approach previously developed by a subset of the authors for Yangian-symmetric Feynman integrals with massless propagators. The massive Yangian symmetry is formulated both in momentum variables and dual-momentum variables, and the connection between the two formulations is made explicit. Also, the authors prove that the bilocal level-one generators of the massive Yangian algebra annihilate a generic one-loop Feynman integral (which is not symmetric under the action of the full massive Yangian algebra), and they show how to use this constraint to bootstrap such Feynman integrals.

The paper is well-written, and it fully meets the acceptance criteria of the Journal. The paper presents a new calculation method for massive Feynman integrals, which could be of particular interest in condensed matter physics. The method itself certainly deserves further studies. The obtained results are of interest to the integrability community and they augment the knowledge on the Yangian symmetry in QFT. Therefore, I am happy to recommend publication of the manuscript in the Journal.

Requested changes

I would like to make several minor remarks: 1. Discussing the multi-loop integrals, the authors should mention explicitly that the Yangian symmetry is applicable only in the planar sector. 2. There is a typo on page 26, the 4-th line from the bottom (repetition “to to”).

  • validity: high
  • significance: good
  • originality: high
  • clarity: high
  • formatting: excellent
  • grammar: perfect

Author:  Julian Miczajka  on 2021-05-12  [id 1422]

(in reply to Report 2 on 2021-04-26)

We thank the Referee for the useful comments and the positive feedback.

Regarding the requested changes we made the following modifications:

  1. Above (3.49) we added the following sentence: "Notably, all integrals at higher loops that are expected to be Yangian invariant are related to planar diagrams. At this point there is no evidence that the Yangian symmetry of single Feynman diagrams can be generalized beyond planar integrals."

  2. We fixed the typo on page 26.

For clarification we also added the following sentence to point 5. in section 4.2: "These constraints can come from e.g. permutation symmetry or kinematic limits that simplify the form of the integral, such as sending a mass to zero or an external point to infinity."

---

## Round 2 · Referee Report · Anonymous (Referee 2) · 2021-6-28

Report

I thank the authors for adding sentences along the lines of my suggestions. I recommend publication of the manuscript in its current form.

---

## Round 2 · Referee Report · Anonymous (Referee 1) · 2021-6-29

Report

The authors have addressed all the points made in the original report. I am happy to recommend publication.

---

## Round 2 · List of Changes

• Added footnote 3 above (3.2).
  • Added footnote 4 above (3.20).
  • Added sentence "Notably, all integrals at higher loops that are expected to be Yangian invariant are related to planar diagrams. At this point there is no evidence that the Yangian symmetry of single Feynman diagrams can be generalized beyond planar integrals." at the end of section 3.2.
  • Added footnote 6 on page 19.
  • Added sentence "These constraints can come from e.g.\ permutation symmetry or kinematic limits that simplify the form of the integral, such as sending a mass to zero or an external point to infinity." to item 5 in section 4.2.
  • Fixed a typo on page 26 (repetition "to to").
  • Fixed a type in equation (7.9).

---

## Editorial Decision

published